# Shifts in energy fluxes linked to drainage-induced changes in permafrost ecosystem structure increase Bowen Ratios, but reduce thaw depth

Mathias Göckede[1], Fanny Kittler[1], Min Jung Kwon[1], Ina Burjack[1], Martin Heimann[1,2], Olaf Kolle[1],
Nikita Zimov[3], Sergey Zimov[3]

[1]Department Biogeochemical Systems, Max-Planck Institute for Biogeochemistry, Jena, Germany
[2]Division of Atmospheric Sciences, Department of Physics, University of Helsinki, Finland
[3]North-East Science Station, Pacific Institute for Geography, Far-Eastern Branch of Russian Academy of Science, Chersky,
Republic of Sakha (Yakutia), Russia

*Correspondence to*: Mathias Göckede (mathias.goeckede@bgc-jena.mpg.de)

**Abstract.** Hydrologic conditions are a key factor in Arctic ecosystems, with strong influences on ecosystem structure and related effects on biogeophysical and biogeochemical processes. With systematic changes in water availability expected for large parts of the Northern high latitude region in the coming centuries, knowledge on shifts in ecosystem functionality triggered by altered water levels is crucial for reducing uncertainties in climate change predictions. Here, we present findings from paired ecosystem observations in Northeast Siberia that comprise a drained and a control site. At the former, the water table has been artificially lowered by up to 30 cm in summer for more than a decade. This sustained primary disturbance in hydrologic conditions has triggered a suite of secondary shifts in ecosystem properties, including vegetation community structure, snow cover dynamics, and radiation budget, all of which influence the net drainage effects. Reduced thermal conductivity in dry organic soils was identified as the dominating drainage effect on energy budget and soil thermal regime. Through this effect, reduced heat transfer into deeper soil layers leads to shallower thaw depths, initially leading to a stabilization of organic permafrost soils, while the long-term effects on permafrost temperature trends still need to be assessed. At the same time, more energy is transferred back into the atmosphere as sensible heat in the drained area, which may trigger a warming of the lower atmospheric surface layer.

**Keywords:** Permafrost, Disturbance, Drainage, Energy Flux

## 1 Introduction

The current state and future evolution of Arctic permafrost, particularly its interactions with the atmosphere, are among the largest uncertainties in our understanding of the Earth's climate system (Schuur and Abbott, 2011; Stocker et al., 2013). The Arctic is one of the most susceptible regions on Earth to climate change (e.g. Serreze et al., 2000; Polyakov et al., 2003; Fyfe et al., 2013; Pithan and Mauritsen, 2014), and altered climate conditions may have enormous consequences for the sustainability of its natural environment (Schuur et al., 2008; Arneth et al., 2010). Interactions between permafrost, climate, hydrology, and ecology have the potential to cause dramatic changes (e.g. McGuire et al., 2002; Hinzman et al., 2005), via mechanisms that are currently poorly monitored and therefore highly unpredictable (Heimann and Reichstein, 2008; van Huissteden and Dolman, 2012; Rawlins et al., 2015). The associated changes in the energy transfer between surface and atmosphere (Eugster et al., 2000; Langer et al., 2011a, b) or the emission patterns of greenhouse gases (Koven et al., 2011) raise the need for experimental studies in this region that address the key uncertainties linked to the functioning of the Arctic in the earth-climate system (Semiletov et al., 2012).

Hydrologic conditions play a pivotal role in shaping Arctic ecosystems. Lakes and rivers in the permafrost region represent an important portion of regional net carbon exchange with the atmosphere, but net emissions over different spatiotemporal scales remain highly uncertain to date (Walter-Anthony et al., 2014; Rasilo et al., 2015). Moreover, the vast carbon pool in northern permafrost regions, currently estimated at 1330-1580 petagrams of organic carbon (Schuur et al., 2015), has accumulated over past millennia since a combination of low mean temperatures and anoxic conditions in areas with flooded or water saturated soils slowed down the decomposition of organic matter (e.g. Ping et al., 2008). In addition to warming, shifts in the water balance in this region are expected to trigger profound changes in the permafrost carbon cycle (e.g. Chapin et al., 2005; Oberbauer et al., 2007). Hydrology can vary at small spatial scales and thus create fine-scale mosaics in surface conditions (Muster et al., 2012; 2013). Even minor differences in mean water levels can impose strong effects on vegetation and microbial community structures or soil thermal regimes (Zona et al., 2011a). It can thus be expected that flooding or draining a site will initiate profound shifts in ecosystem structure, and that the long-term impacts of the disturbance will look very different than the short term changes that can be observed immediately after the event (e.g. Shaver et al., 1992).

Future shifts in hydrologic conditions in the Arctic can be triggered by altered precipitation patterns, where a general trend towards increased rainfall is predicted (Kattsov and Walsh, 2000); however, patterns will vary strongly by region (Huntington, 2006; Bintanja and Selten, 2014), therefore for some areas also lower precipitation can be expected. Geomorphological processes such as subsidence (Jorgenson et al., 2006; O'Donnell et al., 2012) or the formation of a system of connected troughs through the preferential degradation of ice wedges in ice-rich permafrost (Serreze et al., 2000; Liljedahl et al., 2016) can lead to a lateral redistribution of water, and thus create both wetter and drier microsites within a formerly uniform ecosystem. Also, a deepening of the active layer can trigger reductions in waterlogged conditions and wetland extent in permafrost regions (Avis et al., 2011). In-depth insight into the net effect resulting from long-term changes in

hydrology therefore forms a key challenge for improved future predictions of biogeochemical cycles in Arctic ecosystems (e.g. Lupascu et al., 2014).

Over the past two decades, several studies have documented the effect of an experimental manipulation of soil hydrologic conditions on biogeochemical cycles in Arctic ecosystems (e.g. Oechel et al., 1998; Olivas et al., 2010; Natali et al., 2015). Only a few of these studies examined long-term effects of manipulated water tables (e.g. Christiansen et al., 2012; Lupascu et al., 2014; Kittler et al., 2016), and none of them explicitly addressed the net impact of hydrologic disturbance on the energy budget; only shifts in thaw depth (Zona et al., 2011a; Kim, 2015) and indirect effects of heat fluxes on the carbon cycle (Turetsky et al., 2008) were reported. To date, the only study known to the authors that examined a drainage effect on surface-atmosphere energy exchange patterns focused exclusively on the Bowen Ratio, and found no systematic short-term shifts immediately following a drainage disturbance (Merbold et al., 2009).

Our study presents observational evidence of shifts in ecosystem properties and energy fluxes within a wet tussock tundra ecosystem in Northeast Siberia following a decade-long drainage disturbance, providing insight into the sustainability of ice-rich permafrost under future climate change. As a first major objective, we used paired observations within a drainage and a control area to quantify several secondary disturbance effects linked to lower water tables, including changes in vegetation community, radiation budget and soil thermal regime over longer timeframes. Our second objective was to link these shifts in ecosystem properties to year-round eddy-covariance measurements of energy exchange patterns between permafrost ecosystem and atmosphere, and to evaluate potential feedback effects to Arctic warming. Finally, links between heat transfer into deeper layers and summertime thaw depth dynamics were investigated.

## 2 Methods

### 2.1 Study site and drainage experiment

Our field experiments were conducted on a wet tussock tundra site within the floodplain of the Kolyma River (68.61°N, 161.35°E), approximately 15 km south of the town of Chersky in Northeast Siberia. The area was formed by a relatively recent shift of the main channel of the Kolyma (Corradi et al., 2005), therefore the terrain is flat and no thermokarst lakes or natural drainage channels have yet formed in the vicinity of the observation sites. Still, the existing minor differences in terrain height have led to the formation of a fine-scale mosaic in microsite conditions (Figure 1), mostly triggered by redistribution of water from slightly elevated areas towards the minor depressions.

Site hydrology is strongly influenced by a flooding period in spring caused by melting of the local snow cover as well as a northward moving dam of ice floes on the main channel of the Kolyma that delays melt water runoff. Depending on the timing of snowmelt in the tributary watersheds, this process leads to standing water on the site (up to 50 cm above ground level) in most years around late May to early June. After the flood has mostly receded, stagnant water remains in large parts of the area that only gradually decreases over the course of the growing season.

Soils consist of an organic layer of 15-20 cm overlaying alluvial mineral soils (silty clay), with some organic material also present in deeper layers following cryoturbation. More information on site climatology (Section 3.1). spatiotemporal patterns in hydrology (Section 3.2) and vegetation characteristics (Section 3.3) can be found in the Results section. Further details on the study site are presented by Corradi et al. (2005), Merbold et al. (2009) and Kwon et al. (2016).

To study the effects of a lowered water table on a formerly predominantly wet permafrost ecosystem, in the fall of 2004 a drainage ring of approximately 200 m diameter was installed that is connected to the nearby river (Merbold et al. (2009), see also Figure 1). Within its watershed, this drainage system caused systematic changes in the hydrologic regime, lowering the mean water table and shifting the natural wetlands towards conditions with dry soils close to the surface.

## 2.2 Meteorological towers

We established a paired experiment over drained and undisturbed control tundra to facilitate a direct evaluation of drainage effects on ecosystem characteristics and fluxes within this permafrost ecosystem. Since July 2013, two uniformly instrumented eddy-covariance towers with instrument heights around 5 m a.g.l. (Figure 1) captured the turbulent exchange processes between surface and atmosphere continuously throughout the year. Additional meteorological parameters used within the context of this study include air temperature (2 m a.g.l., KPK 1/6-ME-H38, Mela), precipitation (1 m a.g.l., heated

tipping bucket rain gauge, Thies), and long- and shortwave radiation components (5 m a.g.l., CNR4, Kipp & Zonen). For the results presented within Section 3 of this manuscript, all data were aggregated to daily timesteps.

     For more details regarding instrumentation setup, data processing and data quality assessment, please refer to Appendix A and Kittler et al. (2016; 2017), which also summarizes our approach to constrain eddy-covariance data uncertainty. Since we used exactly the same instrumentation at both drainage and control sites, and also the same data processing and quality

assessment approaches, our experimental setup rules out systematic differences in the observational datasets between both treatments that can be linked to methodological issues.

## 2.3 Soil monitoring and vegetation sampling

In each of the two treatment areas (drainage and control), we set up transects of ten sampling locations placed approximately 25 m apart, yielding two transects with a total length of 225 m each (Figure 1). At each location, we measured water table

depth in permanently installed perforated PVC pipes with 25 mm inner diameter. Thaw depth was assessed by pushing a metal pole into the ground. At every other site, profile probes for soil temperature with sensors at 0.05, 0.15, 0.25 and 0.35 m (Th3-s, UMS, Germany) and TDR soil moisture sensors at 0.075, 0.15 and 0.30 m below the soil surface (CS 640, 630 and 605, Campbell Scientific, USA) were permanently installed. For all these parameters measurements used within the context of this study were taken in conjunction with the operation of a flux chamber system during the period June 15 to August 20,

2014. In addition to the manual sampling, at four of the sampling locations, namely a wet and a dry microsite in each of the treatment areas, soil temperatures were continuously sampled at 0.04, 0.16 and 0.64 m since August 2014.

In 2014, we applied a non-destructive point-intercept method to sample vegetation community structure, with 20 sampling spots of 0.60 x 0.60 m within each treatment. Structuring each sampling spot into a subgrid of 6x6 cells with a side length of 0.10 m, we recorded the vegetation species that intercepted a laser pointer beam when pointing downwards from each grid intersection. The final coverage fraction for each vegetation species was approximated through the relative frequency of occurrence of classes found at these intersection points. More details are provided by Kwon et al. (2016).

## 2.4 Site climatology

Long-term temperature trends for the Chersky region were analyzed based on data from the period 1960-2009 provided by the Berkeley Earth project (berkeleyearth.org, station ID 169921). These Berkeley Earth surface temperature (BEST) time series underwent thorough data quality filtering to flag dubious data, or exclude data biased by instrument issues. Long-term precipitation records for the period of 1950-1999 were obtained from a local weather station through the NOAA online climate database (http://www.ncdc.noaa.gov/cdo-web/datasets, station ID 00025123). No precipitation records were available for the Chersky station from 2000-09. Thus, we analyzed long-term trends from the site Ambarchik station (ID 00025034) situated 100 km north of Chersky with nearly identical mean annual distributions of monthly precipitation sums.

## 3 Results

### 3.1 Climate and weather

### 3.1.1 Seasonality in temperature and precipitation

The averaged (1960-2009) seasonal air temperature course (Figure 2, left panel) has an amplitude of ~45 °C, bounded by a minimum monthly average of -33 °C in January and a maximum of 12 °C in July. The mean annual temperature within these five decades is -11 °C. The averaged annual precipitation sum for Chersky amounts to 197 mm, with mean monthly precipitation varying between 7 mm in March and 30 mm in August (Figure 2, right panel). The bulk of the annual precipitation input is received during the summer (JJA, ~39 %) and fall (SON, ~31 %) months.

Figure 2 summarizes the mean monthly air temperatures observed in the period 2013–15, which represent the datasets used in the subsequent sections. These recent observations follow the general amplitude of the long-term seasonal course. Individual outliers tend to be positive, but the annual mean temperature of -10.9 °C for the period 2014/15 is only slightly above the climatological mean. Since BEST data for Chersky are currently available until November 2013, general offsets between local site-level data and long-term records can only be approximated based on a 3-month overlap period. For these three months, we find a cold bias in the local data ranging between -0.53 and -1.25 °C, explaining the good agreement between recent data years and long-term trends despite the warming trends observed in the past decade (see also Figure 3). Precipitation data display a different pattern since April 2014 (start of measurements) compared to the long-term observations. Total precipitation in 2015 was ~71 % of the long-term trend (154 mm), with a much higher fraction provided

during summer (JJA: +40 %), while winter and springtime contributions are negligible in these years; however, since recent wintertime precipitation measurements are persistently low at very low temperatures, these data are likely to be subject to biases caused by insufficient heating of the sensor. Accordingly, focusing just on summertime conditions our records indicate an increase in precipitation in recent years, compared to the period 1950-1999 that was used for generating the climatology.

### 3.1.2 Long-term climate trends

BEST long-term temperature trends were analyzed broken up into four seasons of three months each: springtime warming (MAM), core summer (JJA), fall freeze-up (SON), and core winter periods (DJF); the setup of these periods intentionally deviates from the definition of seasons in the climatological or plant-physiological sense, instead focuses on comparing temperature trends in different parts of the seasonal course.

The results summarized in Figure 3 indicate heterogeneous trends over both decades and seasons. Using the 1960–69 temperatures as a reference, absolute values and year-to-year variability within all seasons did not change much until the end of the last century. In these first four decades of the analysis, the maximum deviation from the reference was found to be 1.01 °C (SON, 1990–1999), and only minor trends towards warmer conditions were observed for the shoulder seasons (MAM, SON) in the decades between 1970 and 2000. In contrast to this, mean temperatures in all seasons but winter display a step-change when transitioning into the 2000–2009 decade, with observed increases of 3.0 (MAM), 1.4 (JJA) and 2.8 (SON) °C between the 1990s and the 2000s. In spite of the abrupt nature of this transition to considerably warmer temperatures, decadal root mean square errors (RMSE) remain at the same level before and after the change. Wintertime temperatures, however, show no systematic trend over the analyzed five decades, but are subject to pronounced interannual variability in seasonal mean temperatures. The hiatus in warming trends since ~2000 corresponds to global trends for that time period (e.g. Trenberth and Fasullo, 2013), and has been linked to changes in atmospheric circulation and ocean currents (Trenberth et al., 2014).

Precipitation patterns were analyzed in the same way as described above for the temperatures (results not shown). In this case, no discernable trends were found, and decadal mean precipitation stayed within ranges of +/-15 mm (JJA) and +/-7 mm (all other seasons) in comparison to the 1950–59 references. Based on Ambarchik data, the 2000–2009 precipitation sums also stayed within the same ranges as observed for the five preceding decades, again without discernable trends.

### 3.2 Changes in hydrologic regime as response to drainage

Patterns in water table depth on this wet tussock tundra site are highly variable in both space and time. Accordingly, the effect of the drainage disturbance varies with the microrelief and changes over the course of summer. The top panels of Figure 4 demonstrate the spatial variability of terrain height and water table depth for a time where water levels had fallen to a seasonal low (here: July 14, 2014). The variability of the terrain height at fine scales leads to lateral water redistribution, with the result that one out of ten observation sites at the drained transect still reflects wet conditions, while two out of ten

sites in the control transect have water tables far below the surface level. Accordingly, the area affected by the drainage is not uniformly dry, but conditions have changed from predominantly wet to predominantly dry.

Temporal patterns of the drainage effect are also complex, as highlighted by the bottom panels of Figure 4. After the high water levels from the spring flooding have mostly receded towards the end of June, soil water conditions stay more or less constant across the control transect, while the drained transect displays a pronounced seasonality. For the latter, the water table drops by an average of 10 cm over the first three weeks of July 2014, resulting in dry top soil conditions for all sites but one. This drop is followed by a persistent recovery in water levels as a response to intensive rainfall (see also Figure 2) in late July and early August. Mean water levels within the drained area are always lower than those in the control area; however, as a consequence of the different sensitivity to precipitation inputs between drained and control transects, the net drainage effect results in water table differences between 4 cm and 15 cm (2014 observations), depending on weather conditions in the respective parts of the growing season.

### 3.3 Vegetation structure

In its natural state, the vegetation community of this wet tussock tundra ecosystem is dominated by common cotton grasses (*E. angustifolium*), with tussock forming sedges (*C. appendiculata* and *lugens*) as the second most important species (Corradi et al., 2005; Kwon et al., 2016). This is reflected in the coverage fractions observed during our 2014 plant species assessment (Table 1), which for the control area agrees well with a 2003 plant survey that was taken before the drainage disturbance was implemented (Corradi et al., 2005, not shown). Within the drained area, the decade-long shift of the water regime has strongly reduced the abundance of cotton grasses, while tussock forming sedges as well as shrub species (mostly birch, some willows) now cover larger fractions of the surface as compared to the control.

While the plot-based vegetation inventories indicate an increase in taller shrub species (*Betula exilis* and *Salix pulchra*, Table 1) from 0.6 % to 10.9 % areal coverage, we do not yet have a final quantitative assessment on the increase in shrubs within the footprint areas of the drainage tower. As an approximation, a land-cover classification based on 2011 WorldView-2 satellite remote sensing data (see also Figure 1) indicates an increase in tall shrub coverage by about a factor of four within a 400x400 m$^2$ area surrounding the drainage tower (from 1.2 to 4.8 %), compared to an area of the same size near the control tower. Also, a doubling of areas where tussocks and shrubs are mixed (from 34 to 67 %) was observed. Our datasets do not include a direct assessment of the average height of shrubs in drained and control areas; however, a comparative analysis between both towers (Figure 5) shows that drainage has led to a median increase of 0.026 m in summertime aerodynamic roughness lengths. Aerodynamic roughness length was derived here based on flux-profile relationships using friction velocity and wind speed at tower top under neutral atmospheric stratification. Increases on surface roughness were found to be omnidirectional, varying sector-wise between 0.015 m (SSE) and 0.091 m (NNE). Since both sites are exposed to the same wind conditions, this shift can only be caused by a higher abundance of tussocks as well as more and higher shrubs.

### 3.4 Snow cover, albedo and radiation budget

The changes in moisture regime and vegetation structure described in Sections 3.2 and 3.3 above have a profound impact on the local radiation budget, with the most pronounced effects attributed to shifts in snow cover. While the longwave radiation budget decreased only marginally as a consequence of drainage (-0.2 % during the summer months, data not shown), shortwave radiation is altered through changes in albedo, with distinct seasonal patterns in the differences between drainage and control areas that allow us to separate the 'light season' (mean daily net shortwave radiation >5 $Wm^{-2}$) into seven distinct periods (see Table 2 and Figure 6 for an analysis of the data year 2014).

In the snow free season, the shift in vegetation in the drainage area causes a minor but persistent increase in the albedo. This albedo change reduces the shortwave radiation budget by about 2 $Wm^{-2}$ throughout the summer since more energy is directly reflected upwards by lighter colored surfaces. This reduction in energy input agrees with Eugster et al. (2000), who observed increases in albedo linked to the gradual decrease in soil moisture over the course of the growing season. Major changes in albedo are observed during the disappearance and buildup of a closed snow cover in spring and fall. As shown in Figure 6, in both cases the process is initiated earlier in the drainage area, an effect that was observed in all data years since the beginning of the experiment in 2013. Since the thawing period in May falls together with high incoming shortwave radiation, even a few extra snow-free days lead to a strong increase in incoming energy (see also Table 2). Also the prolonged period where the snow cover starts to thin in spring leads to an energy gain for the drained terrain. Conversely, the earlier buildup of a closed snow cover in fall, as indicated by a rapidly increasing albedo, has little effect on the energy budget since incoming radiation at that time of the year is already very low.

Table 2 and Figure 6 demonstrate that a negative albedo difference between the drainage and control areas is only found during the snow melt period in spring, which makes up just about 13 % of the total time with sufficient incoming radiation for this kind of analysis (net shortwave radiation >5 $Wm^{-2}$). Since this short period falls together with high incoming radiation, the average impact of albedo changes on the energy budget is still positive for the drained area, with an average gain of 0.6 $Wm^{-2}$ over the total period of almost nine months. However, excluding this short period, the radiative energy input has been reduced as a consequence of drainage and subsequent albedo changes.

### 3.5 Soil thermal regime

The long-term changes in soil water conditions, vegetation coverage, radiation and snow cover listed above have increased both carbon content and bulk density (see Kwon et al., 2017, for details), and also trigger profound shifts on the soil thermal regime within this tundra ecosystem. These effects are summarized in Figure 7, where we compare continuous observations of soil temperatures down to 0.64 m below surface level between a dry and a wet microsite within the drained transect (similar relationships between moisture and soil temperature regime were observed within the control transect). We differentiate three major effects over the course of the year:

1. Drier conditions close to the surface at the dry microsite reduce the heat capacity of the organic soil. As a consequence, the upper few centimeters of the soil profile get warmer in the period June to August, with differences of up to 10 °C, compared to the wet microsite. At the same time, low thermal conductivity of dry organic soil prevents vertical heat transfer, so within the dry microsite deeper layers remain cooler for most of the period June through December.

2. Within the period November to January, the larger latent heat linked to the higher water content in organic soil within the wet microsite extends the zero curtain period, and delays the freezing for several weeks compared to the dry microsite. Accordingly, soils under dry conditions are much colder in early winter, with temperature differences that can reach up to 10 °C.

3. In the second half of February, the soil temperature decrease within the wet microsite finally catches up with the dry
site, and drops down to even colder temperatures. This effect can most likely be linked to the higher thermal conductivity in soils with ice-filled pore space, compared to drier soils with air-filled pores. Deeper snow cover associated with taller vegetation in the dry areas of this tundra site may contribute to this effect by better isolating those sites from the cold wintertime atmosphere. The resulting slightly warmer temperatures across the dry soil profile, which persist until June, were observed within both treatment areas in both years currently covered by the dataset.

Integrating soil temperature measurements of the dry and wet sites within the drainage section for two full data years 2014 and 2015, the average annual soil temperature decreased across the entire soil profile down to 64 cm below surface as a consequence of the drainage. The most pronounced decrease was observed at 16 cm (-0.98 °C), while both shallower (4 cm: -0.42 °C) and deeper (64 cm: -0.27 °C) showed a rather muted response to the disturbance.

To study the effects of moisture on the soil thermal regime in more detail, we analyzed time series of soil temperatures,
soil moisture and thaw depth at eight selected microsites. These sites were distributed across both treatments (drained: 5 sites; control: 3 sites), covering wet and dry conditions within both areas. The dataset was collected during two experiments in summer 2015, and was subsequently aggregated into four blocks of about two weeks each (Figure 8).

Close to the surface at 5 cm depth (Figure 8, left panel), wetter conditions lead to colder soil temperatures, confirming the differences between drained and control conditions during summertime already highlighted in Figure 7, related to the
lower heat capacity of dry organic soils. The temperature gradient between dry and wet microsites declines in late summer following the reduced radiative energy input (Figure 6). At 35 cm depth (Figure 8, center panel), opposite trends are observed. Here, higher soil moisture promotes higher soil temperatures, which can be linked to the fact that dry organic soil is a poor heat conductor. So even though this soil is warmer close to the surface, compared to wet microsites, the energy is not transferred into deeper soil layers, insulating the underlying permafrost from the heat close to the surface. Gradients
grow over the course of the growing season, resulting in temperature differences of almost 2 °C between dry and wet microsites at times.

Higher temperatures in deeper soil layers linked to wetter conditions clearly promote deeper thaw depths (Figure 8, right panel), with differences between dry and wet microsites increasing over the course of the growing season. In September, these differences can amount to more than 20 cm as a result of shifts in the soil water regime.

### 3.6 Sensible and latent heat fluxes

The changes in ecosystem properties following the drainage disturbance have profound impacts on the surface–atmosphere exchange fluxes of energy. In the following paragraphs, all numbers referring to growing season averages or differences represent results for the months June through September (see also Table 3). The winter results are presented after the summer results in the following section.

1. **Net radiation:** Due to a closed snow cover that usually lasts well into the month of May, most of the radiative energy is reflected until around early June (see also Figure 6), when a steep increase in net radiation ($R_{net}$) is observed at both sites (Figure **9**a). For the remaining growing season, the general trend in $R_{net}$ follows the solar zenith angle, with only minor impacts by seasonality in albedo. Systematic differences in $R_{net}$ between drainage and control site can only be observed during those few days in spring when the snow cover disappears earlier at the drained site (Figure **9**f); however, averaged over the growing season months June through September in 2014 and 2015, which largely excludes the snow melt transition, the energy input is reduced by about -1.7 $Wm^{-2}$ (-1.8 %) caused by the higher albedo in the drainage area (see also Table 3).

2. **Sensible heat flux:** Exchange of energy between surface and atmosphere in form of sensible heat (H) displays a sawtooth pattern over the growing season correlated to the net radiation trends, with a steep incline in June followed by a gradual decrease that lasts through fall (Figure **9**b). The timing between the increase in net radiation and H is not aligned, with the onset of high sensible heat fluxes in spring trailing the steep increase in $R_{net}$ by 1-2 weeks. This decrease is interrupted in 2015 by a plateau of H fluctuating around a nearly constant mean within July and August, which is then followed by a steep decline in September. Differences in H between drainage and control area vary strongly between data years, with large differences in 2014 observed only in early summer, while in 2015 the entire growing season displays a pronounced offset (Figure **9**g). Overall, sensible heat fluxes have systematically increased as a consequence of the drainage, with peak differences reaching up to 50 $Wm^{-2}$ for single days in early summer, and an average growing season increase of 2.4 $Wm^{-2}$ per day (+9.4 %, Table 3).

3. **Latent heat flux:** In contrast to the patterns observed for the sensible heat flux, latent heat fluxes (LE) start displaying higher values very shortly after the snow melt, but subsequently increase slowly until reaching a maximum in July (Figure **9**c). For LE, the signal fluctuates strongly over the largest part of the growing season, with longer term averages stable for the months of June and July. Absolute magnitudes in LE differ strongly between data years in late July and August, coinciding with heavy precipitation in 2014 that increased the fluxes that year. We also observed a pronounced interannual variability in differences between drainage and control areas (Figure **9**h), which can be linked to variable weather conditions in different parts of the growing season (see discussion below). In 2014, latent heat fluxes in the drained area were lower compared to the control area by an average of -6.3 $Wm^{-2}$ per day, which can mostly be attributed to reductions in LE in the months of June and July within the drainage. Conversely, during the growing season of 2015 the latent heat fluxes were higher within the drained section (average: 1.9 $Wm^{-2}$ per day), except for a

brief period immediately following the spring flooding. Averaged over both data years, this leads to a net decrease of LE within the summer months of -2.2 $Wm^{-2}$ (-7.3 %,Table 3) within the drainage. However, due to the pronounced interannual variability this mean value may not be representative over longer time periods, and more data years would be required to constrain a net drainage impact.

5   4. **Sum of surface-atmosphere energy exchange:** Summing up shifts in H and LE indicates that the total vertical energy flux from the surface to the atmosphere does not change systematically following the drainage (Figure **9**i). Mean values and temporal patterns, however, differ strongly between years. In 2014, where increases in H are outweighed by much lower LE fluxes for large parts of the summer months, we observed a net decrease in total energy fluxes (-4.2 $Wm^{-2}$ daily average). Conversely, in 2015 the positive offsets in H and LE combine to a continuous positive difference in overall energy exchange between treatments, summing up to a daily mean of 4.5 $Wm^{-2}$ over the four growing season months. In both data years, the largest part of the net offsets was accumulated in the later part of the growing season, indicating that temperature and moisture conditions in July and August dominate the interannual variability. Since we found opposing trends with nearly equal magnitude in both data years, averaged over both years the drainage exerts only a minor increase in summertime energy fluxes of 0.2 $Wm^{-2}$ per day (+0.3 %, Table 3).

15   5. **Bowen-Ratio:** The Bowen-Ratio (BR), which is the ratio of sensible to latent heat fluxes, summarizes how radiative energy is partitioned between the two vertical energy fluxes H and LE. The higher the BR values, the more the available energy is shifted towards the sensible heat flux, which may indicate water limitations and/or drought stress. At our observation sites near Chersky, absolute daily mean values of BR are usually highest during the early summer, then decrease until August and rise again towards the beginning of fall (Figure **9**e). Moving window averages of offsets between drainage and control show a pronounced peak in June, and are positive throughout the summer months (Figure **9**j), implying that the share of energy transferred into sensible heat is systematically higher within the drainage area. Mean summertime Bowen Ratios, calculated based on averaged values for H and LE demonstrate that drainage increases daily mean BR by a value of 0.15 (+18.0 %, Table 3).

**Wintertime fluxes** (here: November 2014 – March 2015) do not contribute notably to net changes in the energy budget at our observation sites near Chersky. Shortwave energy input is extremely low during the polar winter, and outgoing longwave radiation usually exceeds incoming radiation. Accordingly, on average we observed negative net radiation budgets (drainage: -12.3 $Wm^{-2}$; control: -10.5 $Wm^{-2}$) within both treatments, and values for $R_{net}$ were slightly lower (-1.6 $Wm^{-2}$) in the drainage area. Latent heat fluxes were close to zero for this entire period (drainage: 0.0 ±0.13 $Wm^{-2}$; control: 0.2 ±0.02 $Wm^{-2}$), and average values for the sensible heat flux are in the negative range (drainage: -5.6 ±0.20 $Wm^{-2}$; control: -4.4 ±0.11 $Wm^{-2}$). Combined flux shifts in H and LE of -1.4 $Wm^{-2}$ therefore closely balance the changes in $R_{net}$, so that during wintertime the aboveground energy balance ($R_{net}$ – H – LE) is barely affected by the drainage disturbance.

## 4 Discussion

### 4.1 Controls on Arctic energy balance

Previous studies have identified vegetation community, atmospheric conditions, and the underlying permafrost as major controlling factors on the biogeochemical cycles of Arctic wetland ecosystems (e.g. Rouse, 2000). Focusing on the energy
budget, Eugster et al. (2000) identified the following properties as dominant features that characterize the specific feedback processes between the Arctic energy balance and climate change: short growing season with long summer days, permafrost and the existence of massive ground ice, prevalent wetlands and shallow lakes, and a nonvascular ground vegetation. All of these factors are closely linked to each other (e.g. McGuire et al., 2002), and thus the assessment of the net effect of climate change on these ecosystems needs to synthesize inter-related responses to capture the cascade of positive and negative
feedback loops that affect geophysical, hydrological, and biological ecosystem characteristics (McFadden et al., 1998; Hinzman et al., 2005). This is supported by the findings from our experiment, where the impact of soil hydrology was identified as a major control, triggering secondary shifts in other factors and ultimately reshaping the ecosystem.

### 4.1.1 Vegetation impact

Many studies that examined the energy budget across ecosystem boundaries in the Arctic have identified the terrain type as
the dominant control on energy flux rates as well as on the partitioning of energy between sensible, latent and soil heat fluxes (Eugster et al., 2000; Eaton et al., 2001; Beringer et al., 2005; Kasurinen et al., 2014). For example, Beringer et al. (2005) studied a vegetation transect in Alaska, and found increasing leaf area index and decreasing albedo along the transition from Arctic tundra to boreal forest ecosystems, which decreased evaporation while increasing transpiration and sensible heat fluxes. Their results demonstrate that successional changes that convert low tundra vegetation into shrub tundra
or woodlands have the potential to intensify vertical energy exchange with the atmosphere, and thus create a positive feedback to warmer conditions under future climate change.

The results presented in this study confirm that vegetation is a key controlling factor within the cascade of secondary shifts in ecosystem characteristics following the primary drainage disturbance. The higher abundance of shrubs, promoted by drier and warmer conditions in shallow soil layers within the drainage area, exerts the most obvious effect on the snow cover
regime for a couple of reasons. The first is that shrubs and tussocks capture the horizontally drifting snow more effectively than cotton grass meadows in the control tundra sections, leading to an earlier accumulation of a closed snow cover in fall (e.g. Sturm et al., 2001a; 2005b) that contributes to warmer soil conditions in the winter season. The second reason is that shrubs sticking out of the otherwise closed snow cover absorb and re-emit the high incoming radiation in April/May, thus contributing to the earlier snow melt observed within the drainage area (Sturm et al., 2005a; 2005b; Pomeroy et al., 2006).
During the summer, it is likely that ground shading by shrub canopies counteracts the observed warming in shallow soil layers (e.g. McFadden et al., 1998); however, based on our dataset, this effect could not be separated from the soil moisture impact on soil temperatures.

Both plot surveys and gridded maps do not include information on vegetation height. Still, the available ground-based and remote sensing information on shifts in vegetation community structure provides strong evidence that drainage induces shifts towards taller vegetation that play a major role in the overall transformation of the ecosystem structure following the drainage disturbance. During summertime, our eddy-covariance fluxes indicate an omnidirectional increase in aerodynamic roughness length (Figure 5) and mechanically generated turbulence (as represented by the friction velocity, data not shown) at the drainage tower, compared to the control tower, indicating that higher vegetation surrounding that tower intensifies turbulent exchange processes between surface and atmosphere.

The virtual absence of mosses at our observation sites (~1.8% in both transects, data not shown) is also expected to influence energy flux rates and soil thermal regime. Moss cover usually dominates the surface in high latitudes, and has been shown to play a key role in modifying temperature and moisture conditions in Arctic soils (Beringer et al., 2001; Zona et al., 2011b; Kim et al., 2014). Evaporation from mosses contributes a significant portion of the latent heat flux in Arctic ecosystems (McFadden et al., 2003; Beringer et al., 2005). Moss insulation reduces the soil heat flux and increases energy exchange with the atmosphere (Beringer et al., 2001), similar to the insulation effect of dry organic soil that was observed in the present study. It can therefore be assumed that drainage could further reinforce the moss-cover effect on energy flux partitioning and soil thermal regime, as long as the drained soil can still sustain the moss layer.

### 4.1.2 Atmospheric impact

Numerous studies identified atmospheric forcing as the most important control on the energy budget of individual high latitude ecosystems in the absence of vegetation shifts (e.g. Rouse et al., 1992; Harazono et al., 1998; Kodama et al., 2007; Boike et al., 2008; Langer et al., 2011a). Precipitation anomalies were found to be a major determinant for interannual variability in energy partitioning (Boike et al., 2008), and Rouse et al. (1992) listed variations in the total amount of summertime precipitation as a control for active layer depth development. Synoptic patterns exerted a strong influence on energy fluxes in both Alaskan (Harazono et al., 1998) and Siberian (Kodama et al., 2007; Boike et al., 2008; Langer et al., 2011a) study domains. Onshore winds coming from the ocean, characterized by cold and wet conditions, promote sensible heat fluxes since they increase the temperature gradient between surface and atmosphere. Warm and dry offshore winds, however, usually decrease the sensible heat flux. Also, the impact of cloud cover on the longwave radiation budget had an important influence on soil temperatures and freeze-back patterns in fall (Langer et al., 2011a).

Short-term shifts in weather conditions, such as a warm bias in August 2014 (see Figure 2), exerted only a slight influence on the seasonal trajectories of the sensible heat flux. In contrast, an average long-term reduction in net radiation of about 2 % during the period June through September 2015, compared to 2014, decreased the H budget by about 9 %. No systematic differences in the interannual variability of H were observed between drained and control areas, wherein the overall reductions in the flux rates were of equal magnitude, and both areas reacted uniformly to changes in atmospheric forcings. Regarding the longwave radiation, even though this parameter barely changed as a consequence of drainage the

interannual variability of the upward directed component correlated well with the sensible heat fluxes, both in terms of absolute values and differences between treatments.

Regarding the latent heat flux, similar to Boike et al. (2008) we found an influence of precipitation patterns on interannual variability in LE, particularly in late summer and within the drainage section of our study area. For example, heavy rainfall events such as those observed in late July and early August 2014 can lead to partially waterlogged conditions within the drainage ring (Figure 4), and increased evapotranspiration rates at these times often match those in the control area. Moreover, in contrast to the sensible heat flux, we found evapotranspiration rates highly susceptible to day-to-day variability in net radiative energy input. Regarding interannual variability, we observed opposing trends in summertime LE within drained and control areas, wherein compared to 2014, in 2015 latent heat flux rates increased by an average of ~5 % in the drained areas, while the control fluxes decreased by about 20 %.

Comparing weather conditions and energy fluxes between 2014 and 2015, the reduction in net radiation input in 2015 led to a pronounced reduction in vertical energy transfer to the atmosphere in the control section (about -16 %, combining H and LE), while the net change in turbulent fluxes in the drained area was much lower (-2 %). We speculate that the deviating interannual variability between the sections may be driven by differences in soil moisture levels between data years, e.g. linked to the timing of precipitation events, which influenced the feedback of LE flux rates to variability in net radiation. These findings suggest that the timing of shifts in weather conditions played an important role for latent heat fluxes within the drainage section, and dominates interannual variability. In particular, the slightly warmer conditions in June and July of 2015, in combination with the frequent occurrence of light to moderate rainfall events, resulted in higher latent heat fluxes in this period, compared to 2014.

Synoptic influences were not analyzed in detail in the context of this study. Our focus was placed on the differences in ecosystem structure and surface-atmosphere energy exchange between a drained and a control observation site. Since both sites were placed only approximately 600 m apart, it can be assumed that both are exposed to the same atmospheric forcing. A direct comparison of weather conditions measured at both towers, including air temperatures, humidity, pressure and precipitation, resulted in no systematic offsets going beyond the calibration accuracy of the employed instrumentation.

**4.1.3 Impact of permafrost**

A primary characteristic of permafrost landscapes is that the ice-rich frozen layer inhibits vertical water losses, preserving water-logged conditions during large parts of the summer, and thus facilitating the establishment of wetlands even in areas with relatively low precipitation input (Rouse, 2000). Within permafrost landscapes, a large portion of the net energy input from radiation is used to thaw the frozen ground, and increase the thaw depth over the course of the growing season (e.g. Lund et al., 2014). Permafrost constitutes a substantial heat sink, reducing the soil temperatures and also the energy available to feed turbulent heat flux exchange with the atmosphere (Eugster et al., 2000; Langer et al., 2011a). Accordingly, permafrost acts as an efficient buffer against the intensification of energy exchange that might be triggered by a warmer future climate in high latitude regions, but this controlling mechanism would be reduced in case the thaw depth increases

(Lund et al., 2014). Another particular feature of northern permafrost soils is that the very cold frozen ground generates a downward directed heat flux from the snow in spring. This increases the amount of energy required to melt the snowpack, and accordingly delays melt (Eugster et al., 2000). The warmer soil conditions found within the drainage area in late winter, which are likely caused by a higher thermal conductivity in ice-rich soils, supported by a better ground insulation through increased snow depth captured by higher shrubs, would reduce the downward heat flux, and therefore may contribute to an earlier snowmelt.

For the reasons listed above, wet permafrost ecosystems are often characterized by a large fraction of the soil heat flux in the energy budget (Boike et al., 2003; Langer et al., 2011a) that can be of the same order of magnitude as latent and sensible heat fluxes (Eugster et al., 2000). The partitioning of net radiation in permafrost landscapes is particularly sensitive towards the moderation of bulk surface resistance by the vegetation, and the thermal conductivity of shallow soil layers (Harding et al., 2002; Liljedahl et al., 2011). In the current study, soil heat fluxes were not measured directly, so their magnitude can only be approximated based on the difference of energy supply from net radiation and energy demand for latent and sensible heat fluxes. This residual amounted to 39–46 % depending on data year and site, or 14–31 % assuming that 15-25% of the net radiation is attributed to an unclosed energy balance (Foken et al., 2011) or additional energy sinks such as heat storage in water (Harazono et al., 1998). While residuals in 2014 and 2015 were similar in the drainage area, values increased by about 8 % in the control section, mostly driven by shifts in LE.

### 4.1.4 Impact of soil hydrology

The moisture and thermal regulation of Arctic wetland ecosystems is strongly influenced by the presence of an organic soil layer, which features an extremely large water content when wet, and equally large air content when dry (Rouse, 2000). Water saturation conditions affect heat flux rates into soils, and also alter thaw depth and the net ecosystem-atmosphere heat exchange (Jorgenson et al., 2010; Subin et al., 2013). Moreover, waterlogged conditions can increase solar absorption, contributing to increased thaw depths and potentially to a subsequent lowering of the water table with respect to the surface (Olivas et al., 2010). Conversely, lowering the water table can also preserve moisture in the ecosystem in the absence of plants with high leaf area index and deep roots, since higher albedo reduces net radiation, and poor thermal conductivity in dry soils keeps deeper layers colder (Eugster et al., 2000).

We found both direct and indirect effects of shifts in soil hydrology on the energy budget of our study site, with the latter likely to have a stronger impact on the long-term trajectory of this permafrost ecosystem than the former. The combined impact of the most prominent indirect effects, the shifts in vegetation community structure and snow cover regime, will be discussed in more detail in the following section. Direct effects comprise the decrease in both heat capacity as well as thermal conductivity when drying out organic soils, and our results show indications of reduced soil heat fluxes across the seasons. The observed patterns between soil water content and progress of thaw depth over the course of the growing season agree well with findings previously reported for Arctic ecosystems in Alaska (Hinzman et al., 1991; Shiklomanov et al., 2010; Sturtevant et al., 2012). A second direct effect of drier conditions in shallow soil layers is a shift in

energy partitioning towards higher Bowen Ratios, with a larger portion attributed to sensible heat fluxes. Also, after drainage particularly the latent heat fluxes become more dependent on short-term atmospheric forcing, with strong variability observed in LE rates related to precipitation input.

Regarding the net effect of dry soils on soil heat fluxes and thermal regimes, different pathways are possible, depending largely on soil type and the severity of the disturbance. Rouse et al. (1992) observed that higher temperatures and associated moderate increases in evapotranspiration moved water tables beneath the surface, but the peat soils at their study site remained wet. As a result, thermal conductivity decreased less rapidly than heat capacity while the thermal diffusivity was enhanced, which led to deeper thaw depth with warm and dry climate conditions. In contrast, Bonan et al. (1990) postulated that the greater evapotranspiration accompanying climate warming would dry out the surface organic layer, causing a reduction of soil heat fluxes and thus stabilize the existence of permafrost. Our observation of cooler summertime temperatures in deeper soil layers following drainage, and a substantial reduction of thaw depth in the drained section, support the statement by Bonan et al. (1990); however, including secondary drainage effects, where the combination of higher vegetation and higher snow pack leads to a warmer soils during winter, over longer timeframes drainage may also lead to a gradual warming of soil temperatures, and therefore contribute to permafrost degradation.

## 4.2 Implications for feedback processes with climate change

### 4.2.1 Interrelated ecosystem responses

Sophisticated numerical models are needed for assessing the complex feedback processes between permafrost ecosystems and climate change, but is unclear yet which processes need to be explicitly resolved in these models, and which input parameters need to be provided at what resolution, to avoid systematic biases in simulation results (Eugster et al., 2000). Our findings indicate that process-based models representing permafrost ecosystems to a high degree of detail would be required for this objective. We found direct effects of the primary drainage disturbance on the energy budget of our study site, but also secondary effects based on shifts in other ecosystem properties following the lowered water tables. Also interactions between secondary effects were observed, including feedback processes to the primary drainage. Neglecting this network of positive and negative feedback loops on permafrost ecosystem energy budgets will likely lead to biased net effects, and therefore distort both the simulations regarding the sustainability of individual high latitude ecosystems under climate change, as well as their relevance at regional to global scales. In fact, at the time when our observations took place, a clean separation between e.g. primary drainage effect and the changes in vegetation cover that followed it could not be made without additional manipulation to the ecosystem.

The specific hydrologic conditions created by permafrost ecosystems, characterized by a barrier to infiltration posed by the frozen ground, provided the prerequisite to form the original wet tussock tundra ecosystem. Regarding the primary disturbance (the lowering of the water table), impacts on the ecosystem were initially restricted to shifts in the partitioning of energy between sensible, latent and soil heat fluxes during summertime. As a direct secondary effect, a taller vegetation

community with higher aerodynamic roughness was established, causing higher mechanically generated turbulence and also a slightly higher albedo in summer. The vegetation shift then triggered substantial changes in the snow cover regime, with higher albedo and warmer soils in late winter and spring, and an energy pulse in early summer related to the expedited snow melt. Every single effect will influence the net annual energy budget of this permafrost ecosystem, and accordingly they can only be analyzed as a network of closely linked properties when assessing the net impact of drainage disturbance.

One particularly interesting network effect of combined shifts in ecosystem properties is the interaction between drainage and vegetation shifts. The increased abundance of shrubs in high latitude regions has been studied extensively (e.g. Sturm et al., 2001b; Myers-Smith et al., 2011; DeMarco et al., 2014a), and numerous studies have identified links with factors such as snow cover (Sturm et al., 2001a; Pomeroy et al., 2006), radiation regime (Bewley et al., 2007) and nutrient cycling (Myers-Smith and Hik, 2013; DeMarco et al., 2014b). Regarding soil temperatures, the isolated assessment of shrub expansion indicates that the capture of drifting snow by shrubs increases snow depth and soil temperatures in winter, while shading decreases temperatures in summer (e.g. Sturm et al., 2001a). Assessing the net impact of these opposing effects depends on details like shrub density, which complicates the evaluation of this effect at landscape to pan-Arctic level. Further, including a drainage effect increases the complexity. For example, during winter the shifts in snow cover caused by the taller vegetation still persist, but the warming effect is modulated by the low heat capacity and low thermal conductivity of dry organic soil, which decreases soil temperatures earlier in fall, but causes higher soil temperatures towards the end of winter. During summer, in shallow layers the warming triggered by the low heat capacity of dry organic soil dominates over the shading effect of the shrub canopy, while in deeper layers the shrub cooling effect is reinforced by low thermal conductivity. Accordingly, compared to the isolated assessment of shrub expansion, a drained permafrost ecosystem with a higher abundance of shrubs may experience a higher net warming in winter, but shows colder temperatures in deeper layers in summer.

The severity of the drainage needs to be considered when interpreting the outcome of hydrologic disturbance effects. As summarized in the introduction, many studies have treated the impact of a lowered water table on biogeochemical cycles in permafrost ecosystems. However, for most of them the water table was lowered less than 10 cm, so the differences before and after disturbance were not as pronounced as to be expected after the formation of a trough system following the degradation of ice-rich permafrost. According to Rouse et al. (1992), moist organic soils can still act as good thermal conductors, so reduction of thaw depth, as observed in our experiment, can only be found after a dramatic shift in soil water tables that largely dry out the shallow organic layers.

### 4.2.2 Consideration of long-term effects

As pointed out in the previous section, a combination of primary and secondary disturbance effects, and their interactions, generates the net impact of the drainage disturbance on the energy budget. In this context, temporal aspects need to be considered as well when interpreting the results. While the primary disturbance immediately affected the water table, other key components like the vegetation community structure take longer to change. Consequently, very different net effects were

found in the year immediately following the disturbance (Merbold et al., 2009), compared to our more recent results from about 10 years later (Kittler et al., 2016). At the same time, it can be assumed that the system is even now not fully equilibrated towards the new conditions, and further shifts in factors like shrub coverage and canopy height can be expected for the long-term trajectory of this site. Still, we expect that the major elements of change are already established by now, and no major shifts in the overall functionality of the energy budget feedback processes are expected for the future.

Another factor with a long-term trajectory regarding the response towards drainage disturbance is temperature trends in deeper soil layers. Throughout the studied section of the soil profile, we observed alternating periods dominated by warming or cooling over the course of the year, with all layers down to 64 cm below surface experiencing a net cooling following the drainage. Accordingly, particularly in deeper soil layers we see a complex combined impact of a reduced cooling down of soils in winter, probably linked to the differences in thermal conductivity between ice-filled and air-filled pores, and shifts in the insulation by dry organic soils, preventing vertical heat transfer in summer. Whether or not the wintertime warming will eventually be substantial enough to dominate over the summertime cooling and impact thaw depth cannot be evaluated with the currently available database. An assessment of the long-term temperature trends would either require a suitable process-based modeling framework, or longer-term temperature observations, ideally including deeper boreholes, and thus is beyond the objectives of this study.

### 4.2.3 Implications for Arctic climate change

An increase in sensible heat flux rates as a response to shifts in the partitioning of the available net radiation is the most direct pathway to change the temperature of the atmospheric boundary layer (Eugster et al., 2000; Lund et al., 2014). This statement is supported by Chapin et al. (2000), who expect a high potential for positive feedbacks between land-atmosphere energy exchange and regional temperature changes throughout high-latitude regions, disregarding potential shifts in land cover structure. Eugster et al. (2000) therefore count the unknown magnitudes of changes in energy partitioning as well as the lack of long-term energy balance data from Siberia among the most important uncertainties for assessment of susceptibility and vulnerability of Arctic ecosystems to climate change.

Our experiment triggered an artificial lateral redistribution of water, converting a formerly uniform wet tussock tundra ecosystem into severely drained terrestrial areas intersected by a drainage channel. This approach mimics the preferential degradation of ice wedges in ice-rich permafrost under climate change, which can lead to the formation of a system of connected troughs (Serreze et al., 2000; Liljedahl et al., 2016). Our results can therefore provide observational evidence of energy budget shifts that can be expected in regions susceptible for this type of degradation. Ice-rich permafrost (Yedoma) covers large parts of the North American and Siberian Arctic (e.g. Strauss et al., 2013), and in the Siberian plains has been found to contain very high ice contents that made up 40 to 70 % of the soil volume (Zimov et al., 1997). It can be expected that geomorphological evolution and hydrological responses to permafrost degradation as a result of longer-term effects of a warming Arctic climate (e.g. Hinzman et al., 2005) will constitute a large-scale phenomenon with potential pan-Arctic and even global feedback implications on climate change.

Regional changes in energy fluxes will probably interact with the carbon cycle by changing the disturbance regime, regional temperature and precipitation, and the depth of the boundary layer (Chapin et al., 2000). Moreover, shifts in energy budgets may also alter the delicate patterns of regional scale energy transfer within the Arctic and beyond. The redistribution of energy from certain regions in the Arctic and boreal zone to northern areas which is observed under current conditions (for example the heat flows from Alaska in both northerly and easterly directions) may increase under a warming climate whenever the energy transfer from surface to atmosphere increases (Eugster et al., 2000). Accordingly, the effects of shifts in energy exchange patterns following the degradation of ice-rich permafrost will not be restricted to these areas, but are likely to exert effects of substantial magnitude at the pan-Arctic scale.

**5 Conclusion**

Degradation of ice-rich permafrost under future climate change holds the potential to transform geomorphological and hydrological characteristics within large parts of the Arctic. Persistently drier or wetter conditions may alter ecosystem structure dramatically, and lead to systematic shifts in biogeophysical and biogeochemical processes. Both the effects of long-term equilibration and non-linear feedback processes between shifts in ecosystem components complicate the assessment of the net impact of this type of disturbance, therefore the long-term trajectory of ice-rich permafrost ecosystems in the Arctic is highly uncertain to date.

We presented observational evidence on the potential long-term consequences of sustained drainage in a previously wet tussock tundra ecosystem in Northeast Siberia. Our datasets indicate that a decade-long lowering of the water table triggered a cascade of secondary changes at our observation sites, including shifts towards taller vegetation, modification of the snow-cover period, and profound shifts in soil temperatures. Warmer conditions were found throughout the soil profile (i.e. down to 64 cm below surface level) towards the end of winter; during the summer, however, the low heat capacity of dry organic soils most likely led to warmer conditions in shallow layers, while at the same time poor thermal conductivity kept deeper layers colder. Since dry organic soils are efficient insulators, drainage can reduce thaw depth, and can initially protect deep permafrost carbon pools from degradation under warmer climate conditions. However, the net impact of wintertime warming and summertime cooling on long-term permafrost temperature trends following drainage still remains to be assessed.

With heat transfer into the soil systematically reduced in summer, the primary drainage disturbance in combination with related secondary changes led to a minor increase in energy transfer back to the atmosphere, even though higher albedo following vegetation shifts slightly lowered the energy input through net radiation. Mean sensible heat flux rates increased in the drainage area (+2.4 Wm$^{-2}$, or +9.4 %), closely balanced by a parallel decrease in averaged latent heat fluxes (-2.2 Wm$^{-2}$, or -7.3 %). This shift in energy flux partitioning, reflected by a Bowen Ratio increase of ~18 %, may lead to secondary feedback effects in temperature and moisture regimes of the lower atmosphere, and can thus aggravate climate change impacts once the degradation of ice-rich permafrost has initiated hydrologic redistribution. If our local scale results can be

confirmed for different Arctic regions, and different hydrologic disturbance scenarios, the demonstrated effects could therefore be relevant for forecasts of the sustainability of Arctic permafrost ecosystems under future climate change.

**Acknowledgements**

This work was supported through funding by the European Commission (PAGE21 project, FP7-ENV-2011, Grant
Agreement No. 282700, and PerCCOM project, FP7-PEOPLE-2012-CIG, Grant Agreement No. PCIG12-GA-201-333796), the German Ministry of Education and Research (CarboPerm-Project, BMBF Grant No. 03G0836G), and the AXA Research Fund (PDOC_2012_W2 campaign, ARF fellowship M. Göckede). The authors appreciate the contribution of staff members of the Northeast Scientific Station in Chersky for facilitating the field experiments, especially Galina Zimova and Nastya Zimova. We would also like to thank the administration and service departments within the Max-Planck-Institute for
Biogeochemistry, most notably the Field Experiments & Instrumentation group, for their contributions to planning and logistics, and for supporting field work activities.

We applied sequence-determines-credit (authors 1-3) and equal-contribution (alphabetical sequence, all other authors) methods for the order of authors.

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

**Appendix A: Eddy covariance instrumental setup and data processing**

The eddy-covariance observations at our field site near Chersky comprised a drainage tower (68.61 °N and 161.34 °E, instrument height 4.91m a.g.l.) and a control tower (68.62 °N and 161.35 °E, 5.11m a.g.l.) representing undisturbed natural conditions (see also Figure 1). Both towers were placed at an elevation of approximately 6 m a.s.l., with an average vegetation height of 0.7 m during summer. Each eddy tower featured a heated sonic anemometer (uSonic-3 Scientific, METEK GmbH) to monitor three-dimensional wind fields and sonic temperatures, and a closed-path gas analyzer (FGGA, Los Gatos Research Inc.) to capture mixing ratios of $CH_4$, $CO_2$ and $H_2O$. A heated and insulated sampling line (16.0 m and 12.8 m length for drainage and control towers, respectively) connected the inlets close to the sonic anemometer with the gas analyzers. An external vacuum pump (N940, KNF) facilitated a nominal flow rate of 13 L min$^{-1}$, equivalent to a replacement rate of ~ 2–2.5 Hz in the measurement cell.

Eddy-covariance data was recorded at 20 Hz. We collected primary data using the software package EDDYMEAS (Kolle and Rebmann, 2007), and based all further processing of the high frequency data on the TK3 software (Mauder and Foken, 2015). The flux processing sequence included 2D coordinate rotation of the wind field, cross-wind correction (Liu et al., 2001), spectral correction (Moore, 1986), and an online conversion of gas analyzer output to dry mole fractions before processing by the TK3 software. We furthermore applied cut-off frequencies of 2 Hz for the $H_2O$ data to correct losses linked to tubing effects and limited gas replacement rates in the closed-path analyzers.

At the core of our post-processing quality control scheme, we follow the flux data quality flagging scheme presented by Foken (2008), which is an update of the method proposed by Foken and Wichura (1996). Their original tests on stationarity and well-developed turbulence were amended for this study by flags also reflecting overall errors in the log file recorded by the sonic anemometer, accordance of instrument operational limits with prevailing boundary conditions, and sonic anemometer heating status. Gapfilling and flux partitioning were both based on the method of Reichstein et al. (2005), using separate gapfilling and flux partitioning data pools for different seasons.

The assessment of eddy-covariance flux data uncertainty was based on well-established concepts (e.g. Aubinet et al., 2012) that assess the total error as the combination of random and systematic errors. Systematic errors are composed of errors associated with unmet theoretical assumptions and methodological challenges of the eddy-covariance technique, data processing errors, and instrumental calibration issues (Mauder et al., 2013). Since within this study, we were primarily focusing on flux differences between two equally equipped eddy-covariance systems set up close to each other, the first two items in this list could be neglected, based on the fact that a uniform data processing has been applied that considers all recommended quality filters and correction procedures. Regarding instrument calibration, we did not find a systematic offset in the frequency distributions of wind speed, sonic temperature and water vapor mixing ratios between both towers (data not shown), indicating that the calibration of the uniform instrumentation did not introduce a systematic bias into the data as well. Systematic shifts in computed flux rates can also be caused by the setup of the quality flagging system that determines which part of the data to exclude from further analysis. The chosen scheme will influence the frequency and seasonal

distribution of data gaps, and therefore also the performance of the gap-filling procedure. Since our approach aimed at minimizing gaps, while at the same time skilfully reviewing particularly low-quality fluxes through additional quality measures, we can rule out potential systematic biases linked to this methodology as well. Summarizing, systematic errors did not play a role for the uncertainty assessment of the dataset presented herein, and were therefore excluded from the analysis.

Random errors in time series of eddy-covariance fluxes are mostly linked to turbulence sampling errors, instrument errors, and uncertainties of the footprint (Rannik et al., 2016). We neglected the footprint errors in this context, since both towers are exposed to the same wind climatology. Assessment of the turbulent sampling error and instrument errors for each 30-minute flux value are a standard output of the flux processing software employed here (TK3, Mauder and Foken, 2015). These values were treated as independent errors when combined to a total error. Uncertainty assessments for the gap-filled

flux values were taken from the output of the MDS-routine (Reichstein et al., 2005) employed for this purpose. The random errors of the ensemble averaged fluxes were computed following Rannik et al. (2016)

      Regarding the accuracy of radiation measurements, the Kipp & Zonen CNR4 radiation sensors employed here at both observation sites are officially classified as 'first class' instruments (for shortwave radiation e.g. a resolution of +/-5 $Wm^{-2}$, and a stability of +/-2%). However, based on the direct comparison of data from both instruments, we find cumulative

differences below 1% of the total incoming radiation, therefore our sensors would even qualify for the next highest quality rating (secondary standard, e.g. a resolution of +/-1 $Wm^{-2}$, and a stability of +/-1%). The CNR4 instruments have downward looking opening angles of 150 degrees. For sensor heights of 4.5m (drainage tower) and 4.66m (control tower), this translates into circular footprints with a radius of 16.8 m and 17.4 m, or footprint areas of 886 $m^2$ and 950 $m^2$.

      Meteorological data from all other (slow) instruments were collected at 10-second intervals and stored as 10-minute

averages. The final dataset was averaged to 30-minute intervals. A separate quality assessment protocol was developed for this part of the dataset, comprising e.g. a test for failure of the power supply, checks of range and variability of time series, a flat lining test, a spike test, and finally a test for sensors malfunctioning based on plausibility limits.

      All datasets presented within the context of this study were based on the same instrumentation, as well as a uniform data processing and quality assessment routine, for both drainage and control sites. Manual measurements such as vegetation

community studies, or measurement of active layer depth and water table depth, were carried out by the same persons in both study areas within each specific observation period, so that also here site intercomparison results cannot be subject to systematic errors based on subjective decisions by the observer. All datasets, including eddy-covariance fluxes, slow meteorology and manual sampling, have undergone a thorough quality assessment protocol (Kittler et al., 2016; Kwon et al., 2016), which included plausibility checks based on the intercomparison of observations between both study areas. Still, it

cannot be ruled out that parts of the absolute values presented here are subject to systematic offsets; however, since both drainage and control datasets should be affected by such systematic offsets in the same way, the differences between these datasets will not be affected, and can thus be fully attributed to drainage effects on ecosystem characteristics and energy flux rates. A comprehensive description on instrumentation and data processing procedure can be found in Kittler et al. (2016).

**Table 1: Relative abundance of the major vascular plant species found within both disturbance regimes in the context of a non-destructive sampling campaign in 2014. The statistics exclude mosses, dead plants and bare soil, as well as other plant species with a coverage fraction much smaller than one percent.**

| *Species name* | Common name | drained area [%] | control area [%] |
|---|---|---|---|
| *Betula nana subsp. exilis* | Arctic dwarf birch | 9.3 ±21.1 | 0.6 ±2.8 |
| *Calamagrostis purpurascens* | Purple reedgrass | 4.3 ±19.3 | 0.3 ±1.4 |
| *Carex appendiculata* and *lugens* | Tussock forming sedge | 49.2 ±42.8 | 35.7 ±46.5 |
| *Chamaedaphne calyculata* | Leatherleaf | 3.4 ±10.7 | 0.4 ±1.3 |
| *Eriophorum angustifolium* | Cottongrass | 18.6 ±29.0 | 43.3 ±37.7 |
| *Potentila palustris* | Marsh cinquefoil | 2.3 ±8.1 | 9.3 ±20.7 |
| *Salix fuscescens* | Alaska bog willow | 11.4 ±14.5 | 10.3 ±14.6 |
| *Salix pulchra* | Diamondleaf willow | 1.6 ±6.3 | 0.0 ±0.0 |

| *Species name* | Common name | drained area [%] | control area [%] |
|---|---|---|---|

**Table 2: Definition of sub-seasons that reflect the pattern in albedo differences between drainage and control areas. Differences for albedo and net shortwave radiation budget SW(net) were calculated as seasonal means of daily values, drainage minus control. Dates reflect the conditions in data year 2014, and may changes between years.**

| period | start date | end date | duration [days] | albedo difference [-] | SW(net) difference [Wm$^{-2}$] |
|---|---|---|---|---|---|
| closed snow cover | 2/14 | 4/27 | 73 | 2.4 | -2.30 |
| thinning of snow cover | 4/28 | 5/22 | 25 | -2.6 | 7.47 |
| snow cover disappears | 5/23 | 5/31 | 9 | -15.3 | 48.31 |
| flooding transition | 6/1 | 6/18 | 18 | 0.2 | -0.54 |
| snow free season | 6/19 | 9/28 | 102 | 1.9 | -2.18 |
| first snow patches | 9/29 | 10/12 | 14 | 3.4 | -0.98 |
| snow cover buildup | 10/13 | 10/30 | 18 | 12.1 | -2.41 |

**Table 3: Energy flux components averaged over the summer months (June-September) within data years 2014 and 2015, including average and relative differences between drainage and control area observations.**

| parameter | drainage | control | difference | change [%] |
|---|---|---|---|---|
| Net radiation, $R_{net}$ [Wm$^{-2}$] | 94.0 | 95.7 | -1.7 | -1.8 |
| Sensible heat flux, H [Wm$^{-2}$] | 27.9 ±0.13 | 25.5 ±0.12 | 2.4 | 9.4 |
| Latent heat flux, LE [Wm$^{-2}$] | 28.4 ±0.11 | 30.6 ±0.12 | -2.2 | -7.3 |
| Sum of H + LE [Wm$^{-2}$] | 56.2 ±0.18 | 56.1 ±0.17 | 0.2 | 0.3 |
| Bowen Ratio, BR [-] | 0.983 | 0.833 | 0.150 | 18.0 |

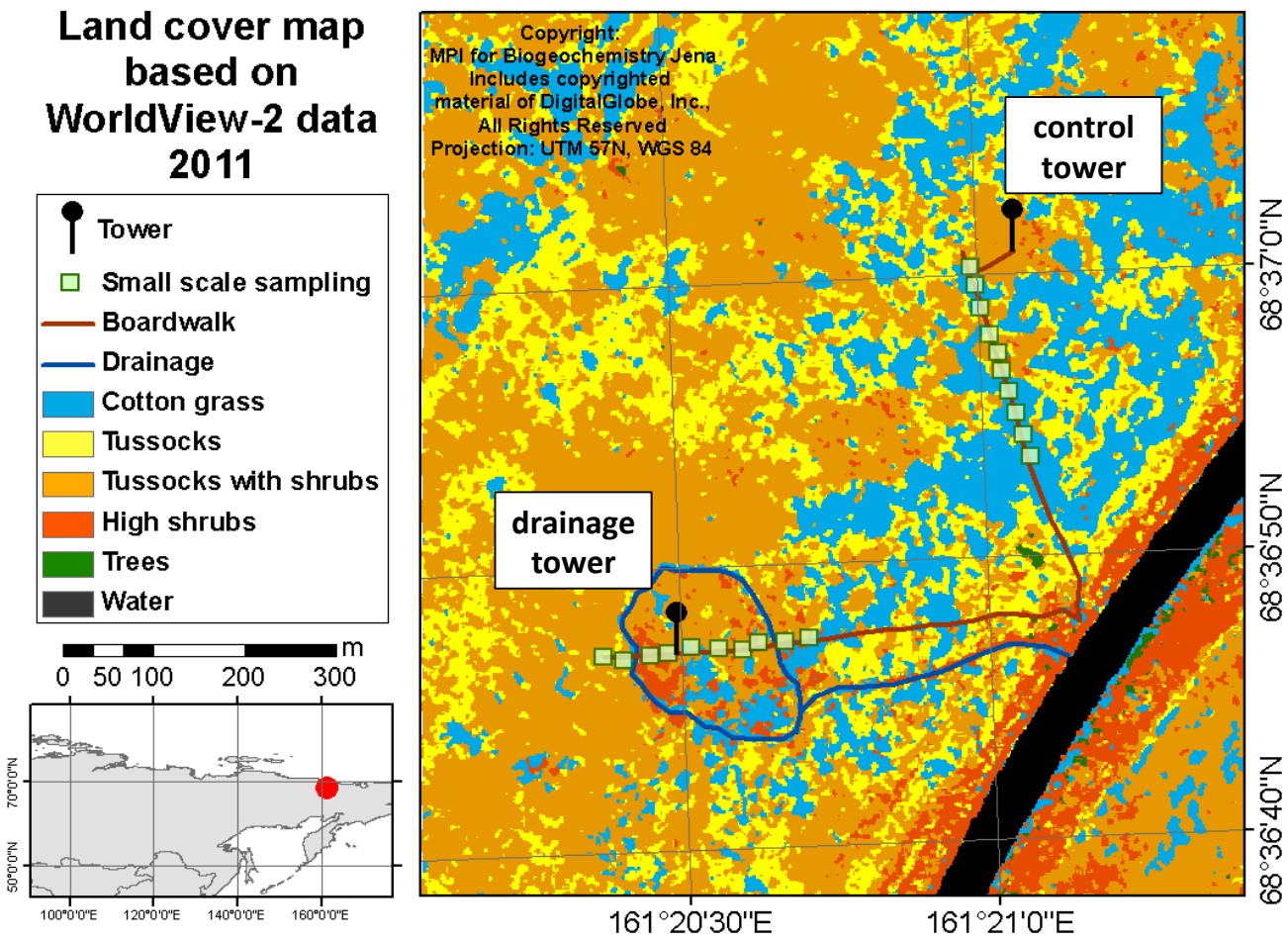

**Figure 1: Land cover structure and instrumentation setup at the Chersky observation site.**

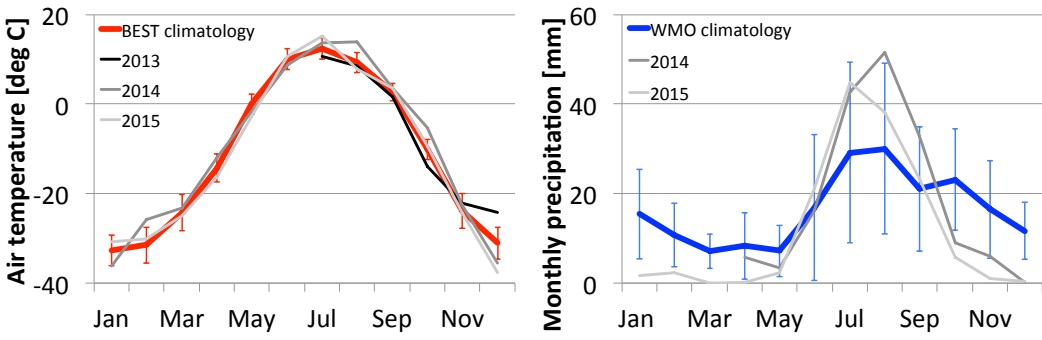

**Figure 2: Left panel: Average annual course (1960-2009) and recent observations of monthly mean air temperature; right panel: Long-term average (1950-1999) and recent observations of monthly precipitation sums for the Chersky region.**

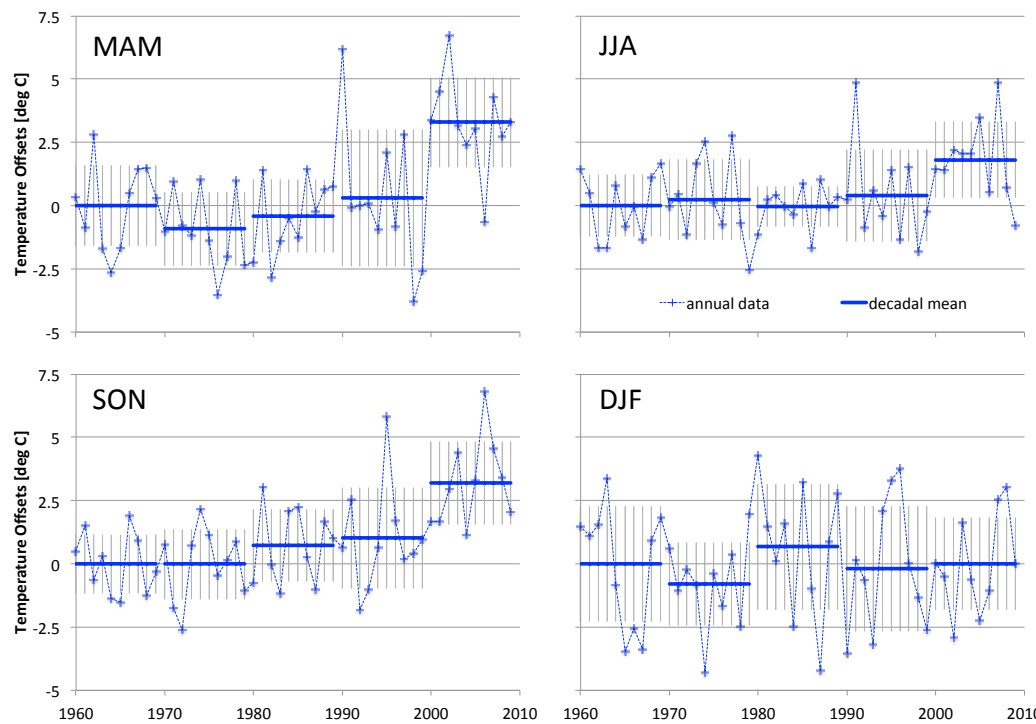

**Figure 3: Temperature trends for the Chersky station, taken from the BEST database. Top left: springtime warming period; top right: core summer period; bottom left: fall freeze-up period; bottom right: core winter period. All temperatures were normalized against the mean seasonal temperatures from the 1960-69 decade. Blue crosses and dashed blue lines give annual data, thick blue lines the decadal mean with vertical grey bars showing decadal RMSE.**

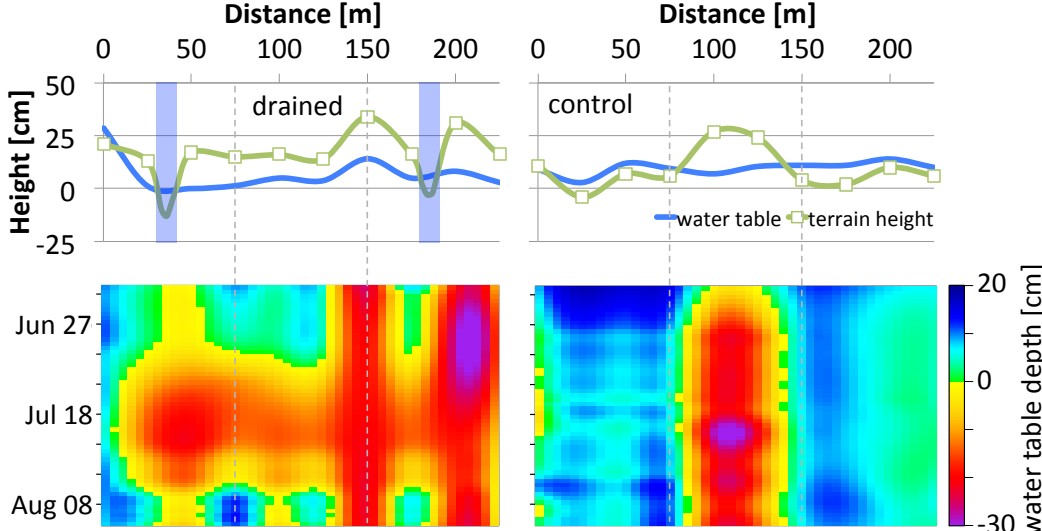

**Figure 4: Top panels: Variability of terrain height (green lines, boxes indicating microsite study positions) and water table depth (blue lines) along transects in the drained (left) and control (right) sections of the observation site (water table depth reflects conditions in mid July 2014). Blue shading in drained panel indicates position of drainage channel, where interpolation of terrain height was modified. The zero level was chosen arbitrarily, so absolute heights do not carry any information in the top panels; bottom panels: Development of water table depth over time for the period Jun 18 to Aug 14, 2014 (interpolated over both space and time). Values are given relative to the surface, with negative values indicating water tables below the surface level.**

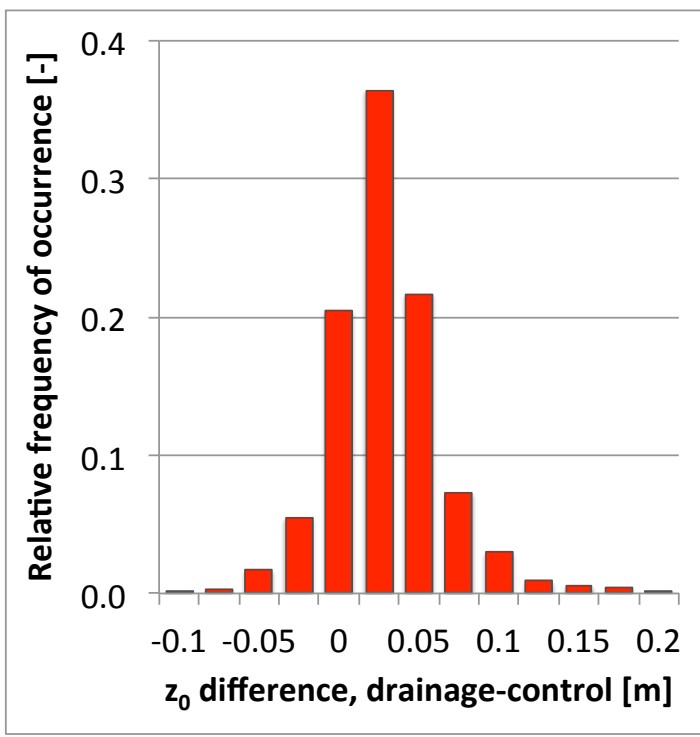

**Figure 5: Frequency distribution of the shifts in aerodynamic roughness length $z_0$ (drained-control) in the drained area, compared to the control area. Data covers the months June through September in 2014/15.**

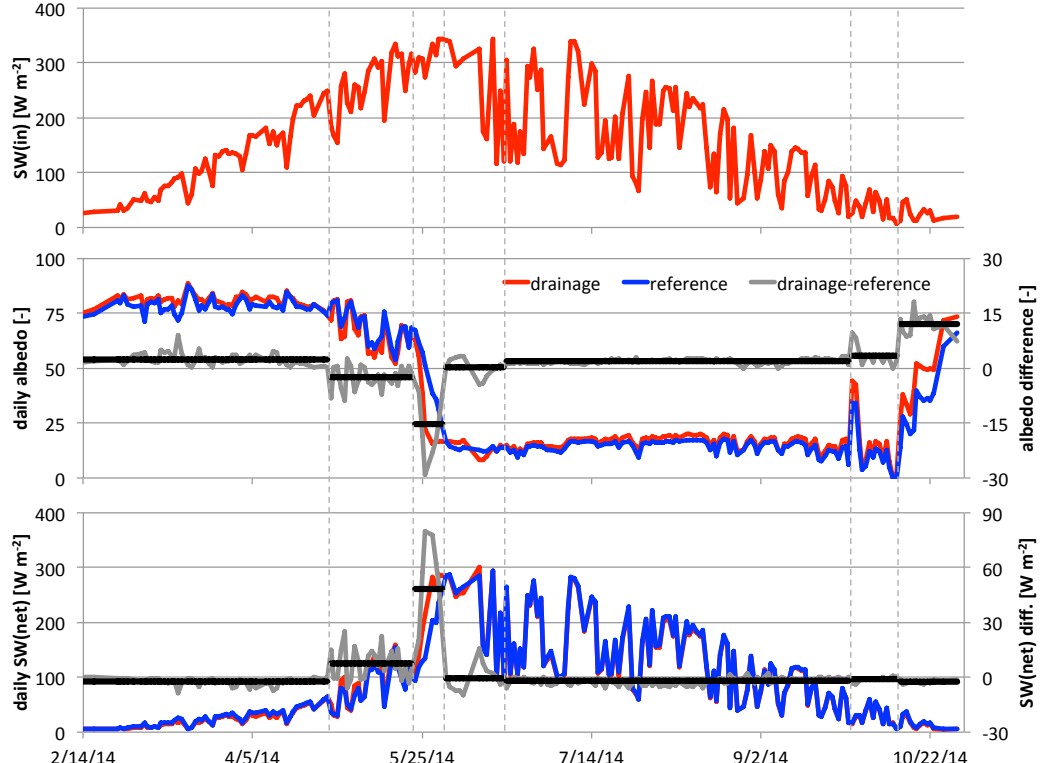

**Figure 6: (top) 2014 annual course of the daily mean incoming shortwave radiation; (center) daily mean albedo derived from the ratio of downward and upward shortwave radiation. The grey line indicates the daily difference between drained and control towers (scales on the right), with the black horizontal bars giving the mean values for seven sub-seasons (separated by vertical dashed lines, see Table 2 for definitions); (bottom) same structure as used in the center panel, here to show the net shortwave radiation budget from both towers and their seasonal differences. All plots have been restricted to the period where SW(in) > 5 Wm$^{-2}$.**

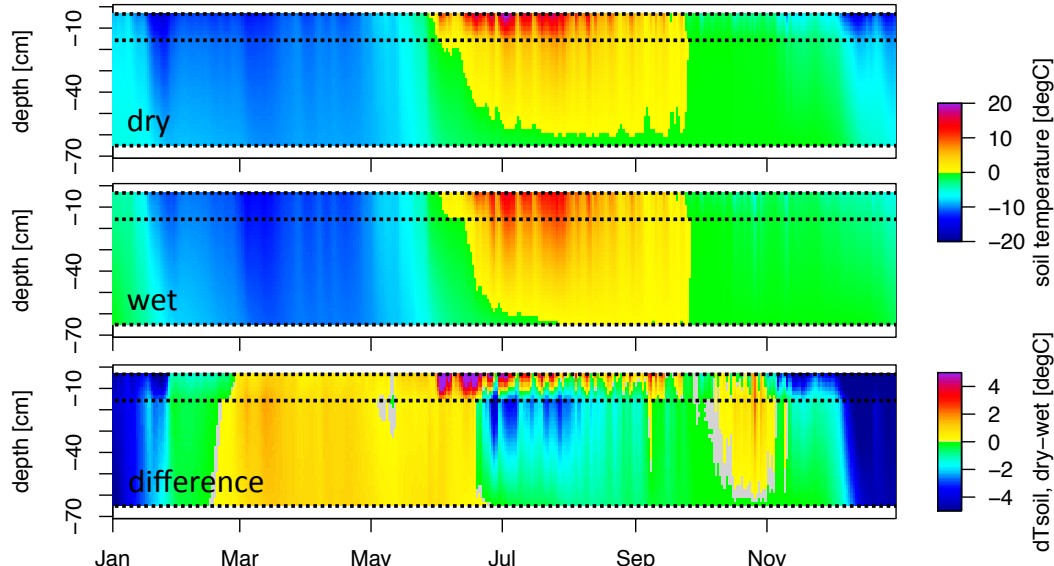

**Figure 7**: **Daily mean soil temperature profiles at a dry (upper panel) and wet (center panel) microsite within the drained transect, and the difference (dry-wet, bottom panel) for the calendar year 2015. Data for three measurement depths (dotted lines) are linearly interpolated along the vertical profile. Values in the difference plot were truncated to the range [-5; 5] °C to enhance visibility of fine-scale patterns, while values fell within the range [-11.3; 10.5] °C.**

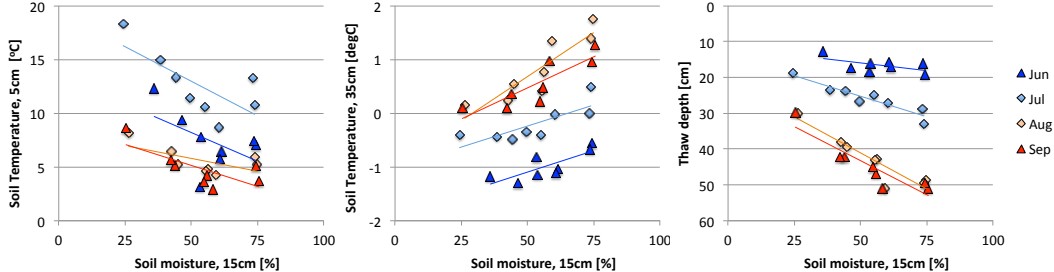

**Figure 8: Influence of soil moisture on soil temperatures and thaw depth at eight microsites covering both disturbance treatments. Observations from summer 2015 are aggregated by site over a period of about two weeks each (Jun 13-30; Jul 01-13; Aug 21-31; Sep 01-12). Lines represent linear regression fits to emphasize trends.**

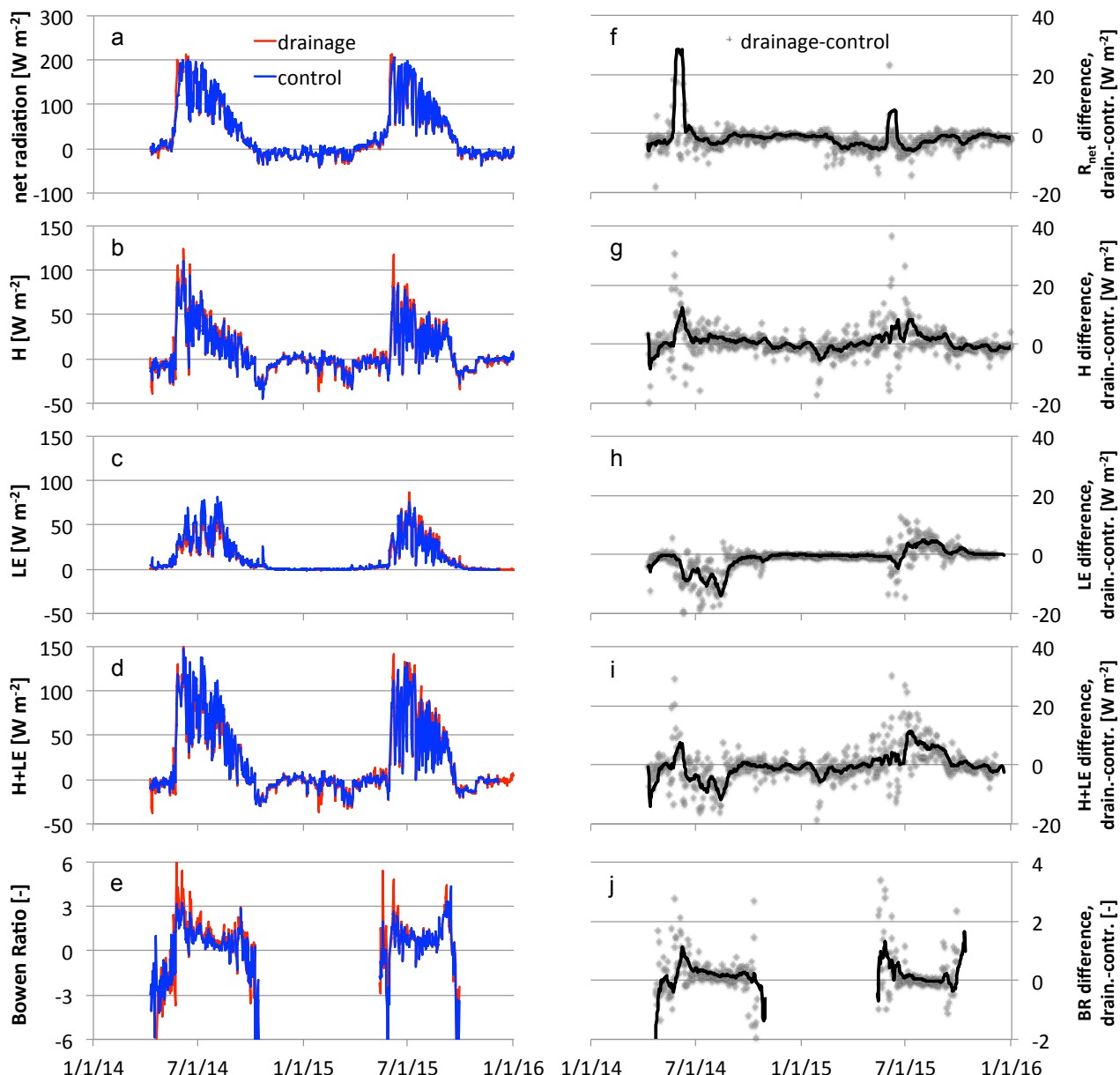

**Figure 9: (a-e)** Summary of daily averaged energy fluxes within drainage (red line) and control (blue line) areas. **(f-j)** Daily differences (drainage minus control, grey crosses) between the two treatments, with black lines giving the average differences for a 15-day moving window. **(a, f)** net radiation; **(b, g)** sensible heat fluxes; **(c, h)** latent heat fluxes; **(d, i)** sum of sensible and latent heat fluxes; **(e, j)** Bowen ratio, which is the ratio of sensible to latent heat fluxes. Vertical scales of difference plots have been truncated to enhance display of fine-scale patterns in overall trends.

# Shifts in energy fluxes linked to drainage-induced changes in permafrost ecosystem structure increase Bowen Ratios, but reduce thaw depth

Mathias Göckede[1], Fanny Kittler[1], Min Jung Kwon[1], Ina Burjack[1], Martin Heimann[1,2], Olaf Kolle[1], Nikita Zimov[3], Sergey Zimov[3]

[1]Department Biogeochemical Systems, Max-Planck Institute for Biogeochemistry, Jena, Germany
[2]Division of Atmospheric Sciences, Department of Physics, University of Helsinki, Finland
[3]North-East Science Station, Pacific Institute for Geography, Far-Eastern Branch of Russian Academy of Science, Chersky, Republic of Sakha (Yakutia), Russia

*Correspondence to*: Mathias Göckede (mathias.goeckede@bgc-jena.mpg.de)

**Abstract.** Hydrologic conditions are a key factor in Arctic ecosystems, with strong influences on ecosystem structure and related effects on biogeophysical and biogeochemical processes. With systematic changes in water availability expected for large parts of the Northern high latitude region in the coming centuries, knowledge on shifts in ecosystem functionality triggered by altered water levels is crucial for reducing uncertainties in climate change predictions. Here, we present findings from paired ecosystem observations in Northeast Siberia that comprise a drained and a control site. At the former, the water table has been artificially lowered by up to 30 cm in summer for more than a decade. This sustained primary disturbance in hydrologic conditions has triggered a suite of secondary shifts in ecosystem properties, including vegetation community structure, snow cover dynamics, and radiation budget, all of which influence the net drainage effects. Reduced thermal conductivity in dry organic soils was identified as the dominating drainage effect on energy budget and soil thermal regime. Through this effect, reduced heat transfer into deeper soil layers leads to shallower thaw depths, initially leading to a stabilization of organic permafrost soils, while the long-term effects on permafrost temperature trends still need to be assessed. At the same time, more energy is transferred back into the atmosphere as sensible heat in the drained area, which may trigger a warming of the lower atmospheric surface layer.

**Keywords:** Permafrost, Disturbance, Drainage, Energy Flux

---

Margin comments:

M G 9/21/2017 10:06 — Deleted: following a decade-long drainage

M G 9/21/2017 09:45 — Deleted: energy

M G 9/21/2017 09:57 — Deleted: transfer to the atmosphere

M G 9/21/2017 10:55 — Deleted: act as a positive feedback to permafrost degradation

M G 9/21/2017 10:55 — Deleted: ed by

M G 9/21/2017 10:55 — Deleted: the

# 1 Introduction

The current state and future evolution of Arctic permafrost, particularly its interactions with the atmosphere, are among the largest uncertainties in our understanding of the Earth's climate system (Schuur and Abbott, 2011; Stocker et al., 2013). The Arctic is one of the most susceptible regions on Earth to climate change (e.g. Serreze et al., 2000; Polyakov et al., 2003; Fyfe et al., 2013; Pithan and Mauritsen, 2014), and altered climate conditions may have enormous consequences for the sustainability of its natural environment (Schuur et al., 2008; Arneth et al., 2010). Interactions between permafrost, climate, hydrology, and ecology have the potential to cause dramatic changes (e.g. McGuire et al., 2002; Hinzman et al., 2005), via mechanisms that are currently poorly monitored and therefore highly unpredictable (Heimann and Reichstein, 2008; van Huissteden and Dolman, 2012; Rawlins et al., 2015). The associated changes in the energy transfer between surface and atmosphere (Eugster et al., 2000; Langer et al., 2011a, b) or the emission patterns of greenhouse gases (Koven et al., 2011) raise the need for experimental studies in this region that address the key uncertainties linked to the functioning of the Arctic in the earth-climate system (Semiletov et al., 2012).

Hydrologic conditions play a pivotal role in shaping Arctic ecosystems. Lakes and rivers in the permafrost region represent an important portion of regional net carbon exchange with the atmosphere, but net emissions over different spatiotemporal scales remain highly uncertain to date (Walter-Anthony et al., 2014; Rasilo et al., 2015). Moreover, the vast carbon pool in northern permafrost regions, currently estimated at 1330-1580 petagrams of organic carbon (Schuur et al., 2015), has accumulated over past millennia since a combination of low mean temperatures and anoxic conditions in areas with flooded or water saturated soils slowed down the decomposition of organic matter (e.g. Ping et al., 2008). In addition to warming, shifts in the water balance in this region are expected to trigger profound changes in the permafrost carbon cycle (e.g. Chapin et al., 2005; Oberbauer et al., 2007). Hydrology can vary at small spatial scales and thus create fine-scale mosaics in surface conditions (Muster et al., 2012; 2013). Even minor differences in mean water levels can impose strong effects on vegetation and microbial community structures or soil thermal regimes (Zona et al., 2011a). It can thus be expected that flooding or draining a site will initiate profound shifts in ecosystem structure, and that the long-term impacts of the disturbance will look very different than the short term changes that can be observed immediately after the event (e.g. Shaver et al., 1992).

Future shifts in hydrologic conditions in the Arctic can be triggered by altered precipitation patterns, where a general trend towards increased rainfall is predicted (Kattsov and Walsh, 2000); however, patterns will vary strongly by region (Huntington, 2006; Bintanja and Selten, 2014), therefore for some areas also lower precipitation can be expected. Geomorphological processes such as subsidence (Jorgenson et al., 2006; O'Donnell et al., 2012) or the formation of a system of connected troughs through the preferential degradation of ice wedges in ice-rich permafrost (Serreze et al., 2000; Liljedahl et al., 2016) can lead to a lateral redistribution of water, and thus create both wetter and drier microsites within a formerly uniform ecosystem. Also, a deepening of the active layer can trigger reductions in waterlogged conditions and wetland extent in permafrost regions (Avis et al., 2011). In-depth insight into the net effect resulting from long-term changes in

M G 9/15/2017 13:03
**Deleted:** Accordingly, besides warming temperatures also

hydrology therefore forms a key challenge for improved future predictions of biogeochemical cycles in Arctic ecosystems (e.g. Lupascu et al., 2014).

Over the past two decades, several studies have documented the effect of an experimental manipulation of soil hydrologic conditions on biogeochemical cycles in Arctic ecosystems (e.g. Oechel et al., 1998; Olivas et al., 2010; Natali et al., 2015). Only a few of these studies examined long-term effects of manipulated water tables (e.g. Christiansen et al., 2012; Lupascu et al., 2014; Kittler et al., 2016), and none of them explicitly addressed the net impact of hydrologic disturbance on the energy budget; only shifts in thaw depth (Zona et al., 2011a; Kim, 2015) and indirect effects of heat fluxes on the carbon cycle (Turetsky et al., 2008) were reported. To date, the only study known to the authors that examined a drainage effect on surface-atmosphere energy exchange patterns focused exclusively on the Bowen Ratio, and found no systematic short-term shifts immediately following a drainage disturbance (Merbold et al., 2009).

Our study presents observational evidence of shifts in ecosystem properties and energy fluxes within a wet tussock tundra ecosystem in Northeast Siberia following a decade-long drainage disturbance, providing insight into the sustainability of ice-rich permafrost under future climate change. As a first major objective, we used paired observations within a drainage and a control area to quantify several secondary disturbance effects linked to lower water tables, including changes in vegetation community, radiation budget and soil thermal regime over longer timeframes. Our second objective was to link these shifts in ecosystem properties to year-round eddy-covariance measurements of energy exchange patterns between permafrost ecosystem and atmosphere, and to evaluate potential feedback effects to Arctic warming. Finally, links between heat transfer into deeper layers and summertime thaw depth dynamics were investigated.

**2 Methods**

**2.1 Study site and drainage experiment**

Our field experiments were conducted on a wet tussock tundra site within the floodplain of the Kolyma River (68.61°N, 161.35°E), approximately 15 km south of the town of Chersky in Northeast Siberia. The area was formed by a relatively recent shift of the main channel of the Kolyma (Corradi et al., 2005), therefore the terrain is flat and no thermokarst lakes or natural drainage channels have yet formed in the vicinity of the observation sites. Still, the existing minor differences in terrain height have led to the formation of a fine-scale mosaic in microsite conditions (Figure 1), mostly triggered by redistribution of water from slightly elevated areas towards the minor depressions.

Site hydrology is strongly influenced by a flooding period in spring caused by melting of the local snow cover as well as a northward moving dam of ice floes on the main channel of the Kolyma that delays melt water runoff. Depending on the timing of snowmelt in the tributary watersheds, this process leads to standing water on the site (up to 50 cm above ground level) in most years around late May to early June. After the flood has mostly receded, stagnant water remains in large parts of the area that only gradually decreases over the course of the growing season.

M G 9/18/2017 15:33
**Deleted:** . Our results demonstrate that drainage disturbance triggers a sequence of changes in ecosystem properties that can only fully develop

M G 9/18/2017 15:34
**Deleted:** . As a net effect, we find increased energy transfer from the surface into the lower atmosphere in the form of sensible heat, which potentially poses a positive

M G 9/18/2017 15:35
**Deleted:** At the same time, severe drainage of shallow organic soil layers reduces

M G 9/18/2017 15:35
**Deleted:** , reducing

M G 9/18/2017 15:35
**Deleted:** , and thus providing a mechanism to temporarily stabilize permafrost

Soils consist of an organic layer of 15-20 cm overlaying alluvial mineral soils (silty clay), with some organic material also present in deeper layers following cryoturbation. More information on site climatology (Section 3.1). spatiotemporal patterns in hydrology (Section 3.2) and vegetation characteristics (Section 3.3) can be found in the Results section. Further details on the study site are presented by Corradi et al. (2005), Merbold et al. (2009) and Kwon et al. (2016).

To study the effects of a lowered water table on a formerly predominantly wet permafrost ecosystem, in the fall of 2004 a drainage ring of approximately 200 m diameter was installed that is connected to the nearby river (Merbold et al. (2009), see also Figure 1). Within its watershed, this drainage system caused systematic changes in the hydrologic regime, lowering the mean water table and shifting the natural wetlands towards conditions with dry soils close to the surface.

## 2.2 Meteorological towers

We established a paired experiment over drained and undisturbed control tundra to facilitate a direct evaluation of drainage effects on ecosystem characteristics and fluxes within this permafrost ecosystem. Since July 2013, two uniformly instrumented eddy-covariance towers with instrument heights around 5 m a.g.l. (Figure 1) captured the turbulent exchange processes between surface and atmosphere continuously throughout the year. Additional meteorological parameters used within the context of this study include air temperature (2 m a.g.l., KPK 1/6-ME-H38, Mela), precipitation (1 m a.g.l., heated

tipping bucket rain gauge, Thies), and long- and shortwave radiation components (5 m a.g.l., CNR4, Kipp & Zonen). For the results presented within Section 3 of this manuscript, all data were aggregated to daily timesteps.

     For more details regarding instrumentation setup, data processing and data quality assessment, please refer to Appendix A and Kittler et al. (2016; 2017), which also summarizes our approach to constrain eddy-covariance data uncertainty. Since we used exactly the same instrumentation at both drainage and control sites, and also the same data processing and quality

assessment approaches, our experimental setup rules out systematic differences in the observational datasets between both treatments that can be linked to methodological issues.

## 2.3 Soil monitoring and vegetation sampling

     In each of the two treatment areas (drainage and control), we set up transects of ten sampling locations placed approximately 25 m apart, yielding two transects with a total length of 225 m each (Figure 1). At each location, we measured water table

depth in permanently installed perforated PVC pipes with 25 mm inner diameter. Thaw depth was assessed by pushing a metal pole into the ground. At every other site, profile probes for soil temperature with sensors at 0.05, 0.15, 0.25 and 0.35 m (Th3-s, UMS, Germany) and TDR soil moisture sensors at 0.075, 0.15 and 0.30 m below the soil surface (CS 640, 630 and 605, Campbell Scientific, USA) were permanently installed. For all these parameters measurements used within the context of this study were taken in conjunction with the operation of a flux chamber system during the period June 15 to August 20,

2014. In addition to the manual sampling, at four of the sampling locations, namely a wet and a dry microsite in each of the treatment areas, soil temperatures were continuously sampled at 0.04, 0.16 and 0.64 m since August 2014.

In 2014, we applied a non-destructive point-intercept method to sample vegetation community structure, with 20 sampling spots of 0.60 x 0.60 m within each treatment. Structuring each sampling spot into a subgrid of 6x6 cells with a side length of 0.10 m, we recorded the vegetation species that intercepted a laser pointer beam when pointing downwards from each grid intersection. The final coverage fraction for each vegetation species was approximated through the relative frequency of occurrence of classes found at these intersection points. More details are provided by Kwon et al. (2016).

**2.4 Site climatology**

Long-term temperature trends for the Chersky region were analyzed based on data from the period 1960-2009 provided by the Berkeley Earth project (berkeleyearth.org, station ID 169921). These Berkeley Earth surface temperature (BEST) time series underwent thorough data quality filtering to flag dubious data, or exclude data biased by instrument issues. Long-term precipitation records for the period of 1950-1999 were obtained from a local weather station through the NOAA online climate database (http://www.ncdc.noaa.gov/cdo-web/datasets, station ID 00025123). No precipitation records were available for the Chersky station from 2000-09. Thus, we analyzed long-term trends from the site Ambarchik station (ID 00025034) situated 100 km north of Chersky with nearly identical mean annual distributions of monthly precipitation sums.

**3 Results**

**3.1 Climate and weather**

**3.1.1 Seasonality in temperature and precipitation**

The averaged (1960-2009) seasonal air temperature course (Figure 2, left panel) has an amplitude of ~45 °C, bounded by a minimum monthly average of -33 °C in January and a maximum of 12 °C in July. The mean annual temperature within these five decades is -11 °C. The averaged annual precipitation sum for Chersky amounts to 197 mm, with mean monthly precipitation varying between 7 mm in March and 30 mm in August (Figure 2, right panel). The bulk of the annual precipitation input is received during the summer (JJA, ~39 %) and fall (SON, ~31 %) months.

Figure 2 summarizes the mean monthly air temperatures observed in the period 2013–15, which represent the datasets used in the subsequent sections. These recent observations follow the general amplitude of the long-term seasonal course. Individual outliers tend to be positive, but the annual mean temperature of -10.9 °C for the period 2014/15 is only slightly above the climatological mean. Since BEST data for Chersky are currently available until November 2013, general offsets between local site-level data and long-term records can only be approximated based on a 3-month overlap period. For these three months, we find a cold bias in the local data ranging between -0.53 and -1.25 °C, explaining the good agreement between recent data years and long-term trends despite the warming trends observed in the past decade (see also Figure 3). Precipitation data display a different pattern since April 2014 (start of measurements) compared to the long-term observations. Total precipitation in 2015 was ~71 % of the long-term trend (154 mm), with a much higher fraction provided

M G 9/22/2017 12:27
**Deleted:** , which

during summer (JJA: +40 %), while winter and springtime contributions are negligible in these years; however, since recent wintertime precipitation measurements are persistently low at very low temperatures, these data are likely to be subject to biases caused by insufficient heating of the sensor. Accordingly, focusing just on summertime conditions our records indicate an increase in precipitation in recent years, compared to the period 1950-1999 that was used for generating the climatology.

### 3.1.2 Long-term climate trends

BEST long-term temperature trends were analyzed broken up into four seasons of three months each: springtime warming (MAM), core summer (JJA), fall freeze-up (SON), and core winter periods (DJF); the setup of these periods intentionally deviates from the definition of seasons in the climatological or plant-physiological sense, instead focuses on comparing temperature trends in different parts of the seasonal course.

The results summarized in Figure 3 indicate heterogeneous trends over both decades and seasons. Using the 1960–69 temperatures as a reference, absolute values and year-to-year variability within all seasons did not change much until the end of the last century. In these first four decades of the analysis, the maximum deviation from the reference was found to be 1.01 °C (SON, 1990–1999), and only minor trends towards warmer conditions were observed for the shoulder seasons (MAM, SON) in the decades between 1970 and 2000. In contrast to this, mean temperatures in all seasons but winter display a step-change when transitioning into the 2000–2009 decade, with observed increases of 3.0 (MAM), 1.4 (JJA) and 2.8 (SON) °C between the 1990s and the 2000s. In spite of the abrupt nature of this transition to considerably warmer temperatures, decadal root mean square errors (RMSE) remain at the same level before and after the change. Wintertime temperatures, however, show no systematic trend over the analyzed five decades, but are subject to pronounced interannual variability in seasonal mean temperatures. The hiatus in warming trends since ~2000 corresponds to global trends for that time period (e.g. Trenberth and Fasullo, 2013), and has been linked to changes in atmospheric circulation and ocean currents (Trenberth et al., 2014).

Precipitation patterns were analyzed in the same way as described above for the temperatures (results not shown). In this case, no discernable trends were found, and decadal mean precipitation stayed within ranges of +/-15 mm (JJA) and +/- 7 mm (all other seasons) in comparison to the 1950–59 references. Based on Ambarchik data, the 2000–2009 precipitation sums also stayed within the same ranges as observed for the five preceding decades, again without discernable trends.

### 3.2 Changes in hydrologic regime as response to drainage

Patterns in water table depth on this wet tussock tundra site are highly variable in both space and time. Accordingly, the effect of the drainage disturbance varies with the microrelief and changes over the course of summer. The top panels of Figure 4 demonstrate the spatial variability of terrain height and water table depth for a time where water levels had fallen to a seasonal low (here: July 14, 2014). The variability of the terrain height at fine scales leads to lateral water redistribution, with the result that one out of ten observation sites at the drained transect still reflects wet conditions, while two out of ten

sites in the control transect have water tables far below the surface level. Accordingly, the area affected by the drainage is not uniformly dry, but conditions have changed from predominantly wet to predominantly dry.

Temporal patterns of the drainage effect are also complex, as highlighted by the bottom panels of Figure 4. After the high water levels from the spring flooding have mostly receded towards the end of June, soil water conditions stay more or less constant across the control transect, while the drained transect displays a pronounced seasonality. For the latter, the water table drops by an average of 10 cm over the first three weeks of July 2014, resulting in dry top soil conditions for all sites but one. This drop is followed by a persistent recovery in water levels as a response to intensive rainfall (see also Figure 2) in late July and early August. Mean water levels within the drained area are always lower than those in the control area; however, as a consequence of the different sensitivity to precipitation inputs between drained and control transects, the net drainage effect results in water table differences between 4 cm and 15 cm (2014 observations), depending on weather conditions in the respective parts of the growing season.

### 3.3 Vegetation structure

In its natural state, the vegetation community of this wet tussock tundra ecosystem is dominated by common cotton grasses (*E. angustifolium*), with tussock forming sedges (*C. appendiculata* and *lugens*) as the second most important species (Corradi et al., 2005; Kwon et al., 2016). This is reflected in the coverage fractions observed during our 2014 plant species assessment (Table 1), which for the control area agrees well with a 2003 plant survey that was taken before the drainage disturbance was implemented (Corradi et al., 2005, not shown). Within the drained area, the decade-long shift of the water regime has strongly reduced the abundance of cotton grasses, while tussock forming sedges as well as shrub species (mostly birch, some willows) now cover larger fractions of the surface as compared to the control.

While the plot-based vegetation inventories indicate an increase in taller shrub species (*Betula exilis* and *Salix pulchra*, Table 1) from 0.6 % to 10.9 % areal coverage, we do not yet have a final quantitative assessment on the increase in shrubs within the footprint areas of the drainage tower. As an approximation, a land-cover classification based on 2011 WorldView-2 satellite remote sensing data (see also Figure 1) indicates an increase in tall shrub coverage by about a factor of four within a 400x400 $m^2$ area surrounding the drainage tower (from 1.2 to 4.8 %), compared to an area of the same size near the control tower. Also, a doubling of areas where tussocks and shrubs are mixed (from 34 to 67 %) was observed. Our datasets do not include a direct assessment of the average height of shrubs in drained and control areas; however, a comparative analysis between both towers (Figure 5) shows that drainage has led to a median increase of 0.026 m in summertime aerodynamic roughness lengths. Aerodynamic roughness length was derived here based on flux-profile relationships using friction velocity and wind speed at tower top under neutral atmospheric stratification. Increases on surface roughness were found to be omnidirectional, varying sector-wise between 0.015 m (SSE) and 0.091 m (NNE). Since both sites are exposed to the same wind conditions, this shift can only be caused by a higher abundance of tussocks as well as more and higher shrubs.

### 3.4 Snow cover, albedo and radiation budget

The changes in moisture regime and vegetation structure described in Sections 3.2 and 3.3 above have a profound impact on the local radiation budget, with the most pronounced effects attributed to shifts in snow cover. While the longwave radiation budget decreased only marginally as a consequence of drainage (-0.2 % during the summer months, data not shown),
shortwave radiation is altered through changes in albedo, with distinct seasonal patterns in the differences between drainage and control areas that allow us to separate the 'light season' (mean daily net shortwave radiation >5 $Wm^{-2}$) into seven distinct periods (see Table 2 and Figure 6 for an analysis of the data year 2014).

In the snow free season, the shift in vegetation in the drainage area causes a minor but persistent increase in the albedo. This albedo change reduces the shortwave radiation budget by about 2 $Wm^{-2}$ throughout the summer since more energy is
directly reflected upwards by lighter colored surfaces. This reduction in energy input agrees with Eugster et al. (2000), who observed increases in albedo linked to the gradual decrease in soil moisture over the course of the growing season. Major changes in albedo are observed during the disappearance and buildup of a closed snow cover in spring and fall. As shown in Figure 6, in both cases the process is initiated earlier in the drainage area, an effect that was observed in all data years since the beginning of the experiment in 2013. Since the thawing period in May falls together with high incoming shortwave
radiation, even a few extra snow-free days lead to a strong increase in incoming energy (see also Table 2). Also the prolonged period where the snow cover starts to thin in spring leads to an energy gain for the drained terrain. Conversely, the earlier buildup of a closed snow cover in fall, as indicated by a rapidly increasing albedo, has little effect on the energy budget since incoming radiation at that time of the year is already very low.

Table 2 and Figure 6 demonstrate that a negative albedo difference between the drainage and control areas is only found
during the snow melt period in spring, which makes up just about 13 % of the total time with sufficient incoming radiation for this kind of analysis (net shortwave radiation >5 $Wm^{-2}$). Since this short period falls together with high incoming radiation, the average impact of albedo changes on the energy budget is still positive for the drained area, with an average gain of 0.6 $Wm^{-2}$ over the total period of almost nine months. However, excluding this short period, the radiative energy input has been reduced as a consequence of drainage and subsequent albedo changes.

### 3.5 Soil thermal regime

The long-term changes in soil water conditions, vegetation coverage, radiation and snow cover listed above have increased both carbon content and bulk density (see Kwon et al., 2017, for details), and also trigger profound shifts on the soil thermal regime within this tundra ecosystem. These effects are summarized in Figure 7, where we compare continuous observations of soil temperatures down to 0.64 m below surface level between a dry and a wet microsite within the drained transect
(similar relationships between moisture and soil temperature regime were observed within the control transect). We differentiate three major effects over the course of the year:

1. Drier conditions close to the surface at the dry microsite reduce the heat capacity of the organic soil. As a consequence, the upper few centimeters of the soil profile get warmer in the period June to August, with differences of up to 10 °C, compared to the wet microsite. At the same time, low thermal conductivity of dry organic soil prevents vertical heat transfer, so within the dry microsite deeper layers remain cooler for most of the period June through December.

2. Within the period November to January, the larger latent heat linked to the higher water content in organic soil within the wet microsite extends the zero curtain period, and delays the freezing for several weeks compared to the dry microsite. Accordingly, soils under dry conditions are much colder in early winter, with temperature differences that can reach up to 10 °C.

3. In the second half of February, the soil temperature decrease within the wet microsite finally catches up with the dry site, and drops down to even colder temperatures. This effect can most likely be linked to the higher thermal conductivity in soils with ice-filled pore space, compared to drier soils with air-filled pores. Deeper snow cover associated with taller vegetation in the dry areas of this tundra site may contribute to this effect by better isolating those sites from the cold wintertime atmosphere. The resulting slightly warmer temperatures across the dry soil profile, which persist until June, were observed within both treatment areas in both years currently covered by the dataset.

Integrating soil temperature measurements of the dry and wet sites within the drainage section for two full data years 2014 and 2015, the average annual soil temperature decreased across the entire soil profile down to 64 cm below surface as a consequence of the drainage. The most pronounced decrease was observed at 16 cm (-0.98 °C), while both shallower (4 cm: -0.42 °C) and deeper (64 cm: -0.27 °C) showed a rather muted response to the disturbance.

To study the effects of moisture on the soil thermal regime in more detail, we analyzed time series of soil temperatures, soil moisture and thaw depth at eight selected microsites. These sites were distributed across both treatments (drained: 5 sites; control: 3 sites), covering wet and dry conditions within both areas. The dataset was collected during two experiments in summer 2015, and was subsequently aggregated into four blocks of about two weeks each (Figure 8).

Close to the surface at 5 cm depth (Figure 8, left panel), wetter conditions lead to colder soil temperatures, confirming the differences between drained and control conditions during summertime already highlighted in Figure 7, related to the lower heat capacity of dry organic soils. The temperature gradient between dry and wet microsites declines in late summer following the reduced radiative energy input (Figure 6). At 35 cm depth (Figure 8, center panel), opposite trends are observed. Here, higher soil moisture promotes higher soil temperatures, which can be linked to the fact that dry organic soil is a poor heat conductor. So even though this soil is warmer close to the surface, compared to wet microsites, the energy is not transferred into deeper soil layers, insulating the underlying permafrost from the heat close to the surface. Gradients grow over the course of the growing season, resulting in temperature differences of almost 2 °C between dry and wet microsites at times.

Higher temperatures in deeper soil layers linked to wetter conditions clearly promote deeper thaw depths (Figure 8, right panel), with differences between dry and wet microsites increasing over the course of the growing season. In September, these differences can amount to more than 20 cm as a result of shifts in the soil water regime.

**3.6 Sensible and latent heat fluxes**

The changes in ecosystem properties following the drainage disturbance have profound impacts on the surface–atmosphere exchange fluxes of energy. In the following paragraphs, all numbers referring to growing season averages or differences represent results for the months June through September (see also Table 3). The winter results are presented after the summer results in the following section.

1. **Net radiation:** Due to a closed snow cover that usually lasts well into the month of May, most of the radiative energy is reflected until around early June (see also Figure 6), when a steep increase in net radiation ($R_{net}$) is observed at both sites (Figure 9a). For the remaining growing season, the general trend in $R_{net}$ follows the solar zenith angle, with only minor impacts by seasonality in albedo. Systematic differences in $R_{net}$ between drainage and control site can only be observed during those few days in spring when the snow cover disappears earlier at the drained site (Figure 9f); however, averaged over the growing season months June through September in 2014 and 2015, which largely excludes the snow melt transition, the energy input is reduced by about -1.7 Wm$^{-2}$ (-1.8 %) caused by the higher albedo in the drainage area (see also Table 3).

2. **Sensible heat flux:** Exchange of energy between surface and atmosphere in form of sensible heat (H) displays a sawtooth pattern over the growing season correlated to the net radiation trends, with a steep incline in June followed by a gradual decrease that lasts through fall (Figure 9b). The timing between the increase in net radiation and H is not aligned, with the onset of high sensible heat fluxes in spring trailing the steep increase in $R_{net}$ by 1-2 weeks. This decrease is interrupted in 2015 by a plateau of H fluctuating around a nearly constant mean within July and August, which is then followed by a steep decline in September. Differences in H between drainage and control area vary strongly between data years, with large differences in 2014 observed only in early summer, while in 2015 the entire growing season displays a pronounced offset (Figure 9g). Overall, sensible heat fluxes have systematically increased as a consequence of the drainage, with peak differences reaching up to 50 Wm$^{-2}$ for single days in early summer, and an average growing season increase of 2.4 Wm$^{-2}$ per day (+9.4 %, Table 3).

3. **Latent heat flux:** In contrast to the patterns observed for the sensible heat flux, latent heat fluxes (LE) start displaying higher values very shortly after the snow melt, but subsequently increase slowly until reaching a maximum in July (Figure 9c). For LE, the signal fluctuates strongly over the largest part of the growing season, with longer term averages stable for the months of June and July. Absolute magnitudes in LE differ strongly between data years in late July and August, coinciding with heavy precipitation in 2014 that increased the fluxes that year. We also observed a pronounced interannual variability in differences between drainage and control areas (Figure 9h), which can be linked to variable weather conditions in different parts of the growing season (see discussion below). In 2014, latent heat fluxes in the drained area were lower compared to the control area by an average of -6.3 Wm$^{-2}$ per day, which can mostly be attributed to reductions in LE in the months of June and July within the drainage. Conversely, during the growing season of 2015 the latent heat fluxes were higher within the drained section (average: 1.9 Wm$^{-2}$ per day), except for a

M G 9/27/2017 11:21
**Deleted:** Figure 9

M G 9/27/2017 11:21
**Deleted:** Figure 9

M G 9/27/2017 11:21
**Deleted:** Figure 9

M G 9/27/2017 11:21
**Deleted:** Figure 9

M G 9/27/2017 11:21
**Deleted:** Figure 9

M G 9/27/2017 11:21
**Deleted:** Figure 9

brief period immediately following the spring flooding. Averaged over both data years, this leads to a net decrease of LE within the summer months of -2.2 Wm$^{-2}$ (-7.3 %,Table 3) within the drainage. However, due to the pronounced interannual variability this mean value may not be representative over longer time periods, and more data years would be required to constrain a net drainage impact.

4.   **Sum of surface-atmosphere energy exchange:** Summing up shifts in H and LE indicates that the total vertical energy flux from the surface to the atmosphere does not change systematically following the drainage (Figure 9j). Mean values and temporal patterns, however, differ strongly between years. In 2014, where increases in H are outweighed by much lower LE fluxes for large parts of the summer months, we observed a net decrease in total energy fluxes (-4.2 Wm$^{-2}$ daily average). Conversely, in 2015 the positive offsets in H and LE combine to a continuous positive difference in
overall energy exchange between treatments, summing up to a daily mean of 4.5 Wm$^{-2}$ over the four growing season months. In both data years, the largest part of the net offsets was accumulated in the later part of the growing season, indicating that temperature and moisture conditions in July and August dominate the interannual variability. Since we found opposing trends with nearly equal magnitude in both data years, averaged over both years the drainage exerts only a minor increase in summertime energy fluxes of 0.2 Wm$^{-2}$ per day (+0.3 %, Table 3).

5.   **Bowen-Ratio:** The Bowen-Ratio (BR), which is the ratio of sensible to latent heat fluxes, summarizes how radiative energy is partitioned between the two vertical energy fluxes H and LE. The higher the BR values, the more the available energy is shifted towards the sensible heat flux, which may indicate water limitations and/or drought stress. At our observation sites near Chersky, absolute daily mean values of BR are usually highest during the early summer, then decrease until August and rise again towards the beginning of fall (Figure 9e). Moving window averages of offsets
between drainage and control show a pronounced peak in June, and are positive throughout the summer months (Figure 9j), implying that the share of energy transferred into sensible heat is systematically higher within the drainage area. Mean summertime Bowen Ratios, calculated based on averaged values for H and LE demonstrate that drainage increases daily mean BR by a value of 0.15 (+18.0 %, Table 3).

**Wintertime fluxes** (here: November 2014 – March 2015) do not contribute notably to net changes in the energy budget at
our observation sites near Chersky. Shortwave energy input is extremely low during the polar winter, and outgoing longwave radiation usually exceeds incoming radiation. Accordingly, on average we observed negative net radiation budgets (drainage: -12.3 Wm$^{-2}$; control: -10.5 Wm$^{-2}$) within both treatments, and values for R$_{net}$ were slightly lower (-1.6 Wm$^{-2}$) in the drainage area. Latent heat fluxes were close to zero for this entire period (drainage: 0.0 ±0.13 Wm$^{-2}$; control: 0.2 ±0.02 Wm$^{-2}$), and average values for the sensible heat flux are in the negative range (drainage: -5.6 ±0.20 Wm$^{-2}$; control: -4.4 ±0.11
Wm$^{-2}$). Combined flux shifts in H and LE of -1.4 Wm$^{-2}$ therefore closely balance the changes in R$_{net}$, so that during wintertime the aboveground energy balance (R$_{net}$ – H – LE) is barely affected by the drainage disturbance.

M G 9/27/2017 11:21
**Deleted:** Figure 9

M G 9/27/2017 11:21
**Deleted:** Figure 9

M G 9/27/2017 11:21
**Deleted:** Figure 9

M G 9/22/2017 12:56
**Deleted:** negative

M G 9/22/2017 12:57
**Deleted:**  indicate a net downward flux from the atmosphere to the surface

## 4 Discussion

### 4.1 Controls on Arctic energy balance

Previous studies have identified vegetation community, atmospheric conditions, and the underlying permafrost as major controlling factors on the biogeochemical cycles of Arctic wetland ecosystems (e.g. Rouse, 2000). Focusing on the energy

budget, Eugster et al. (2000) identified the following properties as dominant features that characterize the specific feedback processes between the Arctic energy balance and climate change: short growing season with long summer days, permafrost and the existence of massive ground ice, prevalent wetlands and shallow lakes, and a nonvascular ground vegetation. All of these factors are closely linked to each other (e.g. McGuire et al., 2002), and thus the assessment of the net effect of climate change on these ecosystems needs to synthesize inter-related responses to capture the cascade of positive and negative

feedback loops that affect geophysical, hydrological, and biological ecosystem characteristics (McFadden et al., 1998; Hinzman et al., 2005). This is supported by the findings from our experiment, where the impact of soil hydrology was identified as a major control, triggering secondary shifts in other factors and ultimately reshaping the ecosystem.

### 4.1.1 Vegetation impact

Many studies that examined the energy budget across ecosystem boundaries in the Arctic have identified the terrain type as

the dominant control on energy flux rates as well as on the partitioning of energy between sensible, latent and soil heat fluxes (Eugster et al., 2000; Eaton et al., 2001; Beringer et al., 2005; Kasurinen et al., 2014). For example, Beringer et al. (2005) studied a vegetation transect in Alaska, and found increasing leaf area index and decreasing albedo along the transition from Arctic tundra to boreal forest ecosystems, which decreased evaporation while increasing transpiration and sensible heat fluxes. Their results demonstrate that successional changes that convert low tundra vegetation into shrub tundra

or woodlands have the potential to intensify vertical energy exchange with the atmosphere, and thus create a positive feedback to warmer conditions under future climate change.

The results presented in this study confirm that vegetation is a key controlling factor within the cascade of secondary shifts in ecosystem characteristics following the primary drainage disturbance. The higher abundance of shrubs, promoted by drier and warmer conditions in shallow soil layers within the drainage area, exerts the most obvious effect on the snow cover

regime for a couple of reasons. The first is that shrubs and tussocks capture the horizontally drifting snow more effectively than cotton grass meadows in the control tundra sections, leading to an earlier accumulation of a closed snow cover in fall (e.g. Sturm et al., 2001a; 2005b) that contributes to warmer soil conditions in the winter season. The second reason is that shrubs sticking out of the otherwise closed snow cover absorb and re-emit the high incoming radiation in April/May, thus contributing to the earlier snow melt observed within the drainage area (Sturm et al., 2005a; 2005b; Pomeroy et al., 2006).

During the summer, it is likely that ground shading by shrub canopies counteracts the observed warming in shallow soil layers (e.g. McFadden et al., 1998); however, based on our dataset, this effect could not be separated from the soil moisture impact on soil temperatures.

M G 9/21/2017 14:05
**Deleted:** the dominant

M G 9/18/2017 20:59
**Deleted:** and deeper

M G 9/18/2017 21:00
**Deleted:** , and subsequently to

Both plot surveys and gridded maps do not include information on vegetation height. Still, the available ground-based and remote sensing information on shifts in vegetation community structure provides strong evidence that drainage induces shifts towards taller vegetation that play a major role in the overall transformation of the ecosystem structure following the drainage disturbance. During summertime, our eddy-covariance fluxes indicate an omnidirectional increase in aerodynamic roughness length (Figure 5) and mechanically generated turbulence (as represented by the friction velocity, data not shown) at the drainage tower, compared to the control tower, indicating that higher vegetation surrounding that tower intensifies turbulent exchange processes between surface and atmosphere.

The virtual absence of mosses at our observation sites (~1.8% in both transects, data not shown) is also expected to influence energy flux rates and soil thermal regime. Moss cover usually dominates the surface in high latitudes, and has been shown to play a key role in modifying temperature and moisture conditions in Arctic soils (Beringer et al., 2001; Zona et al., 2011b; Kim et al., 2014). Evaporation from mosses contributes a significant portion of the latent heat flux in Arctic ecosystems (McFadden et al., 2003; Beringer et al., 2005). Moss insulation reduces the soil heat flux and increases energy exchange with the atmosphere (Beringer et al., 2001), similar to the insulation effect of dry organic soil that was observed in the present study. It can therefore be assumed that drainage could further reinforce the moss-cover effect on energy flux partitioning and soil thermal regime, as long as the drained soil can still sustain the moss layer.

**4.1.2 Atmospheric impact**

Numerous studies identified atmospheric forcing as the most important control on the energy budget of individual high latitude ecosystems in the absence of vegetation shifts (e.g. Rouse et al., 1992; Harazono et al., 1998; Kodama et al., 2007; Boike et al., 2008; Langer et al., 2011a). Precipitation anomalies were found to be a major determinant for interannual variability in energy partitioning (Boike et al., 2008), and Rouse et al. (1992) listed variations in the total amount of summertime precipitation as a control for active layer depth development. Synoptic patterns exerted a strong influence on energy fluxes in both Alaskan (Harazono et al., 1998) and Siberian (Kodama et al., 2007; Boike et al., 2008; Langer et al., 2011a) study domains. Onshore winds coming from the ocean, characterized by cold and wet conditions, promote sensible heat fluxes since they increase the temperature gradient between surface and atmosphere. Warm and dry offshore winds, however, usually decrease the sensible heat flux. Also, the impact of cloud cover on the longwave radiation budget had an important influence on soil temperatures and freeze-back patterns in fall (Langer et al., 2011a).

Short-term shifts in weather conditions, such as a warm bias in August 2014 (see Figure 2), exerted only a slight influence on the seasonal trajectories of the sensible heat flux. In contrast, an average long-term reduction in net radiation of about 2 % during the period June through September 2015, compared to 2014, decreased the H budget by about 9 %. No systematic differences in the interannual variability of H were observed between drained and control areas, wherein the overall reductions in the flux rates were of equal magnitude, and both areas reacted uniformly to changes in atmospheric forcings. Regarding the longwave radiation, even though this parameter barely changed as a consequence of drainage the

interannual variability of the upward directed component correlated well with the sensible heat fluxes, both in terms of absolute values and differences between treatments.

Regarding the latent heat flux, similar to Boike et al. (2008) we found an influence of precipitation patterns on interannual variability in LE, particularly in late summer and within the drainage section of our study area. For example, heavy rainfall events such as those observed in late July and early August 2014 can lead to partially waterlogged conditions within the drainage ring (Figure 4), and increased evapotranspiration rates at these times often match those in the control area. Moreover, in contrast to the sensible heat flux, we found evapotranspiration rates highly susceptible to day-to-day variability in net radiative energy input. Regarding interannual variability, we observed opposing trends in summertime LE within drained and control areas, wherein compared to 2014, in 2015 latent heat flux rates increased by an average of ~5 % in the drained areas, while the control fluxes decreased by about 20 %.

Comparing weather conditions and energy fluxes between 2014 and 2015, the reduction in net radiation input in 2015 led to a pronounced reduction in vertical energy transfer to the atmosphere in the control section (about -16 %, combining H and LE), while the net change in turbulent fluxes in the drained area was much lower (-2 %). We speculate that the deviating interannual variability between the sections may be driven by differences in soil moisture levels between data years, e.g. linked to the timing of precipitation events, which influenced the feedback of LE flux rates to variability in net radiation. These findings suggest that the timing of shifts in weather conditions played an important role for latent heat fluxes within the drainage section, and dominates interannual variability. In particular, the slightly warmer conditions in June and July of 2015, in combination with the frequent occurrence of light to moderate rainfall events, resulted in higher latent heat fluxes in this period, compared to 2014.

Synoptic influences were not analyzed in detail in the context of this study. Our focus was placed on the differences in ecosystem structure and surface-atmosphere energy exchange between a drained and a control observation site. Since both sites were placed only approximately 600 m apart, it can be assumed that both are exposed to the same atmospheric forcing. A direct comparison of weather conditions measured at both towers, including air temperatures, humidity, pressure and precipitation, resulted in no systematic offsets going beyond the calibration accuracy of the employed instrumentation.

### 4.1.3 Impact of permafrost

A primary characteristic of permafrost landscapes is that the ice-rich frozen layer inhibits vertical water losses, preserving water-logged conditions during large parts of the summer, and thus facilitating the establishment of wetlands even in areas with relatively low precipitation input (Rouse, 2000). Within permafrost landscapes, a large portion of the net energy input from radiation is used to thaw the frozen ground, and increase the thaw depth over the course of the growing season (e.g. Lund et al., 2014). Permafrost constitutes a substantial heat sink, reducing the soil temperatures and also the energy available to feed turbulent heat flux exchange with the atmosphere (Eugster et al., 2000; Langer et al., 2011a). Accordingly, permafrost acts as an efficient buffer against the intensification of energy exchange that might be triggered by a warmer future climate in high latitude regions, but this controlling mechanism would be reduced in case the thaw depth increases

(Lund et al., 2014). Another particular feature of northern permafrost soils is that the very cold frozen ground generates a downward directed heat flux from the snow in spring. This increases the amount of energy required to melt the snowpack, and accordingly delays melt (Eugster et al., 2000). The warmer soil conditions found within the drainage area in late winter, which are likely caused by a higher thermal conductivity in ice-rich soils, supported by a better ground insulation through increased snow depth captured by higher shrubs, would reduce the downward heat flux, and therefore may contribute to an earlier snowmelt.

For the reasons listed above, wet permafrost ecosystems are often characterized by a large fraction of the soil heat flux in the energy budget (Boike et al., 2003; Langer et al., 2011a) that can be of the same order of magnitude as latent and sensible heat fluxes (Eugster et al., 2000). The partitioning of net radiation in permafrost landscapes is particularly sensitive towards the moderation of bulk surface resistance by the vegetation, and the thermal conductivity of shallow soil layers (Harding et al., 2002; Liljedahl et al., 2011). In the current study, soil heat fluxes were not measured directly, so their magnitude can only be approximated based on the difference of energy supply from net radiation and energy demand for latent and sensible heat fluxes. This residual amounted to 39–46 % depending on data year and site, or 14–31 % assuming that 15-25% of the net radiation is attributed to an unclosed energy balance (Foken et al., 2011) or additional energy sinks such as heat storage in water (Harazono et al., 1998). While residuals in 2014 and 2015 were similar in the drainage area, values increased by about 8 % in the control section, mostly driven by shifts in LE.

**4.1.4 Impact of soil hydrology**

The moisture and thermal regulation of Arctic wetland ecosystems is strongly influenced by the presence of an organic soil layer, which features an extremely large water content when wet, and equally large air content when dry (Rouse, 2000). Water saturation conditions affect heat flux rates into soils, and also alter thaw depth and the net ecosystem-atmosphere heat exchange (Jorgenson et al., 2010; Subin et al., 2013). Moreover, waterlogged conditions can increase solar absorption, contributing to increased thaw depths and potentially to a subsequent lowering of the water table with respect to the surface (Olivas et al., 2010). Conversely, lowering the water table can also preserve moisture in the ecosystem in the absence of plants with high leaf area index and deep roots, since higher albedo reduces net radiation, and poor thermal conductivity in dry soils keeps deeper layers colder (Eugster et al., 2000).

We found both direct and indirect effects of shifts in soil hydrology on the energy budget of our study site, with the latter likely to have a stronger impact on the long-term trajectory of this permafrost ecosystem than the former. The combined impact of the most prominent indirect effects, the shifts in vegetation community structure and snow cover regime, will be discussed in more detail in the following section. Direct effects comprise the decrease in both heat capacity as well as thermal conductivity when drying out organic soils, and our results show indications of reduced soil heat fluxes across the seasons. The observed patterns between soil water content and progress of thaw depth over the course of the growing season agree well with findings previously reported for Arctic ecosystems in Alaska (Hinzman et al., 1991; Shiklomanov et al., 2010; Sturtevant et al., 2012). A second direct effect of drier conditions in shallow soil layers is a shift in

energy partitioning towards higher Bowen Ratios, with a larger portion attributed to sensible heat fluxes. Also, after drainage particularly the latent heat fluxes become more dependent on short-term atmospheric forcing, with strong variability observed in LE rates related to precipitation input.

Regarding the net effect of dry soils on soil heat fluxes and thermal regimes, different pathways are possible, depending largely on soil type and the severity of the disturbance. Rouse et al. (1992) observed that higher temperatures and associated moderate increases in evapotranspiration moved water tables beneath the surface, but the peat soils at their study site remained wet. As a result, thermal conductivity decreased less rapidly than heat capacity while the thermal diffusivity was enhanced, which led to deeper thaw depth with warm and dry climate conditions. In contrast, Bonan et al. (1990) postulated that the greater evapotranspiration accompanying climate warming would dry out the surface organic layer, causing a reduction of soil heat fluxes and thus stabilize the existence of permafrost. Our observation of cooler summertime temperatures in deeper soil layers following drainage, and a substantial reduction of thaw depth in the drained section, support the statement by Bonan et al. (1990); however, including secondary drainage effects, where the combination of higher vegetation and higher snow pack leads to a warmer soils during winter, over longer timeframes drainage may also lead to a gradual warming of soil temperatures, and therefore contribute to permafrost degradation.

**4.2 Implications for feedback processes with climate change**

**4.2.1 Interrelated ecosystem responses**

Sophisticated numerical models are needed for assessing the complex feedback processes between permafrost ecosystems and climate change, but is unclear yet which processes need to be explicitly resolved in these models, and which input parameters need to be provided at what resolution, to avoid systematic biases in simulation results (Eugster et al., 2000). Our findings indicate that process-based models representing permafrost ecosystems to a high degree of detail would be required for this objective. We found direct effects of the primary drainage disturbance on the energy budget of our study site, but also secondary effects based on shifts in other ecosystem properties following the lowered water tables. Also interactions between secondary effects were observed, including feedback processes to the primary drainage. Neglecting this network of positive and negative feedback loops on permafrost ecosystem energy budgets will likely lead to biased net effects, and therefore distort both the simulations regarding the sustainability of individual high latitude ecosystems under climate change, as well as their relevance at regional to global scales. In fact, at the time when our observations took place, a clean separation between e.g. primary drainage effect and the changes in vegetation cover that followed it could not be made without additional manipulation to the ecosystem.

The specific hydrologic conditions created by permafrost ecosystems, characterized by a barrier to infiltration posed by the frozen ground, provided the prerequisite to form the original wet tussock tundra ecosystem. Regarding the primary disturbance (the lowering of the water table), impacts on the ecosystem were initially restricted to shifts in the partitioning of energy between sensible, latent and soil heat fluxes during summertime. As a direct secondary effect, a taller vegetation

M G 9/21/2017 09:18
**Deleted:** influence factors and ecosystem characteristics

community with higher aerodynamic roughness was established, causing higher mechanically generated turbulence and also a slightly higher albedo in summer. The vegetation shift then triggered substantial changes in the snow cover regime, with higher albedo and warmer soils in late winter and spring, and an energy pulse in early summer related to the expedited snow melt. Every single effect will influence the net annual energy budget of this permafrost ecosystem, and accordingly they can

only be analyzed as a network of closely linked properties when assessing the net impact of drainage disturbance.

One particularly interesting network effect of combined shifts in ecosystem properties is the interaction between drainage and vegetation shifts. The increased abundance of shrubs in high latitude regions has been studied extensively (e.g. Sturm et al., 2001b; Myers-Smith et al., 2011; DeMarco et al., 2014a), and numerous studies have identified links with factors such as snow cover (Sturm et al., 2001a; Pomeroy et al., 2006), radiation regime (Bewley et al., 2007) and nutrient

cycling (Myers-Smith and Hik, 2013; DeMarco et al., 2014b). Regarding soil temperatures, the isolated assessment of shrub expansion indicates that the capture of drifting snow by shrubs increases snow depth and soil temperatures in winter, while shading decreases temperatures in summer (e.g. Sturm et al., 2001a). Assessing the net impact of these opposing effects depends on details like shrub density, which complicates the evaluation of this effect at landscape to pan-Arctic level. Further, including a drainage effect increases the complexity. For example, during winter the shifts in snow cover caused by

the taller vegetation still persist, but the warming effect is modulated by the low heat capacity and low thermal conductivity of dry organic soil, which decreases soil temperatures earlier in fall, but causes higher soil temperatures towards the end of winter. During summer, in shallow layers the warming triggered by the low heat capacity of dry organic soil dominates over the shading effect of the shrub canopy, while in deeper layers the shrub cooling effect is reinforced by low thermal conductivity. Accordingly, compared to the isolated assessment of shrub expansion, a drained permafrost ecosystem with a

higher abundance of shrubs may experience a higher net warming in winter, but shows colder temperatures in deeper layers in summer.

The severity of the drainage needs to be considered when interpreting the outcome of hydrologic disturbance effects. As summarized in the introduction, many studies have treated the impact of a lowered water table on biogeochemical cycles in permafrost ecosystems. However, for most of them the water table was lowered less than 10 cm, so the differences before

and after disturbance were not as pronounced as to be expected after the formation of a trough system following the degradation of ice-rich permafrost. According to Rouse et al. (1992), moist organic soils can still act as good thermal conductors, so reduction of thaw depth, as observed in our experiment, can only be found after a dramatic shift in soil water tables that largely dry out the shallow organic layers.

### 4.2.2 Consideration of long-term effects

As pointed out in the previous section, a combination of primary and secondary disturbance effects, and their interactions, generates the net impact of the drainage disturbance on the energy budget. In this context, temporal aspects need to be considered as well when interpreting the results. While the primary disturbance immediately affected the water table, other key components like the vegetation community structure take longer to change. Consequently, very different net effects were

---

**Deleted comments (margin):**

M G 9/21/2017 14:36
Deleted: t exchange fluxes

M G 9/21/2017 09:20
Deleted: multiple

M G 9/21/2017 09:20
Deleted:  have been identified

M G 9/21/2017 09:22
Deleted: ing

M G 9/21/2017 09:22
Deleted: them

M G 9/22/2017 13:34
Deleted: leads to colder

M G 9/22/2017 13:34
Deleted: in

M G 9/22/2017 13:34
Deleted: y

M G 9/22/2017 13:35
Deleted: s less

M G 9/22/2017 13:36
Deleted: and also

M G 9/21/2017 14:44
Deleted: Accordingly, gradually drying out tundra soils does not lead to a corresponding smooth transition from one state of energy flux partitioning to another, but instead it can be assumed that a tipping point in soil moisture levels exist that, once reached, non-linearly transitions from one state to another.

found in the year immediately following the disturbance (Merbold et al., 2009), compared to our more recent results from about 10 years later (Kittler et al., 2016). At the same time, it can be assumed that the system is even now not fully equilibrated towards the new conditions, and further shifts in factors like shrub coverage and canopy height can be expected for the long-term trajectory of this site. Still, we expect that the major elements of change are already established by now, and no major shifts in the overall functionality of the energy budget feedback processes are expected for the future.

Another factor with a long-term trajectory regarding the response towards drainage disturbance is temperature trends in deeper soil layers. Throughout the studied section of the soil profile, we observed alternating periods dominated by warming or cooling over the course of the year, with all layers down to 64 cm below surface experiencing a net cooling following the drainage. Accordingly, particularly in deeper soil layers we see a complex combined impact of a reduced cooling down of soils in winter, probably linked to the differences in thermal conductivity between ice-filled and air-filled pores, and shifts in the insulation by dry organic soils, preventing vertical heat transfer in summer. Whether or not the wintertime warming will eventually be substantial enough to dominate over the summertime cooling and impact thaw depth cannot be evaluated with the currently available database. An assessment of the long-term temperature trends would either require a suitable process-based modeling framework, or longer-term temperature observations, ideally including deeper boreholes, and thus is beyond the objectives of this study.

### 4.2.3 Implications for Arctic climate change

An increase in sensible heat flux rates as a response to shifts in the partitioning of the available net radiation is the most direct pathway to change the temperature of the atmospheric boundary layer (Eugster et al., 2000; Lund et al., 2014). This statement is supported by Chapin et al. (2000), who expect a high potential for positive feedbacks between land-atmosphere energy exchange and regional temperature changes throughout high-latitude regions, disregarding potential shifts in land cover structure. Eugster et al. (2000) therefore count the unknown magnitudes of changes in energy partitioning as well as the lack of long-term energy balance data from Siberia among the most important uncertainties for assessment of susceptibility and vulnerability of Arctic ecosystems to climate change.

Our experiment triggered an artificial lateral redistribution of water, converting a formerly uniform wet tussock tundra ecosystem into severely drained terrestrial areas intersected by a drainage channel. This approach mimics the preferential degradation of ice wedges in ice-rich permafrost under climate change, which can lead to the formation of a system of connected troughs (Serreze et al., 2000; Liljedahl et al., 2016). Our results can therefore provide observational evidence of energy budget shifts that can be expected in regions susceptible for this type of degradation. Ice-rich permafrost (Yedoma) covers large parts of the North American and Siberian Arctic (e.g. Strauss et al., 2013), and in the Siberian plains has been found to contain very high ice contents that made up 40 to 70 % of the soil volume (Zimov et al., 1997). It can be expected that geomorphological evolution and hydrological responses to permafrost degradation as a result of longer-term effects of a warming Arctic climate (e.g. Hinzman et al., 2005) will constitute a large-scale phenomenon with potential pan-Arctic and even global feedback implications on climate change.

M G 9/22/2017 13:38
**Deleted:** improved snow insulation

Regional changes in energy fluxes will probably interact with the carbon cycle by changing the disturbance regime, regional temperature and precipitation, and the depth of the boundary layer (Chapin et al., 2000). Moreover, shifts in energy budgets may also alter the delicate patterns of regional scale energy transfer within the Arctic and beyond. The redistribution of energy from certain regions in the Arctic and boreal zone to northern areas which is observed under current conditions (for example the heat flows from Alaska in both northerly and easterly directions) may increase under a warming climate whenever the energy transfer from surface to atmosphere increases (Eugster et al., 2000). Accordingly, the effects of shifts in energy exchange patterns following the degradation of ice-rich permafrost will not be restricted to these areas, but are likely to exert effects of substantial magnitude at the pan-Arctic scale.

**5 Conclusion**

Degradation of ice-rich permafrost under future climate change holds the potential to transform geomorphological and hydrological characteristics within large parts of the Arctic. Persistently drier or wetter conditions may alter ecosystem structure dramatically, and lead to systematic shifts in biogeophysical and biogeochemical processes. Both the effects of long-term equilibration and non-linear feedback processes between shifts in ecosystem components complicate the assessment of the net impact of this type of disturbance, therefore the long-term trajectory of ice-rich permafrost ecosystems in the Arctic is highly uncertain to date.

We presented observational evidence on the potential long-term consequences of sustained drainage in a previously wet tussock tundra ecosystem in Northeast Siberia. Our datasets indicate that a decade-long lowering of the water table triggered a cascade of secondary changes at our observation sites, including shifts towards taller vegetation, modification of the snow-cover period, and profound shifts in soil temperatures. Warmer conditions were found throughout the soil profile (i.e. down to 64 cm below surface level) towards the end of winter; during the summer, however, the low heat capacity of dry organic soils most likely led to warmer conditions in shallow layers, while at the same time poor thermal conductivity kept deeper layers colder. Since dry organic soils are efficient insulators, drainage can reduce thaw depth, and can initially protect deep permafrost carbon pools from degradation under warmer climate conditions. However, the net impact of wintertime warming and summertime cooling on long-term permafrost temperature trends following drainage still remains to be assessed.

With heat transfer into the soil systematically reduced in summer, the primary drainage disturbance in combination with related secondary changes led to a minor increase in energy transfer back to the atmosphere, even though higher albedo following vegetation shifts slightly lowered the energy input through net radiation. Mean sensible heat flux rates increased in the drainage area (+2.4 Wm$^{-2}$, or +9.4 %), closely balanced by a parallel decrease in averaged latent heat fluxes (-2.2 Wm$^{-2}$, or -7.3 %). This shift in energy flux partitioning, reflected by a Bowen Ratio increase of ~18 %, may lead to secondary feedback effects in temperature and moisture regimes of the lower atmosphere, and can thus aggravate climate change impacts once the degradation of ice-rich permafrost has initiated hydrologic redistribution. If our local scale results can be

M G 9/21/2017 14:46
**Deleted:** multi-disciplinary

M G 9/21/2017 09:25
**Deleted:** (see also Figure 10)

confirmed for different Arctic regions, and different hydrologic disturbance scenarios, the demonstrated effects could therefore be relevant for forecasts of the sustainability of Arctic permafrost ecosystems under future climate change.

M G 9/21/2017 15:38
**Deleted:** have a profound impact on

## Acknowledgements

This work was supported through funding by the European Commission (PAGE21 project, FP7-ENV-2011, Grant
Agreement No. 282700, and PerCCOM project, FP7-PEOPLE-2012-CIG, Grant Agreement No. PCIG12-GA-201-333796), the German Ministry of Education and Research (CarboPerm-Project, BMBF Grant No. 03G0836G), and the AXA Research Fund (PDOC_2012_W2 campaign, ARF fellowship M. Göckede). The authors appreciate the contribution of staff members of the Northeast Scientific Station in Chersky for facilitating the field experiments, especially Galina Zimova and Nastya Zimova. We would also like to thank the administration and service departments within the Max-Planck-Institute for
Biogeochemistry, most notably the Field Experiments & Instrumentation group, for their contributions to planning and logistics, and for supporting field work activities.

We applied sequence-determines-credit (authors 1-3) and equal-contribution (alphabetical sequence, all other authors) methods for the order of authors.

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

**Appendix A: Eddy covariance instrumental setup and data processing**

The eddy-covariance observations at our field site near Chersky comprised a drainage tower (68.61 °N and 161.34 °E, instrument height 4.91m a.g.l.) and a control tower (68.62 °N and 161.35 °E, 5.11m a.g.l.) representing undisturbed natural conditions (see also Figure 1). Both towers were placed at an elevation of approximately 6 m a.s.l., with an average vegetation height of 0.7 m during summer. Each eddy tower featured a heated sonic anemometer (uSonic-3 Scientific, METEK GmbH) to monitor three-dimensional wind fields and sonic temperatures, and a closed-path gas analyzer (FGGA, Los Gatos Research Inc.) to capture mixing ratios of $CH_4$, $CO_2$ and $H_2O$. A heated and insulated sampling line (16.0 m and 12.8 m length for drainage and control towers, respectively) connected the inlets close to the sonic anemometer with the gas analyzers. An external vacuum pump (N940, KNF) facilitated a nominal flow rate of 13 L min[-1], equivalent to a replacement rate of ~ 2–2.5 Hz in the measurement cell.

Eddy-covariance data was recorded at 20 Hz. We collected primary data using the software package EDDYMEAS (Kolle and Rebmann, 2007), and based all further processing of the high frequency data on the TK3 software (Mauder and Foken, 2015). The flux processing sequence included 2D coordinate rotation of the wind field, cross-wind correction (Liu et al., 2001), spectral correction (Moore, 1986), and an online conversion of gas analyzer output to dry mole fractions before processing by the TK3 software. We furthermore applied cut-off frequencies of 2 Hz for the $H_2O$ data to correct losses linked to tubing effects and limited gas replacement rates in the closed-path analyzers.

At the core of our post-processing quality control scheme, we follow the flux data quality flagging scheme presented by Foken (2008), which is an update of the method proposed by Foken and Wichura (1996). Their original tests on stationarity and well-developed turbulence were amended for this study by flags also reflecting overall errors in the log file recorded by the sonic anemometer, accordance of instrument operational limits with prevailing boundary conditions, and sonic anemometer heating status. Gapfilling and flux partitioning were both based on the method of Reichstein et al. (2005), using separate gapfilling and flux partitioning data pools for different seasons.

The assessment of eddy-covariance flux data uncertainty was based on well-established concepts (e.g. Aubinet et al., 2012) that assess the total error as the combination of random and systematic errors. Systematic errors are composed of errors associated with unmet theoretical assumptions and methodological challenges of the eddy-covariance technique, data processing errors, and instrumental calibration issues (Mauder et al., 2013). Since within this study, we were primarily focusing on flux differences between two equally equipped eddy-covariance systems set up close to each other, the first two items in this list could be neglected, based on the fact that a uniform data processing has been applied that considers all recommended quality filters and correction procedures. Regarding instrument calibration, we did not find a systematic offset in the frequency distributions of wind speed, sonic temperature and water vapor mixing ratios between both towers (data not shown), indicating that the calibration of the uniform instrumentation did not introduce a systematic bias into the data as well. Systematic shifts in computed flux rates can also be caused by the setup of the quality flagging system that determines which part of the data to exclude from further analysis. The chosen scheme will influence the frequency and seasonal

distribution of data gaps, and therefore also the performance of the gap-filling procedure. Since our approach aimed at minimizing gaps, while at the same time skilfully reviewing particularly low-quality fluxes through additional quality measures, we can rule out potential systematic biases linked to this methodology as well. Summarizing, systematic errors did not play a role for the uncertainty assessment of the dataset presented herein, and were therefore excluded from the analysis.

Random errors in time series of eddy-covariance fluxes are mostly linked to turbulence sampling errors, instrument errors, and uncertainties of the footprint (Rannik et al., 2016). We neglected the footprint errors in this context, since both towers are exposed to the same wind climatology. Assessment of the turbulent sampling error and instrument errors for each 30-minute flux value are a standard output of the flux processing software employed here (TK3, Mauder and Foken, 2015). These values were treated as independent errors when combined to a total error. Uncertainty assessments for the gap-filled

flux values were taken from the output of the MDS-routine (Reichstein et al., 2005) employed for this purpose. The random errors of the ensemble averaged fluxes were computed following Rannik et al. (2016)

Regarding the accuracy of radiation measurements, the Kipp & Zonen CNR4 radiation sensors employed here at both observation sites are officially classified as 'first class' instruments (for shortwave radiation e.g. a resolution of +/-5 $Wm^{-2}$, and a stability of +/-2%). However, based on the direct comparison of data from both instruments, we find cumulative

differences below 1% of the total incoming radiation, therefore our sensors would even qualify for the next highest quality rating (secondary standard, e.g. a resolution of +/-1 $Wm^{-2}$, and a stability of +/-1%). The CNR4 instruments have downward looking opening angles of 150 degrees. For sensor heights of 4.5m (drainage tower) and 4.66m (control tower), this translates into circular footprints with a radius of 16.8 m and 17.4 m, or footprint areas of 886 $m^2$ and 950 $m^2$.

Meteorological data from all other (slow) instruments were collected at 10-second intervals and stored as 10-minute

averages. The final dataset was averaged to 30-minute intervals. A separate quality assessment protocol was developed for this part of the dataset, comprising e.g. a test for failure of the power supply, checks of range and variability of time series, a flat lining test, a spike test, and finally a test for sensors malfunctioning based on plausibility limits.

All datasets presented within the context of this study were based on the same instrumentation, as well as a uniform data processing and quality assessment routine, for both drainage and control sites. Manual measurements such as vegetation

community studies, or measurement of active layer depth and water table depth, were carried out by the same persons in both study areas within each specific observation period, so that also here site intercomparison results cannot be subject to systematic errors based on subjective decisions by the observer. All datasets, including eddy-covariance fluxes, slow meteorology and manual sampling, have undergone a thorough quality assessment protocol (Kittler et al., 2016; Kwon et al., 2016), which included plausibility checks based on the intercomparison of observations between both study areas. Still, it

cannot be ruled out that parts of the absolute values presented here are subject to systematic offsets; however, since both drainage and control datasets should be affected by such systematic offsets in the same way, the differences between these datasets will not be affected, and can thus be fully attributed to drainage effects on ecosystem characteristics and energy flux rates. A comprehensive description on instrumentation and data processing procedure can be found in Kittler et al. (2016).

M G 9/21/2017 09:27
**Deleted:** e.g.

**Table 1: Relative abundance of the major vascular plant species found within both disturbance regimes in the context of a non-destructive sampling campaign in 2014. The statistics exclude mosses, dead plants and bare soil, as well as other plant species with a coverage fraction much smaller than one percent.**

| Species name | Common name | drained area [%] | control area [%] |
|---|---|---|---|
| *Betula nana subsp. exilis* | Arctic dwarf birch | 9.3 ±21.1 | 0.6 ±2.8 |
| *Calamagrostis purpurascens* | Purple reedgrass | 4.3 ±19.3 | 0.3 ±1.4 |
| *Carex appendiculata* and *lugens* | Tussock forming sedge | 49.2 ±42.8 | 35.7 ±46.5 |
| *Chamaedaphne calyculata* | Leatherleaf | 3.4 ±10.7 | 0.4 ±1.3 |
| *Eriophorum angustifolium* | Cottongrass | 18.6 ±29.0 | 43.3 ±37.7 |
| *Potentila palustris* | Marsh cinquefoil | 2.3 ±8.1 | 9.3 ±20.7 |
| *Salix fuscescens* | Alaska bog willow | 11.4 ±14.5 | 10.3 ±14.6 |
| *Salix pulchra* | Diamondleaf willow | 1.6 ±6.3 | 0.0 ±0.0 |

M G 9/27/2017 11:02
**Deleted:** (

M G 9/27/2017 11:02
**Deleted:** )

M G 9/27/2017 11:02
**Deleted:** Common c

**Table 2: Definition of sub-seasons that reflect the pattern in albedo differences between drainage and control areas. Differences for albedo and net shortwave radiation budget SW(net) were calculated as seasonal means of daily values, drainage minus control. Dates reflect the conditions in data year 2014, and may changes between years.**

| period | start date | end date | duration [days] | albedo difference [-] | SW(net) difference [Wm$^{-2}$] |
|---|---|---|---|---|---|
| closed snow cover | 2/14 | 4/27 | 73 | 2.4 | -2.30 |
| thinning of snow cover | 4/28 | 5/22 | 25 | -2.6 | 7.47 |
| snow cover disappears | 5/23 | 5/31 | 9 | -15.3 | 48.31 |
| flooding transition | 6/1 | 6/18 | 18 | 0.2 | -0.54 |
| snow free season | 6/19 | 9/28 | 102 | 1.9 | -2.18 |
| first snow patches | 9/29 | 10/12 | 14 | 3.4 | -0.98 |
| snow cover buildup | 10/13 | 10/30 | 18 | 12.1 | -2.41 |

| period | start date | end date | duration [days] | albedo difference [-] | SW(net) difference [Wm$^{-2}$] |
|---|---|---|---|---|---|
| closed snow cover | 2/14 | 4/27 | 73 | 2.4 | -2.30 |
| thinning of snow cover | 4/28 | 5/22 | 25 | -2.6 | 7.47 |
| snow cover disappears | 5/23 | 5/31 | 9 | -15.3 | 48.31 |
| flooding transition | 6/1 | 6/18 | 18 | 0.2 | -0.54 |
| snow free season | 6/19 | 9/28 | 102 | 1.9 | -2.18 |
| first snow patches | 9/29 | 10/12 | 14 | 3.4 | -0.98 |
| snow cover buildup | 10/13 | 10/30 | 18 | 12.1 | -2.41 |

**Table 3: Energy flux components averaged over the summer months (June-September) within data years 2014 and 2015, including average and relative differences between drainage and control area observations.**

| parameter | drainage | control | difference | change [%] |
|---|---|---|---|---|
| Net radiation, $R_{net}$ [$Wm^{-2}$] | 94.0 | 95.7 | -1.7 | -1.8 |
| Sensible heat flux, H [$Wm^{-2}$] | 27.9 ±0.13 | 25.5 ±0.12 | 2.4 | 9.4 |
| Latent heat flux, LE [$Wm^{-2}$] | 28.4 ±0.11 | 30.6 ±0.12 | -2.2 | -7.3 |
| Sum of H + LE [$Wm^{-2}$] | 56.2 ±0.18 | 56.1 ±0.17 | 0.2 | 0.3 |
| Bowen Ratio, BR [-] | 0.983 | 0.833 | 0.150 | 18.0 |

| parameter | drainage | control | difference | change [%] |
|---|---|---|---|---|
| Net radiation, $R_{net}$ [$Wm^{-2}$] | 94.0 | 95.7 | -1.7 | -1.8 |
| Sensible heat flux, H [$Wm^{-2}$] | 27.9 ±0.13 | 25.5 ±0.12 | 2.4 | 9.4 |
| Latent heat flux, LE [$Wm^{-2}$] | 28.4 ±0.11 | 30.6 ±0.12 | -2.2 | -7.3 |
| Sum of H + LE [$Wm^{-2}$] | 56.2 ±0.18 | 56.1 ±0.17 | 0.2 | 0.3 |
| Bowen Ratio, BR [-] | 0.983 | 0.833 | 0.150 | 18.0 |

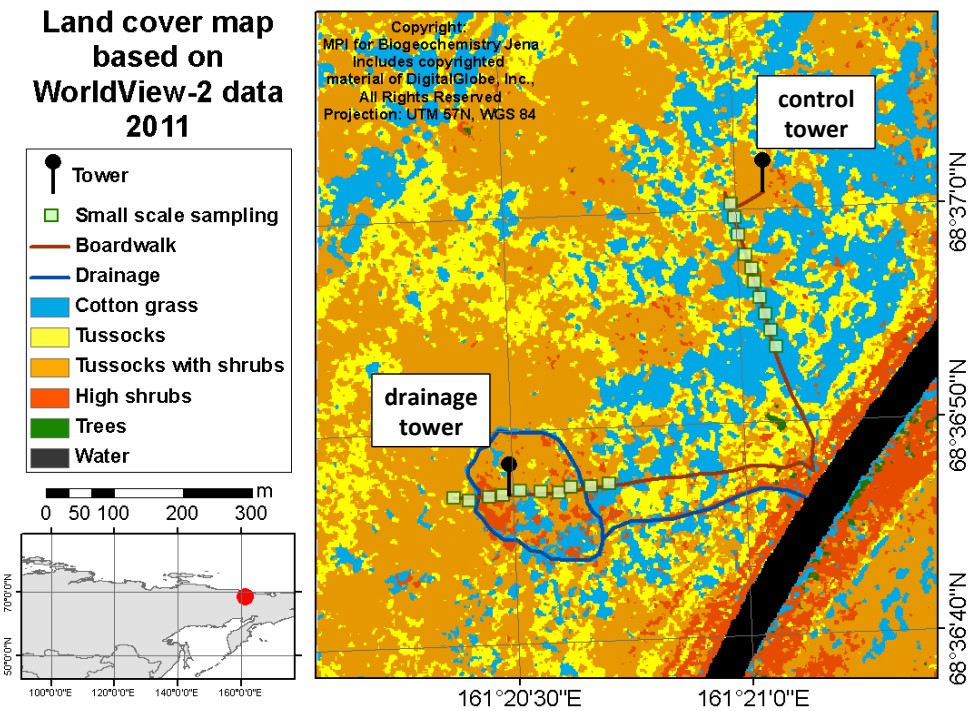

**Figure 1: Land cover structure and instrumentation setup at the Chersky observation site.**

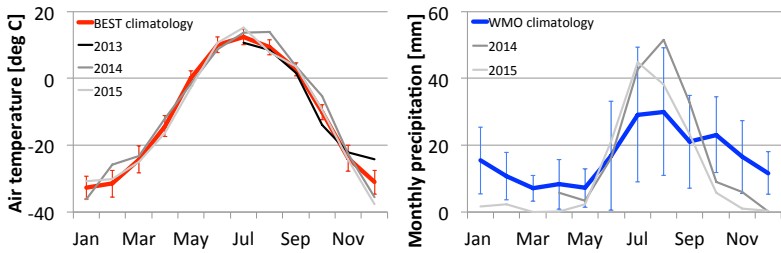

**Figure 2:** Left panel: Average annual course (1960-2009) and recent observations of monthly mean air temperature; right panel: Long-term average (1950-1999) and recent observations of monthly precipitation sums for the Chersky region.

M G 9/22/2017 13:55
**Deleted:** surface

M G 9/22/2017 13:55
**Deleted:** (Tair, 1960-2009)

M G 9/22/2017 13:55
**Deleted:** (PRCP, 1950-1999)

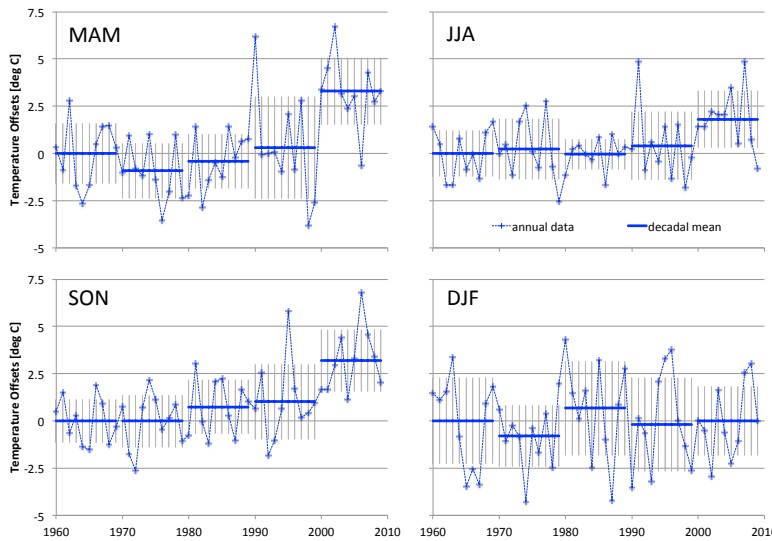

**Figure 3: Temperature trends for the Chersky station, taken from the BEST database. Top left: springtime warming period; top right: core summer period; bottom left: fall freeze-up period; bottom right: core winter period. All temperatures were normalized against the mean seasonal temperatures from the 1960-69 decade. Blue crosses and dashed blue lines give annual data, thick blue lines the decadal mean with vertical grey bars showing decadal RMSE.**

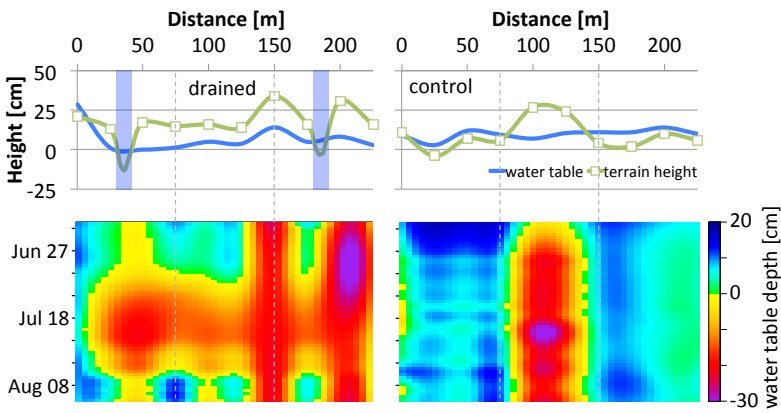

**Figure 4: Top panels:** Variability of terrain height (green lines, boxes indicating microsite study positions) and water table depth (blue lines) along transects in the drained (left) and control (right) sections of the observation site (water table depth reflects conditions in mid July 2014). Blue shading in drained panel indicates position of drainage channel, where interpolation of terrain height was modified. The zero level was chosen arbitrarily, so absolute heights do not carry any information in the top panels; bottom panels: Development of water table depth over time for the period Jun 18 to Aug 14, 2014 (interpolated over both space and time). Values are given relative to the surface, with negative values indicating water tables below the surface level.

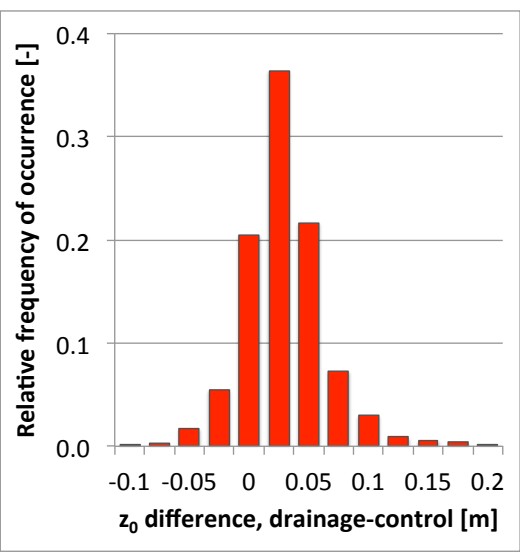

**Figure 5: Frequency distribution of the shifts in aerodynamic roughness length $z_0$ (drained-control) in the drained area, compared to the control area. Data covers the months June through September in 2014/15.**

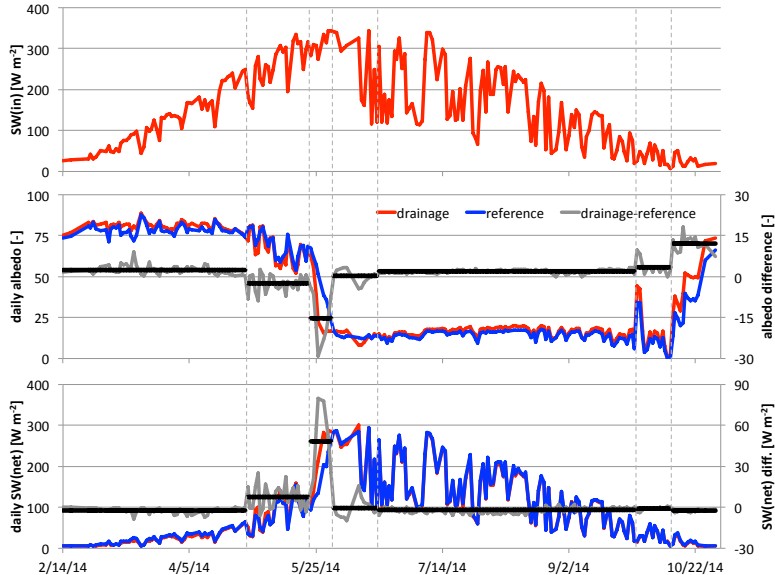

**Figure 6:** (top) 2014 annual course of the daily mean incoming shortwave radiation; (center) daily mean albedo derived from the ratio of downward and upward shortwave radiation. The grey line indicates the daily difference between drained and control towers (scales on the right), with the black horizontal bars giving the mean values for seven sub-seasons (separated by vertical dashed lines, see Table 2 for definitions); (bottom) same structure as used in the center panel, here to show the net shortwave radiation budget from both towers and their seasonal differences. All plots have been restricted to the period where SW(in) > 5 Wm$^{-2}$.

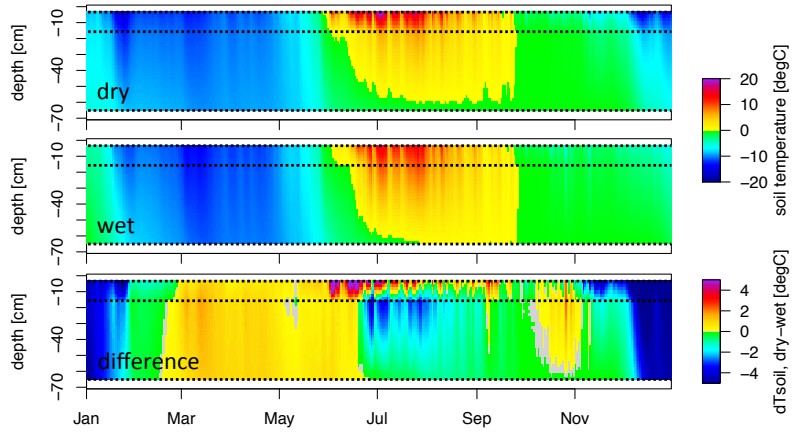

**Figure 7**: **Daily mean soil temperature profiles at a dry (upper panel) and wet (center panel) microsite within the drained transect, and the difference (dry-wet, bottom panel) for the calendar year 2015. Data for three measurement depths (dotted lines) are linearly interpolated along the vertical profile. Values in the difference plot were truncated to the range [-5; 5] °C to enhance visibility of fine-scale patterns, while values fell within the range [-11.3; 10.5] °C.**

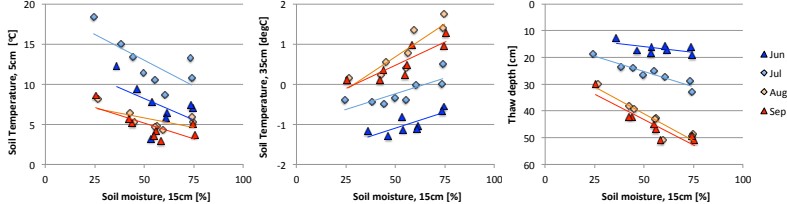

**Figure 8: Influence of soil moisture on soil temperatures and thaw depth at eight microsites covering both disturbance treatments. Observations from summer 2015 are aggregated by site over a period of about two weeks each (Jun 13-30; Jul 01-13; Aug 21-31; Sep 01-12). Lines represent linear regression fits to emphasize trends.**

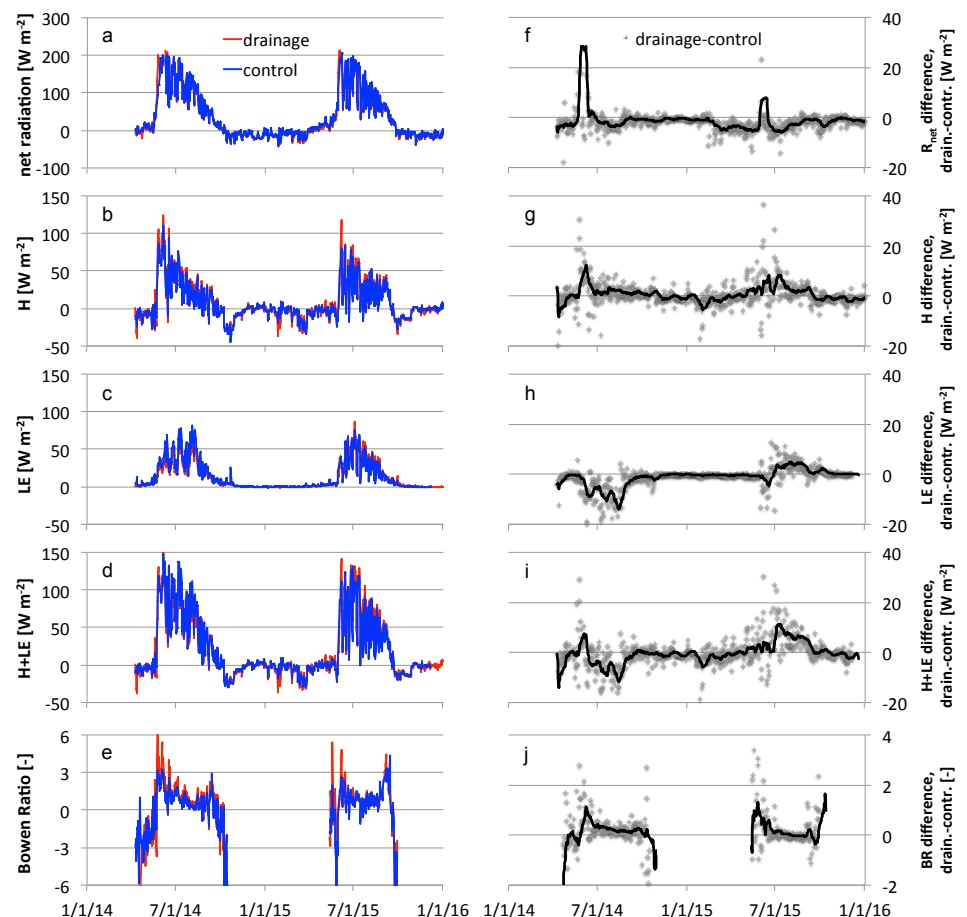

**Figure 9. (a-e)** Summary of daily averaged energy fluxes within drainage (red line) and control (blue line) areas. (f-j) Daily differences (drainage minus control, grey crosses) between the two treatments, with black lines giving the average differences for a 15-day moving window. (a, f) net radiation; (b, g) sensible heat fluxes; (c, h) latent heat fluxes; (d, i) sum of sensible and latent heat fluxes; (e, j) Bowen ratio, which is the ratio of sensible to latent heat fluxes. Vertical scales of difference plots have been truncated to enhance display of fine-scale patterns in overall trends.

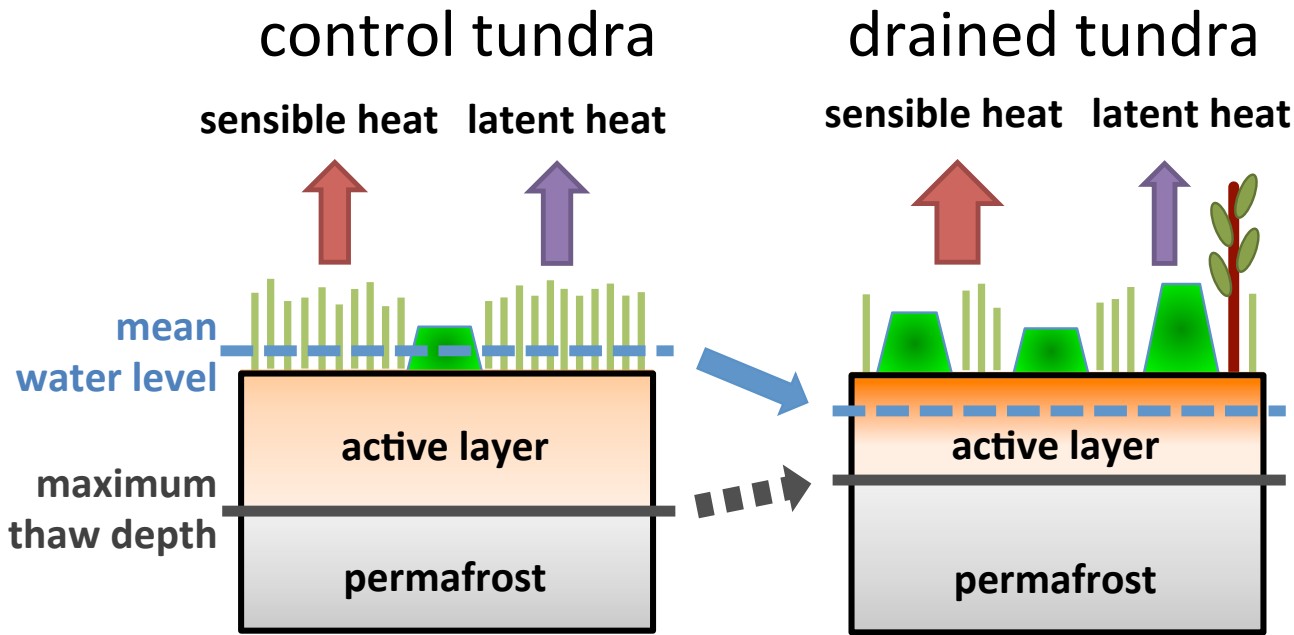

**Figure 10:** Expected changes in growing season properties of a tundra ecosystem as response to drainage disturbance.