# Peer review of "Shifts in energy fluxes linked to drainage-induced changes in permafrost ecosystem structure increase Bowen Ratios, but reduce thaw depth"

_The Cryosphere, 2016_

## Referee Comment (RC1) · Anonymous Referee #1 · 14 Dec 2016

Summary: The study "Shifts in permafrost ecosystem structure following a decade-long drainage increase energy transfer to the atmosphere, but reduce thaw depth" by Göckede et al. investigates the impact of a drainage experiment within a wet tundra landscape on components of the surface energy balance. The study seeks to quantify differences in micro climate between a drainage site and a control site which show marked differences in vegetation cover and soil moisture. The authors, therefore, compare sensible and latent heat fluxes which were measured simultaneously at both sites by eddy covariance. In addition, differences in radiation, near surface soil temperatures, and active layer depths are presented. The found differences be-

tween the drained site and the control site are discussed in the context of differences in vegetation cover and soil moisture. The study reports increased sensible heat fluxes at the drained site, whereas much small differences are reported for the latent heat flux. Only minor differences are also reported for the net radiation. The authors argue that the increased sensible heat flux is related to a decreased ground heat flux which is reflected in decreased soil temperatures and lower active layer depths. The study focuses on a very important issue in permafrost research that addresses possible hydrological changes in the Arctic due to climate warming and permafrost degradation. This study is, therefore, within the scientific scope of the journal The Cryosphere. The study is written in a clear and understandable manner and is well structured. There are, however, some major issues that should be addressed before publication.

General comments: The measured differences in the turbulent heat fluxes and the radiation are essential to the argumentation of this study since the ground heat flux is not determined independently. Thus, it would be highly recommended to perform a comprehensive uncertainty analysis and quality assessment of the fluxes. Uncertainty ranges should be calculated in order to evaluate whether the measured differences are significant. It must be excluded that the differences in the heat fluxes are subject to instrument biases and/or site related methodological biases. Due to the importance of reliable measurements, it seems to be inadequate to just refer to an other study for quality assessment.

The authors state that they do not see any differences in the long wave radiation budget. This is surprising since increased sensible heat fluxes are often related to increased surface temperatures. I think a discussion of this point would be highly interesting for understanding the reasons for the increased sensible heat fluxes.

The authors present differences in friction velocity as indicator for differences in vegetation structure (hight/density). In this context, I think it would be more instructive to calculate roughness lengths during neutral conditions. In addition, a footprint analysis might be useful since vegetation seems to vary within the footprint area of the eddy

covariance tower (Figure 1). It might be interesting for the process understanding to further investigate the impact of changed surface roughness on flux partition.

In general, the data analysis presented in this study is very limited and should be expanded. Besides footprint analysis, the analysis of the diurnal and seasonal signals can provide deeper insights into the processes behind flux partition. Figure 9b, for example, clearly demonstrates that more than just average differences can be observed in the time series of sensible heat fluxes. I think that the very strong and general statements made in the conclusions and the abstract are not possible based on the very limited data analysis presented here and, thus, should be toned down. Without further process analysis leading to a better understanding of heat transfer processes within and below shrubs, the presented data and results must be considered site specific and are not directly transferable to other tundra landscapes.

The authors do not present any data or results related to snow. However, several paragraphs in the discussion include speculations on snow including the impact of vegetation on snow, and the impact of ground heat fluxes on snow melt. As these subjects are essentially not part of the presented investigations, I strongly recommend to exclude them from the discussion and to focus on the presented results.

Specific comments: p.8,l.1: Sections 0 and 0?

p.8, l.2: Why not including the LW budget in the analysis, in particular the outgoing LW radiation?

p.8, l.6: What is the absolute accuracy of the used radiation sensor? What is the footprint of the radiation sensor? Does the sensor measure a representative area of the drained site?

p.9, l.1: Here and throughout the manuscript it should be thermal conductivity.

p.9,l.3-5: This is a very general statement which does not provide any quantitative information on the heat transfer processes within the soil. Please note that increased

soil moisture means a higher content of latent heat which besides thermal conductivity and heat capacity determines the duration of the zero curtain.

p.9,l.25-27: Besides the ground heat flux, the thaw depth is determined by the soil ice content. Are there any information on differences in soil composition between both sites?

p.10,l.6: This is a very small number (1,8%), is this within the accuracy of the sensor?

p.10,l.13: Is it possible to distinguish between the impact of drainage and changed vegetation cover within this study? Or is it more likely that the vegetation cover has adapted to drainage which then has modified the surface energy balance? I think it would be important to separate these things since there drainage can happen on relatively short time scales while changes in vegetation cover requires some time.

p.11,l.13: Does 'slightly lower' mean a more negative sensible heat flux at the drained site?

p.12,l.6-15: See general comment on snow cover discussion.

p.12,l.23-30: Why so much emphasis on discussing the impact of mosses if mosses are absent at the study site?

---

## Referee Comment (RC2) · Anonymous Referee #2 · 4 Jan 2017

This study describes the change in surface energy balance associated with a 10 year manipulation of water table depth in an arctic wetland. They show that vegetation community shifts to greater shrub cover results in altered surface energy balance in terms of albedo and the partitioning of latent and sensible heat flux. The authors also show that the reduced heat conductivity in dry organic soils can create shallower thaw depths and stabilize the permafrost. This study is very interesting, and contributes to our understanding of how shrub cover can impact surface energy balance, but also how changes in the water table depth can promote shrub establishment. This can certainly support other studies that examine the mechanisms of shrub establishment in the arctic. This study also contributes more to our understanding of how prolonged dry soils can impact the surface energy balance through changes in soil heat flux. Thus, this study should be published in this journal.

The only issue I have with this paper is the writing. I recognize that English may not be the first language of some of the authors, so I have made many suggestions to improve the text.

Throughout the text, remove all the "e.g." or "i.e." (like on Page 2, Line 22 and many other locations in the text) because they are being used incorrectly.  Whenever possible, I include suggestions for what to use instead.
Line 17: change "has be accumulated" to "has accumulated"
Line 20: change "hydrology can vary at smallest spatial.." to "hydrology can vary at small spatial…"
Line 28: Change to "so that drier conditions can also be expected over large areas."
Line 33: change "extend" to "extent"
Line 5: change to "only a few"
Line 9: change to "energy exchange patterns based on Bowen Ratio results"
Line 13: Change "use" to "used"
Line 16: change "In a second step," to "Our second objective was"
Line 19: change to "mostly in the form…"
Line 4: change to "(Section 3.1), …"
Line 7: change to "in the fall"
Line 10: remove "thus"
Line 25: remove "i.e."
Line 30: what do you mean "irregular" time intervals? Can you provide some more information?
Line 12: change to "for the period of 1950…"
Line 13 - 16: unclear sentence, change to "No precipitation records were available for the Chersky station from 2000-2009. Thus, we analyzed long-term trends from Ambarchik station situated 100km north of Chersky with nearly identical mean annual precipitation sums."

Line 26: change to "sections, and follow.."
Line 27: change to "tend to be positive, …"
Line 30: change to "months, .."

Line 3: change to "Total precipitation in 2015 was ~71% of the long-term trend (154 mm)"
Line 28: remove "also" and place it after "sums"
Line 31: remove "also"
Line 32: remove comma after "microrelief"
Line 12: change "input" to "inputs"
Line 17: change to "with tussock forming sedges () as the second most abundant species."
Line 22: "birch" not "birches"
Line 23: Rather than "close to zero" can you provide actual percentages?
Line 23: use the % symbol rather than "percent"
Line 27: change to "same size near the control tower."
Line 1: there is a mistake with the Sections; it's shown as "Sections 0 and 0"
Line 5: change to "allow us…"
Line 5-6: I don't understand this statement: allow to structure the 'light season' into seven distinct sections
Line 9: Change to "that is decreased by.."
Line 8-10: This is a run-on sentence. Please split it up in a logical way.
Line 19: remove comma after "areas"
Line 31: change to "(… Table 3). The winter results are presented after the summer results in the following section."
Line 21: change "in contrast to that" to "conversely, during the…"
Line 26: add "the" after "from"
Line 26: remove "the" after "of" so it reads "after drainage"
Line 30: change "in contrast to that" to "conversely, in 2015…"
Line 5: I don't know what this means "Smoothed offsets between drainage and control are positive throughout the summer months"
Line 9 - 12: try something like this to increase clarity: "Shortwave energy input is extremely low during the polar winter and outgoing longwave radiation usually exceeds incoming radiation; values are slightly lower (-1.6 Wm-2) in the drainage area."
Line 9: change to "…regime for a couple of reasons. The first is…. The second is….."
Line 12: change "contribute" to "contributing"
Line 13: change to "during the summer"
Line 19: change to "during the summer"
Line 5: change to "..domains. Onshore winds…"
Line 7: comma after "also"
Line 10: change dampened to slight
Line 12: remove the "ie" and change the text after it to " , wherein the overall reductions in the flux rates were of equal magnitude, and "
Line 15: change "in accordance with the results presented by Boike et al. (2008) our results indicate" to ", similar to Boike et al. (), we found that..."

Line 20: put a comma after "flux"
Line 21: change derived to observed
Line 22: change to "areas, wherein compared to 2014, in 2015...."
Line 28: change to "interannual variability. In particular, the slightly warmer.."
Line 31: change to "did not have a strong effect…"
Line 21: Change to "in the current study"
Line 31: change to "and also alter thaw depth and the net ecosystem – atmosphere heat exchange"
Line 30: please clarify …"unclear the level of detail about?? these models need to accurately represent what ??"
Line 8: change to "disturbance (the lowering of the water table) impacts…"
Line 17: change to ".. and multiple links with factors such as snow cover…."
Line 21: change to "details like shrub density, which complicates the evaluation of this ....."
Line 22: change to "Further, including a drainage effect increases the complexity. For example, during the winter....."
Line 32: change to "ecosystems. However, …"
Line 5: change to "non-linearly transitions from one state to another"
Line 7: I don't know what a "superposition" is
Line 9 - 11: change to "While the primary disturbance immediately affected the water table, other key components like the vegetation community structure take longer to change."
Line 11: this sentence is confusing. It sounds like you're trying to say that until the changes in snow cover have occurred, the full effect of the manipulation cannot be observed. Please clarify.
Line 15: change to "and further shifts in factors like shrub coverage and canopy height…"
Line 24: change to "over the summertime cooling and impact thaw depth cannot be..."
Line 14: change to "changing the disturbance.."
Line 18: change to "conditions (e.g., the heat flows…..) may increase under.."
Paragraph 4.3 is not needed.
Line 12: change to "during the summer"
Line 15: Change to "Since dry organic soils are efficient insulators, drainage can reduce thaw depth, and can initially protect ..... . However, the net......"

---

## Author Comment (AC1) · 21 Feb 2017

*Note: in the following document, the original comments made by the reviewer are copied in black, while the authors' responses to these comments follow in blue.*

**Authors' response to comments submitted by Reviewer #1**

Summary: The study "Shifts in permafrost ecosystem structure following a decade-long drainage increase energy transfer to the atmosphere, but reduce thaw depth" by Göckede et al. investigates the impact of a drainage experiment within a wet tundra landscape on components of the surface energy balance. The study seeks to quantify differences in microclimate between a drainage site and a control site which show marked differences in vegetation cover and soil moisture. The authors, therefore, compare sensible and latent heat fluxes which were measured simultaneously at both sites by eddy covariance. In addition, differences in radiation, near surface soil temperatures, and active layer depths are presented. The found differences between the drained site and the control site are discussed in the context of differences in vegetation cover and soil moisture. The study reports increased sensible heat fluxes at the drained site, whereas much small differences are reported for the latent heat flux. Only minor differences are also reported for the net radiation. The authors argue that the increased sensible heat flux is related to a decreased ground heat flux which is reflected in decreased soil temperatures and lower active layer depths. The study focuses on a very important issue in permafrost research that addresses possible hydrological changes in the Arctic due to climate warming and permafrost degradation. This study is, therefore, within the scientific scope of the journal The Cryosphere. The study is written in a clear and understandable manner and is well structured. There are, however, some major issues that should be addressed before publication.

The authors thank the reviewer for her/his overall supportive evaluation of the findings presented in this manuscript. We would like to remark at this point that due to a minor adjustment in the quality assessment protocol for the eddy-covariance datasets, the numbers of the vertical turbulent heat fluxes presented herein have slightly changed since we submitted the first version of the manuscript to The Cryosphere. The overall trends remain the same, i.e. we find a shift of the energy partitioning from latent to sensible heat fluxes, leading to a significant increase in Bowen Ratios following drainage. With the new numbers, however, the absolute increase in sensible heat H has been reduced, while the latent heat LE is actually slightly reduced, leading to a budget H+LE that is about the same at drainage and control sites, respectively.

General comments: The measured differences in the turbulent heat fluxes and the radiation are essential to the argumentation of this study since the ground heat flux is not determined independently. Thus, it would be highly recommended to perform a comprehensive uncertainty analysis and quality assessment of the fluxes. Uncertainty ranges should be calculated in order to evaluate whether the measured differences are significant.

In the revised version of the manuscript, we will add a comprehensive uncertainty assessment of our flux datasets, as requested by the reviewer.

It must be excluded that the differences in the heat fluxes are subject to instrument biases and/or site related methodological biases.

We used exactly the same instrumentation at both sites. Moreover, we also applied the same data processing protocol for datasets from both towers, and the quality assessment protocol includes a direct comparison of data elements from both sites. Therefore, we believe that systematic methodological bias can be excluded here. We will extend the documentation of these checks in the revised manuscript version (see below).

Due to the importance of reliable measurements, it seems to be inadequate to just refer to another study for quality assessment.

We will add a new appendix in the revised version of the manuscript that will present the core elements of our eddy-covariance data processing protocol, which will include the details of

the quality assessment protocol applied for the flux and meteorological data used within the context of this study.

The authors state that they do not see any differences in the long wave radiation budget. This is surprising since increased sensible heat fluxes are often related to increased surface temperatures. I think a discussion of this point would be highly interesting for understanding the reasons for the increased sensible heat fluxes.
We decided to leave out details on this part of the study because we believe the longwave radiation results do not provide enough additional insight to justify the required extra text passages. 'Virtually unchanged' refers to a shift in longwave radiation budgets of -0.2% (summers 2014/15) between drainage and control sites. However, there is considerable interannual variability, with higher deficits ($LW_{down}$-$LW_{up}$) in 2014 at both sites. Since this interannual variability turned out to be more pronounced at the control site, compared to the drainage site, we measured offsets of opposite signs in both data years, which largely cancelled each other. This pattern in interannual variability agrees well with the patterns in both sensible and latent heat fluxes observed at the two study sites. We will include links to longwave radiation patterns for the revised discussion of sensible and latent heat fluxes in the new paper version

The authors present differences in friction velocity as indicator for differences in vegetation structure (height/density). In this context, I think it would be more instructive to calculate roughness lengths during neutral conditions. In addition, a footprint analysis might be useful since vegetation seems to vary within the footprint area of the eddy covariance tower (Figure 1). It might be interesting for the process understanding to further investigate the impact of changed surface roughness on flux partition.
The author preferred to make this point based on the friction velocity since it provides a broader and thus more representative data basis (no filtering for stability of stratification). However, the same can of course be done based on the roughness length, derived from wind profile relationships during neutral stratification, as suggested by the reviewer. Also in this case, we find a systematic enhancement of roughness lengths (mean increase: 0.035m), with aerodynamically rougher surfaces in all wind sectors for the drainage site, compared to the control. Since this is rather a side aspect of the analysis presented within the context of this manuscript, we do not think a proper footprint analysis will be required to make this point, also because we lack spatially distributed information on vegetation height (a highest resolution (~2m) WorldView land cover map would be available). We will include information on differences in roughness length between both sites broken up by wind sectors to include a spatial context of the differences observed.

In general, the data analysis presented in this study is very limited and should be expanded. Besides footprint analysis, the analysis of the diurnal and seasonal signals can provide deeper insights into the processes behind flux partition. Figure 9b, for example, clearly demonstrates that more than just average differences can be observed in the time series of sensible heat fluxes. I think that the very strong and general statements made in the conclusions and the abstract are not possible based on the very limited data analysis presented here and, thus, should be toned down. Without further process analysis leading to a better understanding of heat transfer processes within and below shrubs, the presented data and results must be considered site specific and are not directly transferable to other tundra landscapes.
As mentioned in the comment above, the authors believe that a footprint analysis in this context will not provide further insights that help interpreting the differences in energy flux signals. However, a directional effect will be added. The general seasonal trends in fluxes and their differences between the two sites have been covered in Section 3.6 already, but we will refine this description in the revised version of the manuscript. This description could in theory be extended by also comparing typical diurnal courses between the treatments, but since the meteorological forcing is virtually the same at both of them, and more influential factors such as e.g. water levels, vegetation structure and soil thermal regimes vary at timescales much longer than diurnal, the authors believe that analyses on such short timescales will not

strengthen the study. We are certainly aware that we can only provide results for a single site, and the derivation of process understanding that can be transferred also to other Arctic sites will be associated with uncertainties. As suggested by the reviewer, we will therefore tone down all statements that may hint at a general applicability of our findings.

The authors do not present any data or results related to snow. However, several paragraphs in the discussion include speculations on snow including the impact of vegetation on snow, and the impact of ground heat fluxes on snow melt. As these subjects are essentially not part of the presented investigations, I strongly recommend to exclude them from the discussion and to focus on the presented results.

We discuss differences in the snow cover periods between the two study sites in Section 3.4, which are reflected in the radiation data shown in Figure 6, as well as in the numbers on albedo and SWnet summarized in Table 2. These results emphasize that snow cover dynamics are influenced by drainage disturbance, and that shifts in snow cover may cause systematic secondary effects on both the radiative budget and the soil thermal regime. We do not measure any snow properties directly at both sites in parallel, snow depth is only monitored at a single location currently. We therefore agree with the reviewer that we do not present any direct observational evidence on snow characteristics herein. Still, we have the indirect evidence on systematic shifts in the snow cover period obtained through the albedo record. The passages in the discussion that the reviewer refers to aim at pointing out mechanisms, mostly links between vegetation and snow cover, that can explain these patterns in albedo. Therefore, we believe that these discussion sections are required to interpret the complex interactions between secondary disturbance effects.

Specific comments:
p.8,l.1: Sections 0 and 0?
This was a typo based on a broken automated cross-reference, and has been updated.

p.8, l.2: Why not including the LW budget in the analysis, in particular the outgoing LW radiation?
As mentioned in more detail above, there is virtually no difference to be seen in the net LW radiation budget, so the authors believe that including more details here would provide only very little information.

p.8, l.6: What is the absolute accuracy of the used radiation sensor? What is the footprint of the radiation sensor? Does the sensor measure a representative area of the drained site?
We use Kipp & Zonen CNR4 as radiation sensors at both observation sites. These instruments are officially classified as 'first class' instruments (for shortwave radiation e.g. a resolution of +/-5 $Wm^{-2}$, and a stability of +/-2%). However, based on the direct comparison of data from both instruments, we find cumulative differences below 1% of the total incoming radiation, therefore our sensors would even qualify for the next highest quality rating (secondary standard, e.g. a resolution of +/-1 $Wm^{-2}$, and a stability of +/-1%).
The CNR4 instruments have downward looking opening angles of 150 degrees. For sensor heights of 4.5m (drainage tower) and 4.66m (control tower), this translates into circular footprints with a radius of ~16.8m and 17.4m, or footprint areas of 886$m^2$ and 950$m^2$. We believe that the area covered for the drained site is indeed representative for the drainage area overall, but of course it is obvious that we can only cover a small fraction of the total area.

p.9, l.1: Here and throughout the manuscript it should be thermal conductivity.
This has been corrected throughout the manuscript.

p.9,l.3-5: This is a very general statement which does not provide any quantitative information on the heat transfer processes within the soil. Please note that increased soil moisture means a higher content of latent heat which besides thermal conductivity and heat capacity determines the duration of the zero curtain.

We obviously chose a misleading term here, since with 'heat capacity linked to higher water content' we actually wanted to refer to the latent heat. This will be corrected in the revised version of the manuscript.

p.9,l.25-27: Besides the ground heat flux, the thaw depth is determined by the soil ice content. Are there any information on differences in soil composition between both sites?
We took soil profiles across the treatment areas during the growing season, and analyzed for e.g. carbon content and nutrients. Regarding ice content, we also have a limited number of cores taken in November, i.e. with frozen ground, where relative ice content was assessed for wet and dry microsites in both drainage and control areas. As a general trend, we found higher ice content in the wet microsites within the top ~30cm of the soil profile, while relative ice content was higher in the dry microsites between ~30-50cm below the surface. Further down, no systematic differences were observed.

p.10,l.6: This is a very small number (1,8%), is this within the accuracy of the sensor?
Please refer to the comments on p.8, l.6 above. The relative deviations in incoming radiation between both sensors were found to be <1% of the total signal. Therefore, the deviations between net energy input described in this section are higher then the accuracy level of the employed sensors.

p.10,l.13: Is it possible to distinguish between the impact of drainage and changed vegetation cover within this study? Or is it more likely that the vegetation cover has adapted to drainage which then has modified the surface energy balance? I think it would be important to separate these things since there drainage can happen on relatively short time scales while changes in vegetation cover requires some time.
Based on the data that we have available at this time, we cannot cleanly separate between a direct impact of water level changes (drainage) and the associated effect of changes in vegetation structure as a secondary disturbance. To be able to do so, we would need more data years with differences in water levels, so that we could do proper statistics on the impact of these year-to-year changes, while vegetation stays more or less constant. The importance of covering the long-term effect of vegetation changes has been included in the discussion section.

p.11,l.13: Does 'slightly lower' mean a more negative sensible heat flux at the drained site?
As mentioned in the text, the mean sensible heat flux for both towers during wintertime is -3.4 $Wm^{-2}$. Here, the drainage tower has a mean value of -4.3 $Wm^{-2}$, and the control tower a mean of -2.5 $Wm^{-2}$. Accordingly, 'slightly lower' referred to a more negative sensible heat flux at the drainage site. This will be clarified in the revised version of the text. Due to the revision of quality flagging as mentioned in the first comment above, these values have slightly changed (drainage: -5.6 $Wm^{-2}$; control: -4.4 $Wm^{-2}$; difference: -1.2 $Wm^{-2}$), but the overall trend remains the same.

p.12,l.6-15: See general comment on snow cover discussion.
As mentioned above, we believe that this interpretation, fortified by the references given in this paragraph, is necessary to provide an interpretation why we see a consistently earlier buildup of a snow cover at the drained site, compared to the control (Figure 6, Table 2).

p.12,l.23-30: Why so much emphasis on discussing the impact of mosses if mosses are absent at the study site?
We decided to include this paragraph to point out a feature of the site characteristics that is different from many other places in the Arctic. As mentioned in the text, we believe that a higher abundance of mosses would significantly alter the findings we observed at our sites. Accordingly, our results may not be representative for sites at other locations that feature a higher moss coverage.

---

## Author Comment (AC2) · 21 Feb 2017

The authors would like to thank the reviewer for her/his supportive comments, and for the careful language editing throughout the text. In short, we accept all suggested changes to wording/phrasing, and will work these into the revised version of the manuscript. For those few occasions listed where a sentence was found to be confusing, or the reviewer pointed out that the meaning needed to be clarified, we will change the sentences to avoid ambiguity.

---

## Author Response (AR2)

*Note: in the following document, the original comments made by the reviewer are copied in black, while the authors' responses to these comments follow in blue.*

**Authors' response to comments submitted by Reviewer #1**

The authors have responded to the comments made in the previous review round, partly modified the manuscript, and added a paragraph about measurement uncertainties as appendix. Despite some improvements, I still see points that should be addressed before publication:

The authors state that the performed quality assessment has slightly changed their results. In total, the difference between the previous and the revised version is on the order of about 4 Watt per square meter which is indeed a minor shift. The authors, however, aim to evaluate flux differences on the order of a few Watt per square meter so that minor shifts might be relevant for some interpretation. For example, the authors have clearly stated in the previous version that atmospheric fluxes are increased at the drained site. This finding was the basis for speculations on positive feedback mechanisms towards permafrost degradation. The new results have significantly changed this point of the study.

Even though the trend in the changes of the Bowen-Ratio remains the same, I think it is necessary to address the sensitivity of the results to the performed quality assessment in the manuscript.
Starting the experiment that is presented herein in summer 2013, it took our research group numerous iterations to finally arrive at an eddy-covariance data processing procedure that we were fully confident with. The last iterations, which also affected this manuscript, were related to developing an optimum balance between data quality filtering and the subsequent gap-filling procedure. In other words, we needed to adjust quality measures that we added to the regular flagging procedure by Foken et al. The resulting optimum solution for our experiment now reliably detects biased fluxes, while producing a lower frequency of gaps with an adequate distribution across the seasons.

The shift in flux budget results between the first and second draft of this manuscript was caused by our choice to revise quality flag selection and gap-filling according to the description above. The resulting changes in net energy flux results demonstrate that the eddy-covariance method overall is quite susceptible to data processing choices by the user, and that a skillful selection and fine-tuning of methods is a prerequisite for producing reliable results. However, we believe that we by now found the optimum approach for our sites, and that shifts in net fluxes as experienced between the previous two manuscript versions will not occur again. Still, we decided to add a reference to the potential influence of this procedure in the appendix of this manuscript.

Furthermore, the title of the manuscript should be changed since the atmospheric heat fluxes are obviously not increased only the flux partition is moderately changed.
The authors acknowledge having missed to adjust the manuscript title when working in the updated eddy-covariance results into the revised version of the manuscript. We therefore highly appreciate that the reviewer pointed this out here, since of course we agree that the old title does not reflect the core message of the study anymore. Our modified title now reads "Shifts in energy fluxes linked to drainage-induced changes in permafrost ecosystem structure increase Bowen Ratios, but reduce thaw depth"

p.1, l.22: The conclusion that an increased sensible heat flux might lead to a positive feedback on permafrost degradation is very arbitrary and after correcting the results I do not see any indication in this study that supports this statement. A very similar argument could be used in order to point out an increased latent heat flux as reason for a positive feedback on permafrost

degradation. Water vapor strongly changes the radiation balance and is a very potent GHG. I suggest to focus on the results presented in this study as already outlined in the previous review round.

Our statement referred to potential effects of shifts in the energy flux partitioning on the local to regional scale lower atmospheric boundary layer. Energy in form of latent heat does not affect local temperatures directly, but instead is often vertically removed through boundary layer transport processes. Sensible heat fluxes, on the other hand, can directly influence local temperatures by heating up the lower boundary layers. Accordingly, a shift towards higher Bowen Ratios following drainage disturbance at least holds the potential to create warmer conditions locally.

Still, we acknowledge that a direct link between local temperature shifts caused by increased Bowen Ratios on the one hand and deeper thaw depths and permafrost degradation on the other remains speculative. We therefore edited the last sub-sentence of the abstract to "..,which may trigger a warming of the lower atmospheric surface layer."

p.3, l.18: Same comment as above. Besides that, the statement is misleading as it reads like that the net heat transfer into the atmosphere is increased. Following the new results this is obviously not true.

We decided to change this last paragraph of the introduction, also based on comments made by reviewer #2. As a result, the statement that reviewer #1 cites here has been removed.

p.10, l.31: The fact that the latent heat flux is not consistently decreased at the drained site in both years indicates that changes in the energy partition depend on various factors. Even though the Bowen-Ratio is consistently higher at the drained site, there is a high interannual variability as stated by the authors themselves. Thus, general conclusions based on observations of two years should be made very carefully if differences in energy partition between the sites also strongly depend on synoptic conditions.

We fully agree with the reviewer on this issue. To emphasize the uncertain long-term effect, we added the following sentence to the end of this paragraph: "However, due to the pronounced interannual variability this mean value may not be representative over longer time periods, and more data years would be required to constrain a net drainage impact.".

p.12, l.11: I think this statement is misleading. Soil hydrology is the only factor modified in this study. How is it possible to identify it as the dominate control factor without testing other cofactors such as atmospheric conditions? It is important to be precises with such statements.

We agree with the reviewer that our analyses are not comprehensive enough to support the identification of 'the dominant' control factor. Even though we could rule out strong impacts by some factors (atmospheric conditions, for example, were virtually the same for both treatment areas), we therefore changed the last sentence to "..the impact of soil hydrology was identified as a major control, ..".

p.15, l.2: Please present an estimate of how much snow could be melted earlier due to differences in soil temperature. The statement that an increased snow depth leads to increased soil temperatures and, thus, to earlier snow melt requires a sound basis.

Based on suggestions by reviewer #2, we changed our interpretation of the mechanisms leading to warmer soil temperatures at the drained site towards the end of winter. In the revised version, we now argue that the high thermal conductivity in ice-filled pores (more abundant in the control section), compared to a low conductivity in air-filled pores (more abundant in the drained section) lead to steeper soil temperature gradients with time, once all the water in the control section has been frozen (latent heat effect). Accordingly, this difference in net thermal conductivity also in winter should be the main controlling factor on trajectories in wintertime soil temperatures, with better insulation through deeper snow cover at the drained site potentially contributing to the process.

We have observational evidence from several years of continuous measurements that the drained soils have warmer (less negative) soil temperatures at the end of winter. We can only

speculate on the mechanisms behind this observation, but regard the thermal conductivity of ice-rich soils as a plausible interpretation. In any case, Eugster et al. (2000) demonstrated that underlying soil temperatures may have an influence on snow melt dynamics. We changed the last sentence in this paragraph to clarify these relationships. A quantitative estimate on the temporal shifts in snow melt linked to this effect at our site, however, is clearly beyond the scope of this study, also because we lack more specific information such as density and water content of snow.

p.15, l.12: This estimate assumes the closure term to be constant. It might be possible that the two sites feature different closure terms and that the closure terms change with time.
We fully agree with the reviewer that the 20% as an estimate of non-closure of the energy balance is only a rough estimate, and that the value may vary between years and between sites. It is clearly stated in this sentence that under the 'assumption' of a 20% non-closure, the residual would be changed as provided in the manuscript. To emphasize that the non-closure is variable, we changed the fixed value of 20% to a range of 15-25%, and adjusted the remaining residuals accordingly.

p.15, l.27: Strictly spoken, the results (thaw depth and uppermost soil temperatures) show indication of a reduced soil heat flux.
The sentence was changed accordingly.

p.16, l.29: After the quality assessment the total turbulent fluxes are not observed to be higher.
We changed this to 'higher mechanically generated turbulence'.

p.17, l.24: I do not see any result in this study justifying speculations on "tipping points". The same is true for speculations on atmosphere-permafrost feedback processes as already remarked above. The authors present nice measurements that demonstrate that the Bowen-Ratio changes due to drainage. Furthermore, they demonstrate that drainage has impacts on thaw depth and the soil thermal regime (at least within the upper decimeters). I strongly recommend that the discussion focuses on the solid results of this study without pushing speculations too far. Without results that clearly show effects such as tipping points and feedback mechanisms the made speculations appear either arbitrary or overstated.
Agreed. We removed the last sentence in this paragraph that contained the 'tipping point' statement.

p.19, l.14: What does "multi-disciplinary" mean in this context? Why is this information necessary?
We agree that this term isn't necessary to make the intended statement, and therefore removed it.

p. 19, l.16: It should be pointed out that temperature changes are limited to the upper most decimeters. The differences in soil temperature are only 0.27K in 64 cm depth. This leads to further questions such as: What is the accuracy of the soil temperature sensors used? How accurate were the soil temperature sensors installed in the ground? How was the soil surface defined? What do these uncertainties mean for the soil temperature comparison (also the observed seasonal differences)?
The information of maximum sensor depth for these analyses has been added to the paragraph (but it was also clearly given already on p.9, where these numbers of soil temperatures were presented first).
We agree with the reviewer that soil temperature measurements are subject to uncertainties, with those listed in the statement above being the most important sources for errors. However, given the temperature gradients over depth observed at our sites, we can postulate that a vertical displacement of the sensors as a result of installation bias cannot significantly change the overall picture. Linearly interpolated soil temperature gradients between 32cm and 64cm, as

well as between 64cm and 128cm (which are available at other sensors not used within the context of this study) yield values ranging between -0.006 K/cm and 0 K/cm, therefore even a height bias of 5cm would only lead to a temperature shift up to 0.03K. The absolute calibration of the sensors can be compared during the zero-curtain period in fall, when very stable temperature levels are reached by all sensors for a prolonged period of time. Also through this comparison, we can confirm that no offset exists that can call into question the validity of the overall shifts in soil temperatures found between wet and dry microsites, respectively.

p.19, l.17: This statement should include the wording "most likely" since the study neither presents data on soil heat capacities, thermal conductivities, nor soil heat fluxes.
OK

p.19, l. 30: Why is there a profound impact on forecasts of the sustainability of Arctic permafrost ecosystems under future climate change? Changes in the atmospheric fluxes below 10% might be judged as rather moderate impact. In particular when the natural spatial heterogeneity of tundra is taken into account.
We toned down the statement to "..,the demonstrated effects could therefore be relevant for forecasts of the sustainability of Arctic permafrost ecosystems under future climate change.".

*Note: in the following document, the original comments made by the reviewer are copied in black, while the authors' responses to these comments follow in blue.*

**Authors' response to comments submitted by Reviewer #2**

This study of a long-term drainage experiment with a focus on energy budget effects is interesting and relevant to ongoing efforts to understand how ecosystem change in the Arctic might feedback on belowground and atmospheric processes. I have a number of comments listed below that I hope the authors will find useful.

Pg 2, line 19 – Improve wording. Perhaps, "In addition to warming, shifts in the water balance in this region are also expected to trigger profound …"
We took over the wording suggested by the reviewer.

Pg 3, paragraph starting line 3 – Include a reference to the Kittler et al. Biogeosciences paper that reports on the $CO_2$ flux analysis at this study site.
As suggested by the reviewer, we added a citation for the paper by Kittler et al. (Biogeosciences, 2016), which analyzed the effects of long-term drainage on summertime $CO_2$ fluxes at the Chersky site.

Pg 3, line 14 – Suggest wording change to " …to quantify several secondary disturbance effects linked to lower water tables, including changes in vegetation community, radiation budget and soil thermal regime."
The sentence was changed according to the suggestion.

Pg 3, lines 18-21 – These last 2 sentences repeat the results summarized in the abstract. I suggest these be rephrased in terms of a hypothesis or leave out entirely.
We removed those parts of the sentences in the last paragraph of the introduction that summarized results.

Pg 4, lines 9 – 12 – This last sentence reviewing Merbold et al. (2009) results is not needed in the methods and it is a repeat of information from the introduction. I suggest it be removed.
As suggested by the reviewer, the last sentence was removed.

Pg 6, Section 3.1.2 – It isn't clear to me why this analysis/detail on long-term temperature and precipitation is needed in this manuscript which focuses on the results of a manipulation experiment.
We agree with the reviewer that the inclusion of long-term climate trends for this region is slightly outside the core focus area of this study, i.e. the comparison of energy processes between the two treatments on our site. Still, we believe that information on trends in regional climate is helpful to put the changes associated with the drainage disturbance in perspective. For this purpose, the survey on decadal mean temperatures and interannual variability presented in Section 3.1.2 is very useful.

Pg 7, line 30 – The method used to calculate aerodynamic roughness needs to be included in the methods.
We added the sentence 'Aerodynamic roughness length was derived here based on flux-profile relationships using friction velocity and wind speed at tower top under neutral atmospheric stratification.' to this paragraph.

Pg 9, lines 10-11 – Is it reasonable to attribute the warmer conditions in the dry microsite to deeper snow as a) no snow depths were measured and b) this microsite might be relatively close in space to the wet microsite (as both are within the drained transect)? Could cooler

winter temperatures at the wet microsite also or instead be due to greater thermal conductivity promoting heat loss in the saturated and frozen vs. dry soil? Or do you expect the dry microsite surface peat to become saturated in fall and be relatively similar?

The dry and the wet microsites equipped with additional instrumentation for continuous soil monitoring are approximately 50m apart in both transects. In case of the results presented in Fig. 7, all values were taken from the 2 sites within the drained transect.
We do not expect the dry microsites to become fully saturated with water when the early fall precipitation events occur. We have witnessed shorter periods of time when repeated heavy rainfall events have created inundated conditions at parts of the drained section during summer, but this water is usually removed through lateral export soon after precipitation stops. The difference in water content is also emphasized by the prolonged zero-curtain period at the wet microsites (item 2 in this list)
Since the thermal conductivity of ice even exceeds that of water, we agree with the reviewer that, once all the water in the wet soil has been frozen, the higher ice content compared to the dry microsites should lead to a higher net thermal conductivity. Since, due to missing direct measurements of snow depth at several locations, our hypothesized snow cover effect must remain speculative, we modified the 3. item on this list with a new emphasis on thermal conductivity.

Pg 9, line 15 – Were these results from just the 2 microsites at the control and the 2 microsites at the drained site or dry vs. wet microsites at the drained site only as shown in Figure 7?

These results are based on the comparison of dry vs. wet microsite within the drained transect, i.e. using the same data source as for Fig.7. To clarify this, we amended the first sentence of this paragraph.

Pg 9, line 26 – Perhaps reword to "Here, higher soil moisture promotes higher soil temperatures…"

OK

Pg 12, lines 22 – 27 – Unless I missed it, there was no direct data to support a finding that snow was deeper in the drained site with greater shrub cover (which was also not assessed for height as noted on pg 13, line 1). Make sure to be clear that your data is not direct evidence for snow depth effects or include some additional data that helps support this conclusion.

We acknowledge that there is no observational evidence on shifts in snow cover depth, since there is only one snow level sensor installed at our experimental site. Therefore, we agree with the reviewer that snow depth should be removed from this discussion item. We changed the paragraph accordingly.

Pg 14, line 13 – Can you speculate on the mechanism leading to this difference in H+LE response to reduced net radiation in 2015?

There are two short periods, i.e. in early July and early September, when $R_{net}$ was much higher in 2015 compared to 2014, offsetting parts of the negative difference between the energy budgets of both years. In both cases, LE contributed the largest share of energy flux reaction to this change in energy input, and the additional flux was found much higher in the drained section than in the control section. The largest part of the different magnitude in interannual variability of net energy exchange between the two treatments can be attributed to these few days.
Our interpretation of the underlying mechanism combines two influence factors: First, our data clearly demonstrates that LE is the dominant factor behind both the interannual variability in net energy exchange (for both treatments) and the differences in interannual variability between treatments. Here, we already stated in the manuscript (same section, previous paragraph) that LE flux rates are highly susceptible to day-to-day variability in radiative energy input. Second, since the reaction in LE to variability in $R_{net}$ differ between treatments, it is likely that soil moisture levels have an influence in this process. However, we can only speculate on the reason why the combination of these factors leads to differences in interannual

variability between treatments. A possible explanation is that antecedent moisture levels play a role, i.e. the timing of precipitation events related to the timing of variability (spikes) in $R_{net}$ may play a role. We added the following section to the paragraph:
"We speculate that the deviating interannual variability between the sections may be driven by differences in soil moisture levels between data years, e.g. linked to the timing of precipitation events, which influenced the feedback of LE flux rates to variability in net radiation."

Pg 15, line 12 – But could this heat storage in water be included in the ground heat flux term if the surface is defined as the water surface?
In this study, we ignored heat storage effects in the soil water, and the potential impact of transfer of energy through the lateral export of soil water, which may be particularly important in the drainage section. Since we have no data available that may enable us to quantify this effect, we can only mention it here as a potential source of uncertainty.

Pg 15, line 13 – This was an increase of 8%? In other words, more energy may have been going to soil heat fluxes in 2015 at the control site?
That is correct. To clarify this, we modified the last sub-sentence to '..,values increased by about 8 % in the control section,.. "

Pg 15, line 28 – Any heat transfer impacts in winter?
As already stated in our answer to comment 'p9, l10-11', we agree with the reviewer that differences in the thermal conductivity of ice-filled (more abundant in control section) and air-filled (more abundant in drained section) pores are likely to cause the steeper negative gradients in soil temperatures in the control section in winter, once all soil water has been frozen. In this specific paragraph, we changed a sentence to "..which in our case resulted in a reduction of heat transfer into the soil across the seasons." to acknowledge this fact.

Pg 16, line 15 – Improve wording.
We changed the sentence to "Sophisticated numerical models are needed for assessing the complex feedback processes between permafrost ecosystems and climate change, but is unclear yet which processes need to be explicitly resolved in these models, and which input parameters need to be provided at what resolution, to avoid systematic biases in simulation results."

Pg 17, line 5 – Reword "multiple links with factors" to improve clarity.
We changed this sub-sentence to "..,and numerous studies have identified links with factors such as snow cover (Sturm et al., 2001a; Pomeroy et al., 2006), radiation regime (Bewley et al., 2007) and nutrient cycling (Myers-Smith and Hik, 2013; DeMarco et al., 2014b)."

Pg 17, line 8 – Suggest revise wording to "…that the capture of drifting snow by shrubs increases snow depth and soil temperatures in winter…"
We changed the sentence according to the suggestion.

Pg 19, line 24 – Figure 10 needs to be introduced/discussed before the conclusion and in greater detail. It could instead be removed if preferred.
We decided to remove Figure 10.

Pg 26, line 19 – Suggest wording change to "were amended for this study by flags also reflecting overall…"
OK

Pg 27, line 7 – Remove "e.g."
OK

Pg 34, Fig 4 caption – What does the height of 0 cm reference to?

We arbitrarily chose a zero-level while conducting the terrain surveys in 2013 and 2014. The absolute values therefore do not carry any information. We added this information to the figure caption.

Pg 40, Fig 10 – What do the green quadrilateral shapes represent?
As mentioned above, we removed Figure 10 from the revised manuscript version. Those shapes the reviewer was referring to were meant to represent tussocks, and their shift in abundance and size following the sustained drainage.

---

## Author Response (AR3)

Dear Mr. Gruber,

Thank you very much for guiding this manuscript through the review process. We worked in all the minor technical changes that you suggested in your latest comments, including a change of the manuscript title.

Yours faithfully, Mathias Göckede